# Discovering modular solutions that generalize compositionally

**Simon Schug**[*]
ETH Zurich

**Seijin Kobayashi**[*]
ETH Zurich

**Yassir Akram**
ETH Zurich

**Maciej Wołczyk**
IDEAS NCBR

**Alexandra Proca**
Imperial College London

**Johannes von Oswald**
ETH Zurich
Google Research

**Razvan Pascanu**[†]
Google DeepMind

**João Sacramento**[†]
ETH Zurich

**Angelika Steger**[†]
ETH Zurich

## Abstract

Many complex tasks can be decomposed into simpler, independent parts. Discovering such underlying compositional structure has the potential to enable *compositional generalization*. Despite progress, our most powerful systems struggle to compose flexibly. It therefore seems natural to make models more modular to help capture the compositional nature of many tasks. However, it is unclear under which circumstances modular systems can discover hidden compositional structure. To shed light on this question, we study a teacher-student setting with a modular teacher where we have full control over the composition of ground truth modules. This allows us to relate the problem of compositional generalization to that of identification of the underlying modules. In particular we study modularity in hypernetworks representing a general class of multiplicative interactions. We show theoretically that identification up to linear transformation purely from demonstrations is possible without having to learn an exponential number of module combinations. We further demonstrate empirically that under the theoretically identified conditions, meta-learning from finite data can discover modular policies that generalize compositionally in a number of complex environments. [1]

## 1 Introduction

Modularity pervades the organization of artificial and biological systems. It allows building complex systems from simpler units: objects are constructed from parts, sentences are compositions of words and complex behaviors can be decomposed into simpler skills. Discovering this underlying modular structure promises great benefits for learning. Once a finite set of primitives has been acquired, it can be recomposed in novel ways to quickly adapt to new situations offering the potential for *compositional generalization*. For instance, having learned to solve tasks that require pushing green boxes and tasks that require jumping over red boxes, an agent that has decomposed its skills appropriately, can flexibly solve tasks that require jumping over green boxes and pushing red boxes.

While monolithic architectures, in which the whole network is engaged for all tasks, are powerful, they lack a built-in mechanism to decompose learned knowledge into modules. This puts them at a potential disadvantage as they might have to learn each of the exponentially many combinations explicitly even if their constituent modules were all previously encountered. For instance, it is unclear to what extent large language models can compositionally generalize beyond the training distribution (Srivastava et al., 2023; Press et al., 2023; Dziri et al., 2023).

This raises the question whether an architecture built from independent modules that can be flexibly combined can more easily extract modular structure from a family of tasks with underlying compositional structure. Typically modular architectures are expressive enough to collapse to the monolithic solution of modeling all task combinations encountered during training and often do so in practice

---

[*]Equal contribution; correspondence to {`sschug,seijink`}`@ethz.ch`.
[†]Shared senior authors
[1]Code available at `https://github.com/smonsays/modular-hyperteacher`

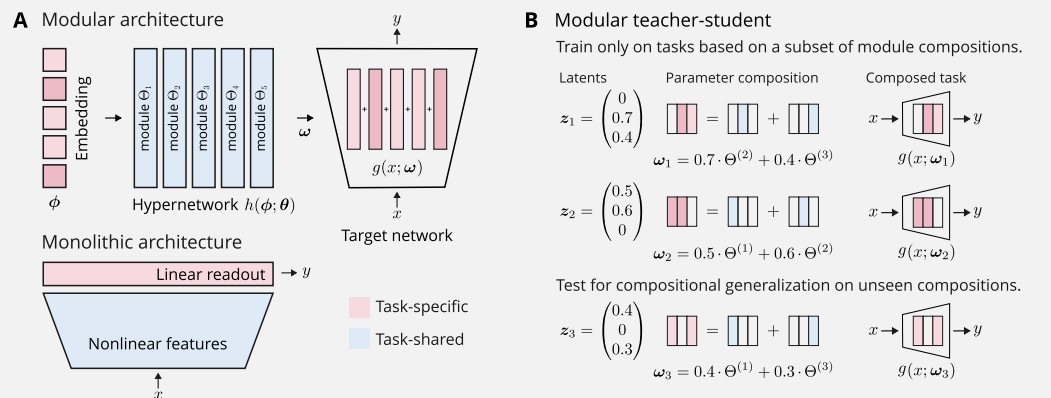

Figure 1: **A** Diagram contrasting a hypernetwork as a modular architecture to a linear combination of features as a monolithic architecture. **B** Toy example illustrating the modular teacher-student setting where tasks are drawn from parameter module compositions of a teacher hypernetwork.

(Shazeer et al., 2017; Kirsch et al., 2018; Rosenbaum et al., 2018; Mittal et al., 2022). Identifying the correct modules is further complicated by the fact that the learning system only ever encounters data generated from mixtures of modules without access to the ground truth decomposition.

Here we aim to illuminate the question what properties of the data generating process allow to discover modular solutions enabling compositional generalization. We do so within the framework of multi-task learning where a learner encounters input/output pairs on a set of tasks composed of few underlying modules. As modular architecture, we consider hypernetworks (Ha et al., 2017), a general form of multiplicative interaction (Jayakumar et al., 2020) which allow to express compositions in weight space used by many multi-task models (Ruvolo & Eaton, 2013; Perez et al., 2018; Dimitriadis et al., 2023; Rame et al., 2023). Our main contributions are as follows:

- We setup a multi-task teacher-student setting with a modular teacher instantiated as a linear hypernetwork that gives us precise control over the underlying modules and their composition. It allows us to cast the problem of compositional generalization as an identification problem.

- We provide a new identifiability result in this setting showing that in the infinite data limit the modules of a perfect student with zero training loss are a linear transformation of the teacher modules. Crucially, observing a linear number of sparse module combinations is sufficient for the student to generalize to any combination of modules.

- In an extended teacher-student setting, we demonstrate empirically how meta-learning from finite data of sparse module combinations can discover modular solutions that generalize compositionally under the theoretically identified conditions in modular but not monolithic architectures.

- We show that our results translate to complex settings of learning action-value functions for compositional preferences and learning of behavior policies for compositional goals.

## 2 METHODS

**Compositionality** To endow our data generating distributions with compositional structure, for each task we recombine sparse subsets of up to $K$ components from a pool of $M$ modules. Each task can be fully described by a latent code $z \in \mathbb{R}^M$ with $\|z\|_0 \leq K$, that specifies the combination of module parameters $(\Theta^{(m)})_{1 \leq m \leq M}$ which parameterizes a task-shared function $g$ with parameters $\omega(z) = \sum_{m=1}^{M} z^{(m)} \Theta^{(m)}$. Being exposed only to demonstrations of such tasks with their underlying modular structure hidden, the naive strategy for a learner would be to learn every combination separately. As the number of modules grows, the quantity of possible combinations increases exponentially, requiring more and more capacity. In contrast, identifying underlying compositional structure not only affords a more compact representation, it also potentially allows the learner to generalize to unseen compositions.

**Modular and monolithic architectures.** For a learning system to be able to leverage the compositional structure of a family of tasks, its model architecture needs to be able to efficiently represent the corresponding class of compositions. Here we contrast modular and monolithic (non-modular)

architectures (see Figure 1A) and investigate under which circumstances they admit functionally modular solutions that reflect the compositional structure of the data generating distribution. To test our models for *compositional generalization*, we hold-out a number of module combinations when we sample tasks for training. At evaluation time we can then test the ability of all models to solve the held-out out-of-distribution (OOD) tasks.

As our modular architecture of choice we consider hypernetworks (Ha et al., 2017) that allow to express a wide class of multiplicative interactions (Jayakumar et al., 2020). A hypernetwork $h$ is a neural network that generates the parameters $\boldsymbol{\omega}$ for a target neural network $g(\boldsymbol{x}; \boldsymbol{\omega})$ which itself processes the task data. The hypernetwork takes as input a task-specific embedding vector $\boldsymbol{\phi}$ and is parameterized by the task-shared parameters $\boldsymbol{\theta}$, i.e. $\boldsymbol{\omega} = h(\boldsymbol{\phi}; \boldsymbol{\theta})$. We can think of $h$ as a function that maps $\boldsymbol{\phi}$ to a selection of flattened module parameters $(\Theta^{(m)})_{1 \leq m \leq M}$. In the case of a *linear hypernetwork*, $h$ is a linear function of $\boldsymbol{\phi}$ resulting in a linear combination of the module parameters. In the case of *nonlinear hypernetworks*, $h$ is a separate neural network whose penultimate layer can be thought of as selecting among the module parameters.

As an example for a monolithic architecture that only supports a less expressive form of composition, we consider the commonly used motif of combining a nonlinear feature encoder shared across tasks with a task-specific linear decoder (e.g. Lee et al., 2019; Bertinetto et al., 2019; Raghu et al., 2020). Here we focus on *ANIL* (Raghu et al., 2020), i.e. $f(\boldsymbol{x}, \boldsymbol{\phi}, \boldsymbol{\theta}) = \boldsymbol{\phi}^\top g(\boldsymbol{x}; \boldsymbol{\theta})$ where $\boldsymbol{\phi}$ is a task-specific readout weight matrix. In addition, we consider the widespread *MAML* architecture (Finn et al., 2017) with $f(\boldsymbol{x}, \boldsymbol{\phi}, \boldsymbol{\theta}) = g(\boldsymbol{x}; \boldsymbol{\phi}(\boldsymbol{\theta}))$ where the task-shared parameters $\boldsymbol{\theta}$ are used to initialize the task-specific parameters $\boldsymbol{\phi}$. For additional details on the architectures please consider Appendix D.

**Meta-learning** Even if a model supports flexible compositional representations, discovering the hidden latent structure of a task distribution simply from observing demonstrations is difficult. A common paradigm to extract structure shared across tasks is meta-learning which we will consider in the following (Finn et al., 2017; Zhao et al., 2020). Formally, we consider a distribution $\mathcal{P}_{\boldsymbol{z}}$ over task latent codes $\boldsymbol{z}$ which has compositional structure. A given task $\boldsymbol{z}$ defines a joint distribution over input $\boldsymbol{x}$ and output $\boldsymbol{y}$, from which two datasets are sampled, $\mathcal{D}_{\boldsymbol{z}}^{\text{support}} = \{x_i, y_i\}_{i=1}^{N^{\text{support}}}$ and $\mathcal{D}_{\boldsymbol{z}}^{\text{query}} = \{x_i', y_i'\}_{i=1}^{N^{\text{query}}}$. We allow the model to infer the task latent code from the demonstrations in the support set $\mathcal{D}_{\boldsymbol{z}}^{\text{support}}$ by optimizing the task-specific parameters $\phi$, obtaining $\phi_{\boldsymbol{z}}(\theta)$, before querying the model on the query set $\mathcal{D}_{\boldsymbol{z}}^{\text{query}}$ to update the task-shared parameters $\theta$. The meta-learning problem can be formalized as a bilevel optimization problem

$$\min_{\theta} \ \mathbb{E}_{\boldsymbol{z} \sim \mathcal{P}_{\boldsymbol{z}}}[L(\phi_{\boldsymbol{z}}(\theta), \theta; \mathcal{D}_{\boldsymbol{z}}^{\text{query}})] \qquad \text{s.t.} \ \ \phi_{\boldsymbol{z}}(\theta) \in \arg\min_{\phi} L(\phi, \theta; \mathcal{D}_{\boldsymbol{z}}^{\text{support}}) \tag{1}$$

where $L$ is a loss function; here we are using the mean-squared error and the cross-entropy loss. To derive our theory, we assume the infinite data limit which allows us to consider a simpler multi-task learning problem with $\mathcal{D}_{\boldsymbol{z}} := \mathcal{D}_{\boldsymbol{z}}^{\text{query}} = \mathcal{D}_{\boldsymbol{z}}^{\text{support}}$ resulting in a single objective

$$\min_{\theta} \ \mathbb{E}_{\boldsymbol{z} \sim \mathcal{P}_{\boldsymbol{z}}} \left[ \min_{\phi_{\boldsymbol{z}}} \ L(\phi_{\boldsymbol{z}}, \theta; \mathcal{D}_{\boldsymbol{z}}) \right]. \tag{2}$$

We optimize it in practice by running gradient descent on the task-specific variable $\phi_{\boldsymbol{z}}$ until convergence for each gradient update of the task-shared parameters $\theta$. In contrast to the bilevel optimization, no second order gradients need to be computed. At inference, we consider a new distribution $\mathcal{P}_{\boldsymbol{z}}^{\text{test}}$, for which the support can be disjoint with that of $\mathcal{P}_{\boldsymbol{z}}$, and report the expected loss $\mathbb{E}_{\boldsymbol{z} \sim \mathcal{P}_{\boldsymbol{z}}^{\text{test}}}[\min_{\phi_{\boldsymbol{z}}} \ L(\phi_{\boldsymbol{z}}, \theta; \mathcal{D}_{\boldsymbol{z}})]$.

## 3 THEORY

We seek to understand what properties of the data generating distribution allow a modular architecture to discover a functionally modular solution and thereby enable compositional generalization. The core idea is to extend the classic teacher-student problem to a multi-task setting, where the teacher has a compositional representation. This allows us to relate the problem of compositional generalization to that of parameter identification in the student as a necessary and sufficient condition. For both the teacher and the student we choose linear hypernetworks (Ha et al., 2017) which afford a natural interpretation in terms of module composition. Each task can then be described concisely given a task latent variable that conditions the teacher hypernetwork to generate demonstrations given the selected sparse, linear combination of modules. We show that in the infinite data

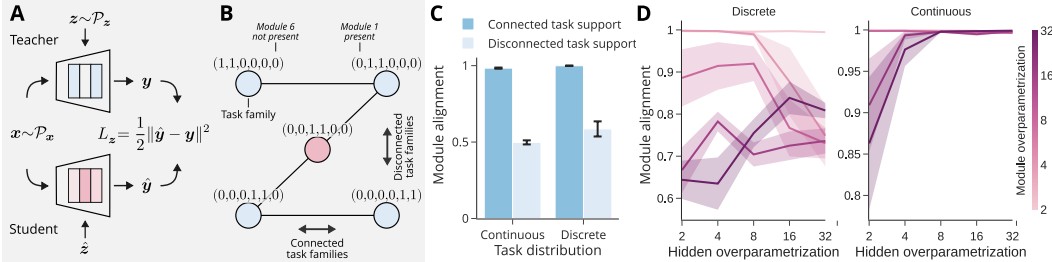

Figure 2: **A** Schematic of the multi-task teacher-student setup. **B** Visualization of connected and disconnected task distributions in a teacher with $M = 6$, $K = 2$. Without the central node support would be disconnected. **C** Module alignment between the student and teacher is high for both continuous and discrete task distributions with connected but not disconnected support. **D** Module alignment is sensitive to overparameterization. Numbers denote the factor by which the student dimension is larger than the teacher. Error bars in C,D denote standard error of the mean for 3 seeds.

limit the student can identify the teacher modules up to linear transformation without access to the task latent variable if the following conditions are satisfied:

(i) The training distribution has *compositional support*, i.e. all modules we wish to compose in the OOD task have been encountered at least one training task (c.f. Wiedemer et al., 2023).

(ii) The training distribution has *connected support*, i.e. no subset of modules appears solely in isolation from the rest, as otherwise we cannot guarantee modules can be consistently composed in the student.

(iii) *No overparameterization* of the student wrt. the teacher. For instance, if the student has more modules available than there are tasks in the training distribution, it could use one module per task, fitting all of them perfectly but remaining ignorant of their shared structure.

### 3.1 MULTI-TASK TEACHER-STUDENT SETUP

In a standard teacher-student setting, a student network is trained to match the output of a teacher given inputs $\boldsymbol{x} \in \mathbb{R}^n$ drawn from some distribution $\mathcal{P}_{\boldsymbol{x}}$. Here, we consider a multi-task counterpart to this setting (Figure 2A). In addition to the input, we sample a hidden task latent variable $\boldsymbol{z}$ from some distribution $\mathcal{P}_{\boldsymbol{z}}$ independently from $\boldsymbol{x}$. In the following, we denote all quantities relating to the student with a hat, e.g. $\hat{\Theta}$, and those relating to the teacher without.

For both the teacher and student architecture we use task-conditioned hypernetworks (c.f. Figure 1A). Specifically, we consider a 2-layer multi-layer perceptron (MLP) with $h$ hidden units and one output unit, for which the first layer weights are generated by a linear hypernetwork as

$$f(\boldsymbol{x}; \boldsymbol{a}, \Theta, \boldsymbol{z}) = \boldsymbol{a}^\top \psi(\boldsymbol{W}(\Theta, \boldsymbol{z})\boldsymbol{x}) \tag{3}$$

where $\psi$ is an activation function, $\boldsymbol{a}$ is the task-shared readout vector, and $\Theta = (\Theta^{(m)})_{1 \leq m \leq M}$ are the parameter modules parameterizing the hypernetwork. The first layer weight $\boldsymbol{W}$ is generated by linearly combining the $(\Theta^{(m)})_{1 \leq m \leq M}$ with the task latent variable $\boldsymbol{z} = (z^{(1)}..z^{(M)})^\top$, i.e. $\boldsymbol{W}(\Theta, \boldsymbol{z}) = \sum_m^M z^{(m)} \Theta^{(m)}$. The teacher then produces task-conditioned demonstrations as in Equation 3. The task thus has a compositional structure as defined in Section 2.

The student is also implemented as a linear hypernetwork with the hidden dimension potentially different from the teacher. Crucially, the student only has access to the input $\boldsymbol{x}$ but does not observe the task latent variable $\boldsymbol{z}$. We allow the student however to infer a task-specific parameter $\hat{\boldsymbol{z}}$ of dimension $\hat{M}$ for each task given contextual information which will then condition the forward pass. We estimate $\hat{\boldsymbol{z}}$ such that it minimizes the objective detailed in Section 2, given demonstrations from the teacher. The student then optimizes the multi-task objective stated in Equation 2, where given the task shared parameters $\theta = (\hat{\Theta}, \hat{\boldsymbol{a}})$, and the task-specific parameters $\phi_{\boldsymbol{z}} = \hat{\boldsymbol{z}}$, the task loss is defined as

$$L(\hat{\boldsymbol{z}}, (\hat{\Theta}, \hat{\boldsymbol{a}}); \mathcal{D}_{\boldsymbol{z}}) = \frac{1}{2}\mathbb{E}_{\boldsymbol{x}}\big[\|f(\boldsymbol{x}; \hat{\boldsymbol{a}}, \hat{\Theta}, \hat{\boldsymbol{z}}) - f(\boldsymbol{x}; \boldsymbol{a}, \Theta, \boldsymbol{z})\|^2\big] \tag{4}$$

### 3.2 IDENTIFICATION & COMPOSITIONAL GENERALIZATION

We will now show under what conditions a student, that perfectly fits the teacher on the training distribution, recovers the teacher's modules and achieves compositional generalization.

Let us first study a motivating example for a teacher with $M = 6$ modules illustrated in Figure 2B to build some intuition. For that purpose we consider task families associated with binary masks that indicate which sparse subset of $K = 2$ modules is present. For instance, the binary mask $\boldsymbol{b}_1 = (1, 1, 0, 0, 0, 0)^\top$ indicates that $Z_1$ is a task family consisting of latent vectors for which exactly the first and second entry are non-zero, e.g. $\boldsymbol{z} = (0.6, 0.7, 0, 0, 0, 0)^\top$.

We now consider the case where during training the student observes tasks from four different task families specified by the binary masks $\boldsymbol{b}_1 = (1, 1, 0, 0, 0, 0)^\top, \boldsymbol{b}_2 = (0, 1, 1, 0, 0, 0)^\top, \boldsymbol{b}_3 = (0, 0, 0, 1, 1, 0)^\top, \boldsymbol{b}_4 = (0, 0, 0, 0, 1, 1)^\top$. For each task, we assume that the student sees an infinite number of samples covering the whole input space and learns to perfectly fit the teacher. After training we ask the student to solve a new task specified by the task latent code $\boldsymbol{z}_? = (1, 0, 0, 0, 0, 1)^\top$. Given that it has seen all modules present in this new task, we might expect it to be able to solve it. It turns out that this is not the case since the neuron permutations between modules indexed by $\{1, 2, 3\}$ and $\{4, 5, 6\}$ are likely inconsistent as it has never encountered a task that *connects* these two subsets of modules (also see Appendix A.3 for a more detailed explanation). We can avoid this pathology by adding a fifth task family to our set of training tasks given by the binary mask $\boldsymbol{b}_5 = (0, 0, 1, 1, 0, 0)^\top$. As we will show now, this leads to *compositional generalization* and the student will perfectly solve any task including $\boldsymbol{z}_?$. In fact, the student modules will be a linear combination of the teacher modules which we refer to as *linear identification*.

Let us now consider a general set of task families $Z = \{Z_1, Z_2, \dots\}$ where each task family $Z_k \in Z$ is a non-empty set of $\mathbb{R}^M$ with an associated binary mask $\boldsymbol{b}_k$. In the following we require that $\mathrm{span}(Z_k) = \{\boldsymbol{v} \odot \boldsymbol{b}_k \mid \boldsymbol{v} \in \mathbb{R}^M\}$, where $\mathrm{span}(A)$ denotes the set of linear combinations of elements of $A$. With this definition at hand we can formulate sufficient conditions on the training distribution $\mathcal{P}_{\boldsymbol{z}}$ over task latent variables whose support contains the union of all task families $Z$.

**Definition 3.1 (Compositional support).** $\mathcal{P}_{\boldsymbol{z}}$ has compositional support if all dimensions of $\mathbb{R}^M$ are covered by at least one of the binary masks $\boldsymbol{b}_k$, i.e. $\sum_k^{|Z|} \boldsymbol{b}_k$ has no entry that is 0.

Intuitively having compositional support simply ensures that all modules have been encountered at least once. As we have seen in the example above this is not enough, leading to our second condition.

**Definition 3.2 (Connected support).** $\mathcal{P}_{\boldsymbol{z}}$ has connected support if there exists a path between any two task families, where two task families $Z_k, Z_l$ are said to be connected if their associated binary masks share a non-zero element, i.e. $\boldsymbol{b}_k \wedge \boldsymbol{b}_l \neq 0$.

With these two definitions we can now state a simplified version of our main result showing that a perfect student will generalize compositionally. Formal versions of the following theorems as well as their proofs can be found in Appendix A.4 and A.2, shown for both ReLU as well as a general class of smooth activation functions.

**Theorem 1 (Compositional generalization, informal).** *Assuming that $\mathcal{P}_{\boldsymbol{z}}$ has compositional and connected support, $\mathcal{P}_{\boldsymbol{x}}$ has full support in the input space and the student dimensions match that of the teacher, i.e. $\hat{M} = M \leq n, \hat{h} = h$, then under an additional smoothness condition on $\mathcal{P}_{\boldsymbol{z}}$ and non-degeneracy of $(\Theta, \boldsymbol{a})$, we have that*

$$\mathbb{E}_{\boldsymbol{z} \sim \mathcal{P}_{\boldsymbol{z}}} \left[ \min_{\hat{\boldsymbol{z}}} L(\hat{\boldsymbol{z}}, (\hat{\Theta}, \hat{\boldsymbol{a}}); \mathcal{D}_{\boldsymbol{z}}) \right] = 0 \quad \implies \quad \min_{\hat{\boldsymbol{z}}} L(\hat{\boldsymbol{z}}, (\hat{\Theta}, \hat{\boldsymbol{a}}); \mathcal{D}_{\boldsymbol{z}}) = 0 \quad \forall \boldsymbol{z} \in \mathbb{R}^M,$$

*i.e. if the student optimizes $(\hat{\Theta}, \hat{\boldsymbol{a}})$ to fit the teacher on $\mathcal{P}_{\boldsymbol{z}}$, it achieves compositional generalization.*

*Furthermore, if the student achieves zero loss on a finite number, $N = \dim(\mathrm{span}(Z_k)) + 1$, of i.i.d. samples of tasks for each $Z_k$ from $\mathcal{P}_{\boldsymbol{z}}$ it will generalize compositionally almost surely.*

This result is noteworthy since it implies that fitting a number of tasks linear in the number of modules suffices to generalize to all exponentially many module combinations.

The assumptions of compositional, connected support and $h = \hat{h}$ are not only sufficient, but also necessary conditions for the implication to hold (all other things being equal), since we can construct counterexamples, detailed in Appendix A.3, when one of the conditions is violated. Furthermore, achieving compositional generalization under the above assumptions is equivalent to *linear identification* of the teacher modules in the student. For any hidden neuron index $i$, we denote by $\Theta_i$ the slice of $\Theta$ of dimension $n \times M$ which maps the task latent variable $\boldsymbol{z}$ to $\boldsymbol{w}_i$, i.e. $\boldsymbol{w}_i = \Theta_i \boldsymbol{z}$.

**Theorem 2 (Linear identification,** informal**).** *Assuming $\mathcal{P}_{\boldsymbol{x}}$ has full support in the input space, $\hat{M} = M \leq n$, $\hat{h} = h$, non-degeneracy of $\Theta$ and $\boldsymbol{a}$, then $\min_{\hat{\boldsymbol{z}}} L(\hat{\boldsymbol{z}}, (\hat{\Theta}, \hat{\boldsymbol{a}}); \mathcal{D}_{\boldsymbol{z}}) = 0 \quad \forall \boldsymbol{z}$ is equivalent to the existence of an invertible matrix $\boldsymbol{F}$, a permutation $\sigma$ and a sign flip $\epsilon \in \{-1, 1\}^h$ such that*

$$a_{\sigma i} \Theta_{\sigma(i)} = \epsilon_i \hat{a}_i \hat{\Theta}_i \boldsymbol{F} \quad \forall i \in [1, h]$$

*i.e. the learned student modules are a linear combination of the teacher modules.*

## 3.3 EMPIRICAL VERIFICATION IN THE THEORETICAL SETTING

We empirically verify our theoretical findings by randomly initializing a teacher model following Section 3.1, and measuring the degree of identification achieved by the student network after training until convergence on various task distributions. See Appendix C.1 for details on the experiments. We measure module alignment as $\min_i(\max_j |s(\Theta_i, \hat{\Theta}_j \boldsymbol{F})|)$ where $s$ is the cosine similarity function, and $\boldsymbol{F}$ is obtained by regressing $\hat{\boldsymbol{z}}$ on $\boldsymbol{z}$. A module alignment of 1 implies linear identification.

**Identification requires connected support.** We construct task distributions with both connected and disconnected support. Figure 2C shows that the student parameter modules have a high alignment both for a continuous ($\boldsymbol{z} \in \mathbb{R}^m$) and discrete ($\boldsymbol{z} \in \{0, 1\}^m$) task distribution when the task support is connected. Disconnecting the task support leads to a noticeable degradation.

**Identification is sensitive to overparameterization.** Given a task distribution with compositional and connected support, we investigate the effect of overparameterization on identification. Specifically, we vary the ratio of the number of hidden units $\hat{h}$ and modules $\hat{M}$ of the student with respect to the teacher. While Theorem 1 holds only for $\hat{h} = h$ and $M = \hat{M}$, optimizing the loss well enough typically requires slight overparameterization. Figure 2D shows that a slight overparameterization is indeed beneficial for linear identification. Yet, beyond $\hat{h} = 8h$ and $\hat{M} = 2M$, the identification noticeably degrades for discrete training tasks and cannot be alleviated by adjusting the weight decay (c.f. Table A1 in Appendix B.1). For a continuous task distribution, overparameterization appears unproblematic, indicating that the learning dynamics introduce a beneficial inductive bias.

## 4 EXPERIMENTS

We now empirically investigate if modular architectures realized as hypernetworks can compositionally generalize when meta-learning from finite data, assess whether a highly expressive monolithic architecture can match this performance, and determine if abstract latent space compositionality can be discovered despite an unknown functional mapping to the task data. We further verify that both theoretically identified conditions - compositional and connected task support - are important for compositional generalization beyond the teacher-student setting.

### 4.1 HYPERTEACHER

We generalize the teacher-student setup from Section 3.1 to a teacher hypernetwork that modularly parameterizes multiple layers of a target neural network and only show a limited amount of data to the student per task. To more faithfully reflect the symbolic nature of many real-world compositional tasks, each task is a sparse and discrete combination of modules as exemplified in Figure 3A. See Appendix C.2 for more details. As introduced in Section 2, we contrast monolithic architectures - namely ANIL and MAML - to modular architectures: the nonlinear and linear hypernetwork, see Appendix D for a detailed description of all models. The nonlinear hypernetwork specifically allows us to investigate the effects of architectural mismatch with the teacher network.

**Modular but not monolithic architectures compositionally generalize.** To test the ability for compositional generalization, we hold out a fraction of the possible teacher module combinations during training. Only if we ensure that all teacher modules part of the out-of-distribution (OOD) set have been encountered individually as part of another combination during training does our training distribution have compositional support. Figure 3B shows that both the nonlinear and linear hypernetwork achieve high OOD accuracy while ANIL and MAML fail to do so. When the training support is non-compositional the performance of the hypernetworks drops below that of ANIL and MAML. Figure 3E demonstrates that this holds as long as the number of modules chosen for task combinations is $K > 1$. In the case where exactly one module is chosen per task, $K = 1$, the

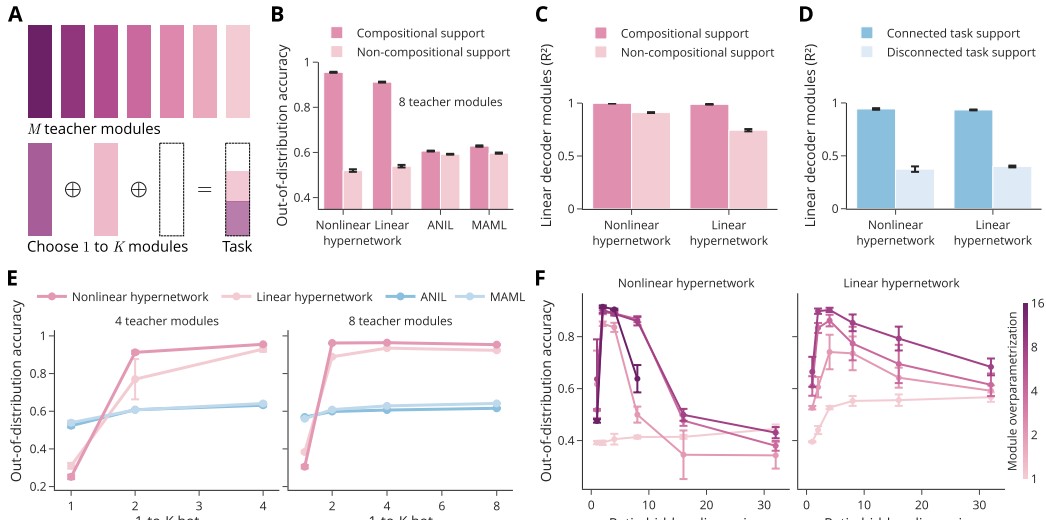

Figure 3: **A** In the hyperteacher we pick between 1 to $K$ of the $M$ teacher modules, adding them in parameter space to create a task. **B** ANIL and MAML fail to generalize to OOD tasks regardless of the support of the training distribution, while hypernetworks achieve good OOD accuracy only when the task support is compositional. **C+D** When the task distribution is compositional and connected the teacher modules can be linearly decoded from the student task embeddings. **E** Hypernetworks have high OOD accuracy for $K > 1$ but low OOD accuracy for $K = 1$. **F** OOD performance of hypernetworks is sensitive to overparameterization in both the hidden dimension and module dimension. Error bars in B-F denote the standard error of the mean over 3 seeds.

condition of *connected support* is not satisfied and as predicted by our theory, both the nonlinear and linear hypernetwork perform poorly in terms of compositional generalization despite the task support being compositional.

**Modules can be linearly decoded in compositional solutions given connected task support.** Given our theoretical result, we expect that a linear hypernetwork that achieves high OOD accuracy to recover the module parameters of the teacher up to linear transformation. To test this we train a linear decoder to predict the ground truth task latent variables of a task given the learned embeddings of the student on a validation set and then test linear decoding performance on the OOD set. Shown in Figure 3C,D, we find that in line with the theory, we get high decodability with an $R^2$ score close to one in the case where the task support is compositional and connected. In the case where the support is non-compositional, the decoding performance is reduced reflecting the fact that there was no opportunity to learn about one of the modules. Despite having seen all modules during training, performance further drops significantly when the task support is disconnected as predicted by our theory. Surprisingly, these results extend to the nonlinear hypernetwork where there is an architectural mismatch between the teacher and the student and there is no prior reason for the relationship between learned embeddings and ground-truth task latent variables to be linear.

**Compositional generalization is sensitive to overparameterization.** The theory in the discrete setting requires the student to have the same hidden and module dimension as the teacher. We test how sensitive the ability for compositional generalization in terms of OOD performance is to overparameterization on these two axes. In Figure 3F, we observe that while a certain overparameterization is necessary for the optimization to succeed, performance starts to decrease for larger overparameterization.

**Meta-learning overfits to the number of modules entering the composition.** Despite linear identification as detailed above, we further observe that learners overfit to the particular number $K$ of modules combined within a task, with OOD accuracy decreasing for $K$ larger than what was encountered during training, e.g. see Table A6. Preliminary experiments suggest that this can be partially alleviated by allowing for more gradient steps to perform task inference during evaluation, but does not present a full remedy of the issue.

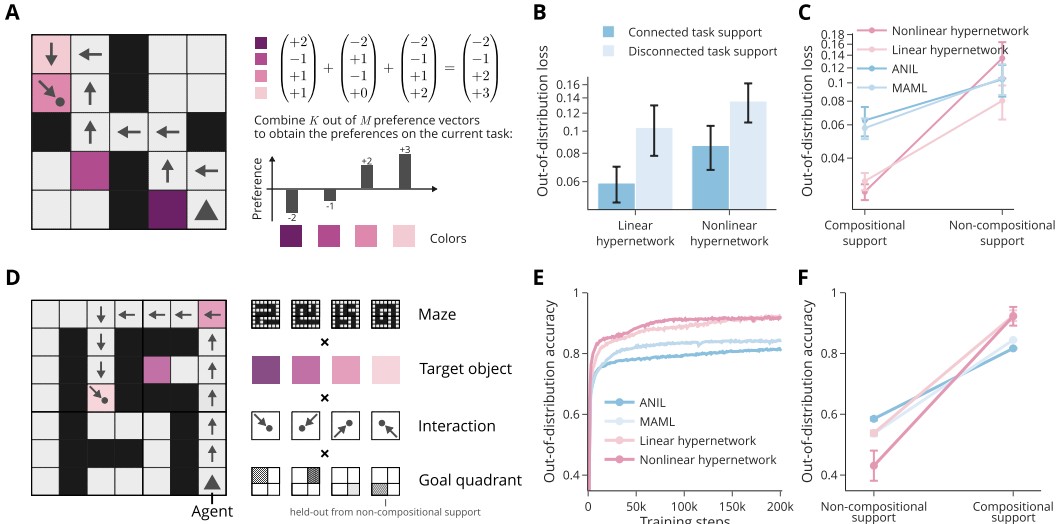

Figure 4: **A** In the compositional preference grid world the agent has modular preferences over colors and gets a reward corresponding to the current preference for the color of an object. **B** Disconnecting the task support increases the OOD loss of hypernetworks. **C** Hypernetworks achieve better OOD loss than ANIL and MAML when the task support is compositional. **D** In the compositional goal grid world an agent has to walk to a target object and perform the correct target action. Goals are a composition of the maze, target object, target action and goal quadrant. **E** Hypernetworks achieve better OOD accuracy wrt to the optimal policy than ANIL and MAML. **F** When holding out one of the goal quadrants the OOD accuracy decreases for hypernetworks more strongly than for ANIL and MAML. Error bars in B,C,F denote the standard error of the mean over 3 seeds.

## 4.2 COMPOSITIONAL PREFERENCES

In the teacher-student setup we ensure by construction that the compositional representation of the data generating process is contained in the class of compositions the student can efficiently express. We now consider a setting where an agent has compositional preferences shared across tasks but the functional mapping to its action-value function for each specific task is unknown. In particular, we study the compositional preference grid world inspired by Barreto et al. (2020) where an agent can obtain rewards by collecting objects with different colors. For each task, up to $K = 3$ out of $M = 8$ preference modules are sampled and summed to obtain a hidden preference vector which specifies the reward the agent obtains when moving to an object of a specific color (see Figure 4A). We test the ability for compositional generalization by presenting the agent with tasks of unseen combinations of preference modules. The agent then has to infer the current preferences from observing optimal trajectories and is evaluated to predict the action-value function for a new environment configuration where agent and object locations are resampled. For more details on the task see Appendix C.3.

**Modular architectures learn composable action-value functions.** Figure 4C demonstrates that the linear and nonlinear hypernetwork achieve a lower mean-squared error loss predicting action-values on OOD tasks compared to ANIL and MAML when the task support is compositional. The OOD loss significantly increases when one of the preference vectors is held-out from the training set, resulting in non-compositional support.

**Encountering preference vectors in disjoint clusters interferes with compositional generalization.** We can emulate disconnected task support similar to our definition in the teacher-student setting by constructing the training tasks to contain subsets of preference vectors in disconnected clusters. We observe that this intervention has a noticeable effect on the OOD loss achieved by both the linear and nonlinear hypernetwork, as shown in Figure 4B, despite the number of observed combinations being comparable for the connected and disconnected support.

## 4.3 COMPOSITIONAL GOALS

Many real-world tasks have compositional goals raising the question whether a modular architecture can discover behavior policies that decompose according to the latent goal structure. To this end, we consider the compositional goal grid world depicted in Figure 4D where an agent has to navigate to a target object to employ a specific target action. The goals have compositional structure and we can test for compositional generalization by presenting novel compositions of goals. A goal is specified

by four factors: maze layout, target object, target interaction and goal quadrant. For example a possible goal would be "*In maze 1, perform interaction 3 on the purple object in quadrant 4*". We measure the accuracy of the agent with respect to the optimal policy. For a more detailed task description see Appendix C.4.

**Modular architectures learn composable behavior policies.** Figure 4E demonstrates that both the linear and nonlinear hypernetwork achieve higher OOD accuracy of predicting the optimal behavior policy on held-out goal compositions than ANIL and MAML.

**Modular architectures outperform monolithic ones by leveraging compositional structure.** To construct non-compositional support in this task, we hold-out one of the goal quadrants from the training distribution. Figure 4F shows that this intervention has a more pronounced effect on the linear and nonlinear hypernetwork than on ANIL and MAML, indicating that generalization to OOD tasks is due to the hypernetworks leveraging the compositional structure of the task.

## 5 RELATED WORK & DISCUSSION

The findings we present here are grounded in a theoretical teacher-student analysis (Gardner & Derrida, 1989) building on recent results that characterize the global minima of the loss of MLPs in the infinite data limit (Tian, 2020; Simsek et al., 2021). We extend the classical teacher-student setting to the multi-task setting where in addition to matching the teacher, the student needs to infer a hidden task latent variable and we instantiate the teacher as a linear hypernetwork to create compositional task structure. We show that parameter identification is both a necessary and sufficient condition for compositional generalization and we prove a new identifiability result that requires the task support of the training distribution to be *compositional* and *connected* - two conditions we empirically verify to be crucial for compositional generalization.

Our results complement Lachapelle et al. (2023) who demonstrate that L1-regularization enables disentangled identification in multi-task learning of sparse, linear feature combinations. Generally, there has been widespread interest in assessing compositional generalization (Lake & Baroni, 2018; Ruis et al., 2020; Hupkes et al., 2020) as a crucial shortcoming of deep learning (Loula et al., 2018; Dessì & Baroni, 2019; Keysers et al., 2020) across various subfields (Battaglia et al., 2018; Atzmon et al., 2020; Montero et al., 2021; Liu et al., 2023). While with the advent of pretrained large language models, compositional abilities have seemingly come into reach (Zhou et al., 2023; Orhan, 2022; Furrer et al., 2021; Csordás et al., 2021) and many specialized architectures (Russin et al., 2019; Li et al., 2019; Gordon et al., 2020; Andreas, 2020; Liu et al., 2020; Lake, 2019; Nye et al., 2020; Kaiser & Sutskever, 2016) are outperformed by pretrained large language models (Furrer et al., 2021), it remains an open question whether the ability for compositional generalization extends beyond the training distribution (Srivastava et al., 2023; Press et al., 2023; Dziri et al., 2023).

Different from prior work we put particular emphasis on the multi-task setting and compositionality in weight space - a common and powerful motif in deep learning (Kumar & Daume III, 2012; Ruvolo & Eaton, 2013; Perez et al., 2018; Jayakumar et al., 2020; Dimitriadis et al., 2023). We show that linearly combining parameters can capture nonlinear latent structure raising an important question for future research on the interpretability of the resulting modules. We derive our theory in the infinite data regime while in practice we resort to meta-learning from finite samples. Indeed the sample complexity appears to scale unfavorably as the number of teacher modules increases (c.f. Figure A3). Running meta-learning on this scale is computationally expensive and future work would ideally amortize the inference procedure when scaling to larger problem instances (Zhmoginov et al., 2022). Moreover, while our identification result in the multi-task teacher-student setting indicates that overparameterization of the student can be detrimental for compositional generalization, the situation is less clear outside the teacher-student setting necessitating further investigation. Finally, we note that we characterized the solutions of the modular teacher-student setting at equilibrium in the infinite data limit. Even when the modular solution constitutes a global optimum, we cannot provide any guarantees that it can be reached with gradient-based learning. Jarvis et al. (2023) have made an important first step towards analyzing the learning dynamics of modular and monolithic architectures using deep linear networks in synthetic, linearly-separable datasets. Understanding and analyzing the learning dynamics of the full modular teacher-student setting, as has previously been done for more restricted architectures (Seung et al., 1992; Saad & Solla, 1995; Goldt et al., 2019), remains an important open question, the answer to which might help elucidate why even modular architectures often collapse to monolithic solutions.

ACKNOWLEDGEMENTS

We would like to thank Flavio Martinelli and Wulfram Gerstner for fruitful discussions and Nino Scherrer for valuable feedback on the manuscript. Simon Schug would like to kindly thank the TPU Research Cloud (TRC) program for providing access to Cloud TPUs from Google. This research was supported by an Ambizione grant (PZ00P3_186027) from the Swiss National Science Foundation and an ETH Research Grant (ETH-23 21-1). Alexandra Proca is funded by the Imperial College London President's PhD Scholarship.

AUTHOR CONTRIBUTIONS

This paper was a collaborative effort. To do this fact better justice we give an idea of individual contributions in the following.

**Simon Schug**[*] Original idea, conceptualization of the project, development of theory, experimental design, implementation, running experiments, writing of the main text, writing of the appendix, literature review, creation of figures.

**Seijin Kobayashi**[*] Original idea, conceptualization of the project, development of theory, proving main theorems, experimental design, implementation, running experiments, helped writing of the main text, writing of the appendix, helped creation of figures.

**Yassir Akram** Helped develop theory, helped proving main theorems, provided feedback on the manuscript, helped writing of the appendix.

**Maciej Wołczyk** Worked on a precursor to this project whose negative results formed the basis for the current project, provided feedback in regular project meetings, provided feedback on the manuscript.

**Alexandra Proca** Worked on a precursor to this project whose negative results formed the basis for the current project, provided feedback on the manuscript.

**Johannes von Oswald** Worked on a precursor to this project whose negative results formed the basis for the current project, conceptualization of the project, provided feedback in regular project meetings, provided feedback on the manuscript.

**Razvan Pascanu**[†] Senior project supervision, conceptualization of the project, provided feedback on the theory, provided feedback on experimental design, provided feedback in regular project meetings, provided feedback on the manuscript.

**João Sacramento**[†] Senior project supervision, original idea, conceptualization of the project, helped develop theory, experimental design, provided feedback in regular project meetings, provided feedback on the manuscript.

**Angelika Steger**[†] Senior project supervision, conceptualization of the project, development of theory, helped proving main theorems, provided feedback in regular project meetings, provided detailed feedback on the manuscript.

REPRODUCIBILITY STATEMENT

To ensure the reproducibility of our work, we are providing the code for all our experiments online at `https://github.com/smonsays/modular-hyperteacher` and complement the main text with experimental details in Appendix C. For the theoretical results, we are complementing our intuitive exposition in the beginning of Section 3 and empirical verification in Section 3.3 with detailed proofs in Section A and example failure cases to build intuition in Section A.3.

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

# Appendix

# Table of Contents

# A    THEORETICAL RESULT

Before presenting the theoretical result for the multi-task teacher-student setting, we first focus on the standard teacher student setting, which can be seen as a special case of our multi-task setting where there exists only a single task.

We will use the notation introduced in Section 3.1 to describe a network, in particular for any hidden neuron index $i$, we denote by $\boldsymbol{w}_i$ the $i$-th row of the generated weight $\boldsymbol{W}(\Theta, \boldsymbol{z})$, $\Theta_i$ the slice of $\Theta$ of dimension $n \times M$ which maps the task latent variable $\boldsymbol{z}$ to $\boldsymbol{w}_i$, i.e. $\boldsymbol{w}_i = \Theta_i \boldsymbol{z}$, and $a_i$ the $i$-th column of $\boldsymbol{a}$.

## A.1    SINGLE TASK PARAMETER IDENTIFICATION FOR ReLU MLP

This section presents rigidity results that allows us, under the right conditions, to identify the learned parameters of a student MLP learning to imitate the outgput of a teacher. Here the student may have the same or even a wider architecture. To our knowledge this is the first such results for ReLU two-layer MLP.

There are clearly many functionally equivalent ways to implement a neural network. In a first step we thus aim at characterizing the exact set of neural networks that are functionally equivalent to to a given teacher network. In order to get a nice characterization we require the teacher to be minimal. E.g., if the read-out value of a hidden unit is zero, this unit has no influence on the output and we can delete it. Formally, we assume that the teacher is irreducible:

**Definition A.1. Irreducibility**: The weights $(\boldsymbol{W}, \boldsymbol{a})$ of a two-layer MLP $f$ are called irreducible when there exists no two-layer MLP $\hat{f}$ with strictly less hidden units such that they are functionally equivalent, i.e. $\forall \boldsymbol{x} \in \mathbb{R}^n, f(\boldsymbol{x}) = \hat{f}(\boldsymbol{x})$.

Clearly, we have the following necessary conditions for irreducibility:

- no output weight $a_k$ is 0
- no input weight vector $\boldsymbol{w}_k$ is 0
- no two input weight vectors $\boldsymbol{w}_k, \boldsymbol{w}_l$ are identical

For a special class of smooth activation functions, $\psi \in \mathcal{C}^\infty$, and $\psi^{(n)}(0) \neq 0$ for infinitely many even and odd values of $n$, Simsek et al. (2021) show that the above conditions are sufficient for a network to be irreducible.

For the ReLU nonlinearity, the above conditions are no longer sufficient for a teacher to be irreducible. As an illustration, if a pair of weights $\boldsymbol{w}_k, \boldsymbol{w}_l$ are positively colinear, then the 2 corresponding hidden neurons can be fused into a single hidden neuron without functionally changing the network. See section A.1.2 for more details.

A simple sufficient condition for the irreducibility of a ReLU MLP is that the teacher has no two input weights $\boldsymbol{w}_k$ that are colinear, and no output weight $a_k$ that is 0. While this is not a necessary condition, we will restrict our identification results to such teacher networks for simplicity, cf. Section A.1.2 for the general case.

### A.1.1    IDENTIFICATION WITH ReLU MLP

We now state the main theorem of this subsection. We will first treat the theorem for the special case that teacher and student network have the same number of hidden units, i.e. $\hat{h} = h$. Subsequently, we will then state the theorem for $\hat{h} \geq h$ and illustrate how to extend the proof. From here on, whenever we talk about a measure, we refer to the Lebesgue measure.

**Theorem 3.** *Assume a teacher and student two-layer network parameterized by resp. $(\boldsymbol{W}, \boldsymbol{a})$ and $(\hat{\boldsymbol{W}}, \hat{\boldsymbol{a}})$, of hidden dimensions $h = \hat{h}$ and a single output unit such that*

$$\mathbb{E}_{\boldsymbol{x} \sim \mathcal{P}_x}[L(\boldsymbol{x})] = 0, \tag{5}$$

*where*

$$L(\boldsymbol{x}) = \frac{1}{2}\|\boldsymbol{a}^\top \psi(\boldsymbol{W}\boldsymbol{x}) - \hat{\boldsymbol{a}}^\top \psi(\hat{\boldsymbol{W}}\boldsymbol{x})\|^2 \tag{6}$$

*and $\psi$ is the ReLU nonlinearity. If*

- *$\mathcal{P}_x$ has full support in the input space and*

- *$(\boldsymbol{W}, \boldsymbol{a})$ is such that*

  - *no output weight $a_k$ is 0*
  - *no two input weights $\boldsymbol{w}_k, \boldsymbol{w}_l$ are colinear*

*then there exists a partition $\mathcal{S}_1, \mathcal{S}_2$ of $[1..h]$ and a permutation $\sigma$ of the neurons in the hidden layer such that we have*

$$\forall i, \hat{a}_{\sigma(i)} a_i > 0, \tag{7}$$
$$\forall i \in \mathcal{S}_1, \hat{a}_{\sigma(i)} \hat{\boldsymbol{w}}_{\sigma(i)} = a_i \boldsymbol{w}_i, \tag{8}$$
$$\forall i \in \mathcal{S}_2, \hat{a}_{\sigma(i)} \hat{\boldsymbol{w}}_{\sigma(i)} = -a_i \boldsymbol{w}_i, \tag{9}$$
$$\sum_i a_i \boldsymbol{w}_i = \sum_i \hat{a}_i \hat{\boldsymbol{w}}_i. \tag{10}$$

*Moreover, any two-layer teacher and student networks that satisfy equation 7-equation 10, also satisfies equation 5.*

The idea of the proof consists in noticing that 0 loss means that a ReLU MLP constructed by concatenating the teacher and student hidden neurons need to be functionally 0 everywhere. By grouping together the teacher and student weight vectors that are colinear with each other, we then show that the sum of each of hidden neurons belonging to the same group need to be 0 or a linear function in $\boldsymbol{x}$. Finally, because no two teacher weight vectors are colinear, we show that necessarily a student weight needs to be colinear with each teacher weight. The permutation $\sigma$ consists in this mapping, and the partition $\mathcal{S}_1, \mathcal{S}_2$ in respectively student weights that are positively, resp. negatively colinear with the teacher.

We now formally prove the claim.

*Proof.* We collect in $\mathcal{W}$ all the row vectors $\boldsymbol{w}$ of the matrices $\boldsymbol{W}$ from the teacher and student network. By using $c_{\boldsymbol{w}}$ to denote the corresponding weight from $\boldsymbol{a}$ in the teacher network resp. minus one times the corresponding weight in the student network, we can rewrite the fact that we are at zero loss as

$$\sum_{\boldsymbol{w} \in \mathcal{W}} c_w \psi(\boldsymbol{w}^\top \boldsymbol{x}) = 0 \qquad \text{for all } \boldsymbol{x} \in \mathbb{R}^n \tag{11}$$

To now deduce some consequences from this equation we introduce some notation. For every $\boldsymbol{w} \in \mathcal{W}$ we denote by $\mathcal{C}_{\boldsymbol{w}}$ the set of all vectors in $\mathcal{W}$ that are collinear with $\boldsymbol{w}$ (including $\boldsymbol{w}$ itself) and with $\mathcal{N}_{\boldsymbol{w}} := \mathcal{W} \setminus \mathcal{C}_{\boldsymbol{w}}$ all vectors in $\mathcal{W}$ that are not collinear with $\boldsymbol{w}$. Furthermore we use $\mathcal{C}_{\boldsymbol{w}} = \mathcal{C}_{\boldsymbol{w}}^+ \cup \mathcal{C}_{\boldsymbol{w}}^-$ to denote those vectors that are positively resp. negatively collinear with $\boldsymbol{w}$.

Consider some arbitrary $\boldsymbol{w}' \in \mathcal{W}$. By definition of $\mathcal{C}_{\boldsymbol{w}'}$ and $\mathcal{N}_{\boldsymbol{w}'}$ we can find a vector $\boldsymbol{x}_0 \in \mathbb{R}^n$ such that

$$\boldsymbol{w}^\top \boldsymbol{x}_0 = 0 \quad \text{for all } \boldsymbol{w} \in \mathcal{C}_{\boldsymbol{w}'} \qquad \text{and} \qquad \boldsymbol{w}^\top \boldsymbol{x}_0 \neq 0 \quad \text{for all } \boldsymbol{w} \in \mathcal{N}_{\boldsymbol{w}'}$$

and an open ball $B$ around $\boldsymbol{x}_0$ such that we have for all $\boldsymbol{w} \in \mathcal{N}_{\boldsymbol{w}'}$: $\boldsymbol{w}^\top \boldsymbol{x} \neq 0$ for all $\boldsymbol{x} \in B$. Using $\mathbb{1}_{\mathcal{E}}$ to denote the indicator for the event $\mathcal{E}$, i.e. $\mathbb{1}_{\mathcal{E}} \equiv 1$ if $\mathcal{E}$ holds and $\mathbb{1}_{\mathcal{E}} \equiv 0$ otherwise, we thus have

$$\psi(\boldsymbol{w}^\top \boldsymbol{x}) = \mathbb{1}_{\boldsymbol{w}^\top \boldsymbol{x}_0 > 0} \cdot \boldsymbol{w}^T x \qquad \text{for all } \boldsymbol{x} \in B, \boldsymbol{w} \in \mathcal{N}_{\boldsymbol{w}'}.$$

As $\boldsymbol{w}^\top \boldsymbol{x}_0 = 0$ for $\boldsymbol{w} \in \mathcal{C}_{\boldsymbol{w}'}$ and $B$ is a ball around $\boldsymbol{x}_0$ we know that both $B^+ := \{\boldsymbol{x} \in B : {\boldsymbol{w}'}^\top \boldsymbol{x} > 0\}$ and $B^- := \{\boldsymbol{x} \in B : {\boldsymbol{w}'}^\top \boldsymbol{x} < 0\}$ are open sets in $\mathbb{R}^n$. By definition of the sets $\mathcal{C}_{\boldsymbol{w}'}^+$ and $\mathcal{C}_{\boldsymbol{w}'}^-$ we have

$$\psi(\boldsymbol{w}^\top \boldsymbol{x}) = \begin{cases} \boldsymbol{w}^\top \boldsymbol{x} & \text{if } \boldsymbol{w} \in \mathcal{C}_{\boldsymbol{w}'}^+ \\ 0 & \text{if } \boldsymbol{w} \in \mathcal{C}_{\boldsymbol{w}'}^- \end{cases} \qquad \text{for all } \boldsymbol{x} \in B^+$$

resp.

$$\psi(\boldsymbol{w}^\top \boldsymbol{x}) = \begin{cases} \boldsymbol{w}^\top \boldsymbol{x} & \text{if } \boldsymbol{w} \in \mathcal{C}_{\boldsymbol{w}'}^- \\ 0 & \text{if } \boldsymbol{w} \in \mathcal{C}_{\boldsymbol{w}'}^+ \end{cases} \qquad \text{for all } \boldsymbol{x} \in B^-$$

From equation 11 we thus get that

$$0 = (\sum_{\boldsymbol{w} \in \mathcal{N}_{\boldsymbol{w}'}} c_{\boldsymbol{w}} \mathbb{1}_{\boldsymbol{w}^\top \boldsymbol{x}_0 > 0} \cdot \boldsymbol{w} + \sum_{\boldsymbol{w} \in \mathcal{C}_{\boldsymbol{w}'}^+} c_{\boldsymbol{w}} \boldsymbol{w})^\top \boldsymbol{x} \qquad \text{for all } \boldsymbol{x} \in B^+$$

and

$$0 = (\sum_{\boldsymbol{w} \in \mathcal{N}_{\boldsymbol{w}'}} c_{\boldsymbol{w}} \mathbb{1}_{\boldsymbol{w}^\top \boldsymbol{x}_0 > 0} \cdot \boldsymbol{w} + \sum_{w \in \mathcal{C}_{\boldsymbol{w}'}^-} c_{\boldsymbol{w}} \boldsymbol{w})^\top \boldsymbol{x} \qquad \text{for all } \boldsymbol{x} \in B^-$$

As both $B^+$ and $B^-$ are open sets and we know that for any open set $O$ we have that $\boldsymbol{v}^\top \boldsymbol{x} = 0$ for all $\boldsymbol{x} \in O$ implies $\boldsymbol{v} = 0$ we thus deduce that

$$\sum_{\boldsymbol{w} \in \mathcal{C}_{\boldsymbol{w}'}^+} c_{\boldsymbol{w}} \boldsymbol{w} = \sum_{\boldsymbol{w} \in \mathcal{C}_{\boldsymbol{w}'}^-} c_{\boldsymbol{w}} \boldsymbol{w} \qquad \text{for all } \boldsymbol{w}' \in \mathcal{W}. \tag{12}$$

We now interpret this equation in terms of our teacher-student setting. Recall that we assumed that no two rows in the teacher network are collinear and that in the teacher network no weight in $\boldsymbol{a}$ is zero. That is, for any $\boldsymbol{w}' \in \mathcal{W}$ from the teacher network $\mathcal{C}_{\boldsymbol{w}'}^+ \cup \mathcal{C}_{\boldsymbol{w}'}^-$ needs to contain at least one vector from the student network. As teacher and student network have the same size, this thus implies the existence of a permutation $\sigma \in \mathfrak{S}_h$ such that for all weight vectors $\boldsymbol{w}_i \in \mathcal{W}$ from the teacher network there needs to exist a weight vector $\hat{\boldsymbol{w}}_{\sigma(i)}$ from the student network so that $\boldsymbol{w}_i$ and $\hat{\boldsymbol{w}}_{\sigma(i)}$ are collinear. Denote by $\mathcal{S}_1$ the indices $i$ such that $\hat{\boldsymbol{w}}_{\sigma(i)} \in \mathcal{C}_{\boldsymbol{w}_i}^+$ and by $\mathcal{S}_2 = [1..h] \setminus \mathcal{S}_1$ the ones such that $\hat{\boldsymbol{w}}_{\sigma(i)} \in \mathcal{C}_{\boldsymbol{w}_i}^-$. Observe that this implies that we can rewrite equation 12 as

$$a_i \boldsymbol{w}_i = a_{\sigma(i)} \boldsymbol{w}_{\sigma(i)} \quad \forall i \in \mathcal{S}_1 \qquad \text{and} \qquad a_i \boldsymbol{w}_i = -a_{\sigma(i)} \boldsymbol{w}_{\sigma(i)} \quad \forall i \in \mathcal{S}_2.$$

Here we used that we defined $c_{\boldsymbol{w}}$ for weights from the student network as minus one times the corresponding weight from $\boldsymbol{a}$. This shows equation 8 and equation 9. equation 7 also follows immediately from the definition of $\mathcal{S}_1$ and $\mathcal{S}_2$.

To prove equation 10 and the moreover-part of the theorem, we show that if equation 7 - equation 9 hold, then equation 5 and equation 10 are equivalent. As $\sigma$ is a permutation we can rewrite equation 5 as

$$\sum_i a_i \boldsymbol{w}_i \boldsymbol{x} \cdot \mathbb{1}_{\boldsymbol{w}_i \boldsymbol{x} > 0} = \sum_i \hat{a}_{\sigma(i)} \hat{\boldsymbol{w}}_{\sigma(i)} \boldsymbol{x} \cdot \mathbb{1}_{\hat{\boldsymbol{w}}_{\sigma(i)} \boldsymbol{x} > 0} \qquad \forall \boldsymbol{x} \in \mathbb{R}^m \tag{13}$$

and equation 10 as

$$\sum_i a_i \boldsymbol{w}_i \boldsymbol{x} \cdot (\mathbb{1}_{\boldsymbol{w}_i \boldsymbol{x} > 0} + \mathbb{1}_{\boldsymbol{w}_i \boldsymbol{x} < 0}) = \sum_i \hat{a}_{\sigma(i)} \hat{\boldsymbol{w}}_{\sigma(i)} \boldsymbol{x} \cdot (\mathbb{1}_{\hat{\boldsymbol{w}}_{\sigma(i)} \boldsymbol{x} > 0} + \mathbb{1}_{\hat{\boldsymbol{w}}_{\sigma(i)} \boldsymbol{x} < 0}) \qquad \forall \boldsymbol{x} \in \mathbb{R}^m. \tag{14}$$

For indices $i \in \mathcal{S}_1$ we have $\boldsymbol{w}_i \boldsymbol{x} > 0$ if and only if $\hat{\boldsymbol{w}}_{\sigma(i)} \boldsymbol{x} > 0$, thus equation 8 implies that the terms for $i$ and $\sigma(i)$ contribute identical values to the sums both in equation 13 as well as in equation 14. We can thus concentrate on the values $i \in \mathcal{S}_2$. For these we have $a_i \boldsymbol{w}_i \boldsymbol{x} > 0$ implies $\hat{a}_{\sigma(i)} \hat{\boldsymbol{w}}_{\sigma(i)} \boldsymbol{x} < 0$ and vice versa. Using this and equation 9 on both sides we see that

$$\sum_{i \in \mathcal{S}_2} a_i \boldsymbol{w}_i \boldsymbol{x} \cdot \mathbb{1}_{\boldsymbol{w}_i \boldsymbol{x} > 0} = \sum_{i \in \mathcal{S}_2} \hat{a}_{\sigma(i)} \hat{\boldsymbol{w}}_{\sigma(i)} \boldsymbol{x} \cdot \mathbb{1}_{\hat{\boldsymbol{w}}_{\sigma(i)} \boldsymbol{x} > 0}$$

is equivalent to

$$-\sum_{i \in \mathcal{S}_2} \hat{a}_{\sigma(i)} \hat{\boldsymbol{w}}_{\sigma(i)} \boldsymbol{x} \cdot \mathbb{1}_{\hat{\boldsymbol{w}}_{\sigma(i)} \boldsymbol{x} < 0} = -\sum_{i \in \mathcal{S}_2} a_i \boldsymbol{w}_i \boldsymbol{x} \cdot \mathbb{1}_{\boldsymbol{w}_i \boldsymbol{x} < 0}.$$

The equivalence of equation 13 and equation 14 follows. $\qquad \square$

If the student is larger than the teacher, ie. $\hat{h} > h$, then there is more flexibility: instead of having a mapping from teacher hidden neuron $i$ to a single student hidden neuron $\sigma(i)$, each teacher neuron $i$ now maps to a *set* of student neurons. In addition, this set partitions into positively colinear and negatively colinear subsets, resp. $\mathcal{S}_1(i), \mathcal{S}_2(i)$. Furthermore, we can have additional student neurons that do not correspond to any teacher neurons, provided they cancel out eventually. The sets $\mathcal{S}_0$ as well as $\mathcal{S}_1', \mathcal{S}_2'$ in the following theorem correspond to such neurons.

From here on, we refer to $\oplus$ as the operation that concatenates lists or vectors.

**Theorem 4.** *Under the assumptions from Theorem 4, but with $\hat{h} \geq h$, there exists a partition $\mathcal{S} = (\mathcal{S}_1(i), \mathcal{S}_2(i))_{i \in [1..h]} \oplus (\mathcal{S}'_1(i), \mathcal{S}'_2(i))_{i \in [1..h']} \oplus (\mathcal{S}_0)$ of $[1..\hat{h}]$ for some $h'$ such that we have*

$$\forall i \in [1..h], \; \forall j \in \mathcal{S}_1(i), \exists \alpha > 0 : \hat{\boldsymbol{w}}_j = \alpha \boldsymbol{w}_i, \tag{15}$$

$$\forall j \in \mathcal{S}_2(i), \exists \alpha < 0 : \hat{\boldsymbol{w}}_j = \alpha \boldsymbol{w}_i, \tag{16}$$

$$a_i \boldsymbol{w}_i = \sum_{j \in \mathcal{S}_1(i)} \hat{a}_j \hat{\boldsymbol{w}}_j - \sum_{j \in \mathcal{S}_2(i)} \hat{a}_j \hat{\boldsymbol{w}}_j, \tag{17}$$

$$\forall i \in [1..h'], \; \forall (j,k) \in \mathcal{S}'_1(i) \times \mathcal{S}'_1(i) \cup \mathcal{S}'_2(i) \times \mathcal{S}'_2(i), \exists \alpha > 0 : \hat{\boldsymbol{w}}_j = \alpha \hat{\boldsymbol{w}}_k, \tag{18}$$

$$\forall (j,k) \in \mathcal{S}'_1(i) \times \mathcal{S}'_2(i), \exists \alpha < 0 : \hat{\boldsymbol{w}}_j = \alpha \hat{\boldsymbol{w}}_k, \tag{19}$$

$$0 = \sum_{j \in \mathcal{S}'_1(i)} \hat{a}_j \hat{\boldsymbol{w}}_j - \sum_{j \in \mathcal{S}'_2(i)} \hat{a}_j \hat{\boldsymbol{w}}_j, \tag{20}$$

$$\forall j \in \mathcal{S}_0, \; 0 = \hat{a}_j \hat{\boldsymbol{w}}_j, \tag{21}$$

$$\sum_i a_i \boldsymbol{w}_i = \sum_i \hat{a}_i \hat{\boldsymbol{w}}_i. \tag{22}$$

*Moreover, any two-layer teacher and student networks that satisfy equation 15-equation 22, also satisfies equation 5.*

*Proof.* The proof is identical up to equation 12.

Firstly, we denote by $\mathcal{S}_0$ the indices of student vectors such that $\hat{a}\hat{\boldsymbol{w}} = 0$. They correspond to student neurons that have $0$ activities everywhere. This gives equation 21. Without loss of generality, we will exclude all student vectors in $\mathcal{S}_0$ from $\mathcal{W}$, and redefine $\mathcal{C}$, etc accordingly.

Secondly, recall that we assumed that no two rows in the teacher network are colinear and that in the teacher network no weight in $\boldsymbol{a}$ is zero. That is, for any $\boldsymbol{w}' \in \mathcal{W}$ from the teacher network $\mathcal{C}^+_{\boldsymbol{w}'} \cup \mathcal{C}^-_{\boldsymbol{w}'}$ needs to contain at least one vector from the student network. For all teacher vector $\boldsymbol{w}_i$ where $i \in [1..h]$ we denote by resp. $\mathcal{S}_1(i), \mathcal{S}_2(i)$ the subsets of $[1..\hat{h}]$ corresponding to indices of student vectors that belong to $\mathcal{C}^+_{\boldsymbol{w}_i}$ resp. $\mathcal{C}^-_{\boldsymbol{w}_i}$. Observe that we can then rewrite equation 12 as

$$a_i \boldsymbol{w}_i = \sum_{j \in \mathcal{S}_1(i)} \hat{a}_j \hat{\boldsymbol{w}}_j - \sum_{j \in \mathcal{S}_2(i)} \hat{a}_j \hat{\boldsymbol{w}}_j \quad \forall i \in [1..h]$$

Here we used that we defined $c_{\boldsymbol{w}}$ for weights from the student network as minus one times the corresponding weight from $\boldsymbol{a}$. This shows equation 17. equation 15 and equation 16 also follows immediately from the definition of $\mathcal{S}_1$ and $\mathcal{S}_2$.

Finally, consider the student vectors $\mathcal{W}'$ that have not been assigned to one of the sets $\mathcal{S}_1(i), \mathcal{S}_2(i)$ or $\mathcal{S}_0$. We fix some $(\hat{\boldsymbol{w}}_i)_{i \in [1..h']}$ such that $\{\mathcal{C}_{\hat{\boldsymbol{w}}_i}\}_{i \in [1..h']}$ forms a partition of this set. We then partition each $\mathcal{C}_{\hat{\boldsymbol{w}}_i}$ into $\mathcal{C}^+_{\hat{\boldsymbol{w}}_i}, \mathcal{C}^-_{\hat{\boldsymbol{w}}_i}$ and denote by $\mathcal{S}'_1(i), \mathcal{S}'_2(i)$ the set containing the indices of the student vectors belonging to resp. $\mathcal{C}^+_{\hat{\boldsymbol{w}}_i}, \mathcal{C}^-_{\hat{\boldsymbol{w}}_i}$. Then we can rewrite equation 12 as

$$\sum_{j \in \mathcal{S}'_1(i)} \hat{a}_j \hat{\boldsymbol{w}}_j = \sum_{j \in \mathcal{S}'_2(i)} \hat{a}_j \hat{\boldsymbol{w}}_j \quad \forall i \in [1..h']$$

This shows equation 20. equation 18 and equation 19 also follows immediately from the definition of $\mathcal{S}'_1$ and $\mathcal{S}'_2$.

Clearly, $\mathcal{S} = (\mathcal{S}_1(i), \mathcal{S}_2(i))_{i \in [1..h]} \oplus (\mathcal{S}'_1(i), \mathcal{S}'_2(i))_{i \in [1..h']} \oplus (\mathcal{S}_0)$ form a partition of $[1..\hat{h}]$.

To prove equation 22 and the moreover-part of the theorem, we show that if equation 15 - equation 21 hold, then equation 5 and equation 22 are equivalent. We can rewrite equation 5 as

$$\sum_{i \in [1..h]} a_i \boldsymbol{w}_i \boldsymbol{x} \cdot \mathbb{1}_{\boldsymbol{w}_i \boldsymbol{x} > 0} = \sum_{i \in [1..\hat{h}]} \hat{a}_i \hat{\boldsymbol{w}}_i \boldsymbol{x} \cdot \mathbb{1}_{\hat{\boldsymbol{w}}_i \boldsymbol{x} > 0} \qquad \forall \boldsymbol{x} \in \mathbb{R}^m \tag{23}$$

which gives

$$\sum_{i\in[1..h]} \big[ \sum_{j\in\mathcal{S}_1(i)} \hat{a}_j \hat{\boldsymbol{w}}_j - \sum_{j\in\mathcal{S}_2(i)} \hat{a}_j \hat{\boldsymbol{w}}_j \big] \boldsymbol{x} \cdot \mathbb{1}_{\boldsymbol{w}_i \boldsymbol{x}>0} \tag{24}$$

$$= \sum_{i\in[1..h]} \big[ \sum_{j\in\mathcal{S}_1(i)} \hat{a}_j \hat{\boldsymbol{w}}_j \cdot \mathbb{1}_{\hat{\boldsymbol{w}}_j \boldsymbol{x}>0} + \sum_{j\in\mathcal{S}_2(i)} \hat{a}_j \hat{\boldsymbol{w}}_j \cdot \mathbb{1}_{\hat{\boldsymbol{w}}_j \boldsymbol{x}>0} \big] \boldsymbol{x} \tag{25}$$

$$+ \sum_{i\in[1..h']} \big[ \sum_{j\in\mathcal{S}'_1(i)} \hat{a}_j \hat{\boldsymbol{w}}_j \cdot \mathbb{1}_{\hat{\boldsymbol{w}}_j \boldsymbol{x}>0} + \sum_{j\in\mathcal{S}'_2(i)} \hat{a}_j \hat{\boldsymbol{w}}_j \cdot \mathbb{1}_{\hat{\boldsymbol{w}}_j \boldsymbol{x}>0} \big] \boldsymbol{x} \qquad \forall \boldsymbol{x} \in \mathbb{R}^m. \tag{26}$$

which simpifies into

$$0 = \sum_{i\in[1..h]} \big[ \sum_{j\in\mathcal{S}_2(i)} \hat{a}_j \hat{\boldsymbol{w}}_j \cdot \mathbb{1}_{-\hat{\boldsymbol{w}}_j \boldsymbol{x}>0} + \sum_{j\in\mathcal{S}_2(i)} \hat{a}_j \hat{\boldsymbol{w}}_j \cdot \mathbb{1}_{\hat{\boldsymbol{w}}_j \boldsymbol{x}>0} \big] \boldsymbol{x} \tag{27}$$

$$+ \sum_{i\in[1..h']} \big[ \sum_{j\in\mathcal{S}'_2(i)} \hat{a}_j \hat{\boldsymbol{w}}_j \cdot \mathbb{1}_{-\hat{\boldsymbol{w}}_j \boldsymbol{x}>0} + \sum_{j\in\mathcal{S}'_2(i)} \hat{a}_j \hat{\boldsymbol{w}}_j \cdot \mathbb{1}_{\hat{\boldsymbol{w}}_j \boldsymbol{x}>0} \big] \boldsymbol{x} \qquad \forall \boldsymbol{x} \in \mathbb{R}^m. \tag{28}$$

Noticing that $\mathbb{1}_{\boldsymbol{w}\boldsymbol{x}>0} + \mathbb{1}_{-\boldsymbol{w}\boldsymbol{x}>0} = 1$ for almost any $\boldsymbol{x}$ and for any $\boldsymbol{w}\neq 0$, we get

$$0 = \sum_{i\in[1..h]} \sum_{j\in\mathcal{S}_2(i)} \hat{a}_j \hat{\boldsymbol{w}}_j \boldsymbol{x} + \sum_{i\in[1..h']} \sum_{j\in\mathcal{S}'_2(i)} \hat{a}_j \hat{\boldsymbol{w}}_j \boldsymbol{x} \tag{29}$$

for almost any $\boldsymbol{x}$, and thus

$$0 = \sum_{i\in[1..h]} \sum_{j\in\mathcal{S}_2(i)} \hat{a}_j \hat{\boldsymbol{w}}_j + \sum_{i\in[1..h']} \sum_{j\in\mathcal{S}'_2(i)} \hat{a}_j \hat{\boldsymbol{w}}_j \tag{30}$$

the equivalence of equation 5 and equation 22 follows by adding equation 17 ,equation 20 ,equation 21 over all indices and using equation 30.

$\square$

### A.1.2 IRREDUCIBILITY CONDITION FOR RELU NONLINEARITY

Here, we comment on the formal sufficient and necessary conditions for a ReLU two-layer MLP to be irreducible. As it is not relevant for the remainder of the appendix, it can be skipped.

Consider a two-layer ReLU MLP, $f(x) = \boldsymbol{a}^\top \psi(\boldsymbol{W}\boldsymbol{x}) = \sum_k a_k \psi(\boldsymbol{w}_k^\top \boldsymbol{x})$.

Firstly, if a pair of weights $\boldsymbol{w}_k, \boldsymbol{w}_l$ are positively colinear, then they can be fused into one hidden unit. Thus, a necessary condition for irreducibility is that no two weights are positively colinear.

Secondly, if some pairs of weights are negatively colinear, we can decompose the corresponding function into the sum of a new hidden neuron and a linear function, i.e. for any $(\boldsymbol{w}_k, \boldsymbol{w}_l, a_k, a_l)$ such that there exists $\alpha > 0 : \boldsymbol{w}_l = -\alpha\boldsymbol{w}_k$, for any $\boldsymbol{x}$,

$$a_k\psi(\boldsymbol{w}_k^\top \boldsymbol{x}) + a_l\psi(\boldsymbol{w}_l^\top \boldsymbol{x}) = a_k\psi(\boldsymbol{w}_k^\top \boldsymbol{x}) + a_l\psi(-\alpha\boldsymbol{w}_k^\top \boldsymbol{x}) + \alpha a_l\boldsymbol{w}_k^\top \boldsymbol{x} - \alpha a_l\boldsymbol{w}_k^\top \boldsymbol{x} \tag{31}$$

$$= a_k\psi(\boldsymbol{w}_k^\top \boldsymbol{x}) - a_l\psi(\alpha\boldsymbol{w}_k^\top \boldsymbol{x}) - \alpha a_l\boldsymbol{w}_k^\top \boldsymbol{x} \tag{32}$$

$$= (a_k - \alpha a_l)\psi(\boldsymbol{w}_k^\top \boldsymbol{x}) - \alpha a_l\boldsymbol{w}_k^\top \boldsymbol{x} \tag{33}$$

$$\tag{34}$$

where we obtain the second line by noticing that for any $\boldsymbol{w}, \boldsymbol{x}$, $\psi(\boldsymbol{w}^\top \boldsymbol{x}) - \boldsymbol{w}^\top \boldsymbol{x} = -\psi(-\boldsymbol{w}^\top \boldsymbol{x})$.

Thus, any two-layer ReLU MLP can be transformed into a the sum of a ReLU MLP with no pairs of weight being *colinear*, and a linear function. Since a linear function can be implemented by two hidden units with negatively colinear weights, any irreducible network should have at most a single pair of negatively colinear input weight.

Thirdly, the obtained linear function can further be absorbed by the ReLU hidden neurons if, by denoting $\boldsymbol{w}$ the weight of the linear function, there exists a subset of hidden neurons $\mathcal{I}$ such that

$$\boldsymbol{w} = -\sum_{i \in \mathcal{I}} a_i \boldsymbol{w}_i \tag{35}$$

since in that case, we have, for any $\boldsymbol{x}$,

$$\sum_{i \in \mathcal{I}} a_i \psi(\boldsymbol{w}_i^\top \boldsymbol{x}) + \boldsymbol{w}^\top \boldsymbol{x} = \sum_{i \in \mathcal{I}} a_i \psi(\boldsymbol{w}_i^\top \boldsymbol{x}) - \sum_{i \in \mathcal{I}} a_i \boldsymbol{w}_i^\top \boldsymbol{x} \tag{36}$$

$$= \sum_{i \in \mathcal{I}} -a_i \psi(-\boldsymbol{w}_i^\top \boldsymbol{x}) \tag{37}$$

Thus, for $f$ to be irreducible, the following necessary conditions need to be fulfilled:

- There exists no output weight $a_k$ that is 0

- There exists no pair of input weights that are positively colinear

- There exists not more than one pair of input weights that are negatively colinear

- If a pair of negatively colinear input weight exists, by denoting $k, l$ the corresponding hidden neuron indices, there are no subset $\mathcal{I}$ of $[1..h] \setminus \{k, l\}$ such that $a_k \boldsymbol{w}_k = -\sum_{i \in \mathcal{I}} a_i \boldsymbol{w}_i$ or $a_l \boldsymbol{w}_l = -\sum_{i \in \mathcal{I}} a_i \boldsymbol{w}_i$

The above conditions constitute in fact sufficient conditions for irreducibility of a two-layer ReLU MLP. The proof follows a similar structure as that of Theorem 4, and notice that necessarily $\hat{h} \geq h$.

## A.2 MULTI-TASK PARAMETER IDENTIFICATION

We now come back to the multi-task setting, where the teacher generates a task-specific network using a hypernetwork conditioned on a task latent variable. Specifically, the weights of the second layer are fixed across the tasks, but the weights of the first layer are obtained as a linear combination of the modules: $\boldsymbol{W}(\Theta, \boldsymbol{z}) = \sum_m^M z^{(m)} \Theta^{(m)}$. See section 3.1 for the full description.

Given a task latent variable $\boldsymbol{z}$, clearly the theoretical results from the last section can be applied to the generated teacher and student network in order to establish a relation between the teacher and student parameters. We will first extend the theoretical result establishing the sufficient and necessary conditions on the student parameters for the student to be able to fit the teacher on any task.

We provide results for MLPs with ReLU activation function as well as MLPs where the activation function belongs to a general class of smooth functions, for student networks that can be wider than the teacher network. Theorem 2 in the main text is a special case of the following results.

### A.2.1 RELU MLP

**Theorem 5.** *Assume $\mathcal{P}_x$ has full support in the input space, $M \leq n$, $\hat{M} = M$, $\hat{h} \geq h$, and that there exists a full rank matrix in the linear span of $(\Theta_i)_{1 \leq i \leq h}$, and $(a_i)_{1 \leq i \leq h}$ are all non-zero. Assume furthermore that there exists at least one $\boldsymbol{z}$ such that no two rows of $W(\Theta, \boldsymbol{z})$ are colinear. Finally, assume $\psi$ is the ReLU nonlinearity. Then, $\min_{\hat{\boldsymbol{z}}} L(\hat{\boldsymbol{z}}, (\hat{\Theta}, \hat{\boldsymbol{a}}); \mathcal{D}_{\boldsymbol{z}}) = 0$ for almost any $\boldsymbol{z}$ is equivalent to the existence of an invertible matrix $\boldsymbol{F}$, a partition $\mathcal{S} = (\mathcal{S}_1(i), \mathcal{S}_2(i))_{i \in [1..h]} \oplus$*

$(\mathcal{S}'_1(i), \mathcal{S}'_2(i))_{i \in [1..h']} \oplus (\mathcal{S}_0)$ *of* $[1..\hat{h}]$ *such that we have*

$$\sum_i a_i \Theta_i = \sum_i \hat{a}_i \hat{\Theta}_i \boldsymbol{F} \tag{38}$$

$$\forall i \in [1..h], \ \forall j \in \mathcal{S}_1(i), \exists \alpha > 0 : \hat{\Theta}_j \boldsymbol{F} = \alpha \Theta_i, \tag{39}$$

$$\forall j \in \mathcal{S}_2(i), \exists \alpha < 0 : \hat{\Theta}_j \boldsymbol{F} = \alpha \Theta_i, \tag{40}$$

$$a_i \Theta_i = \sum_{j \in \mathcal{S}_1(i)} \hat{a}_j \hat{\Theta}_j \boldsymbol{F} - \sum_{j \in \mathcal{S}_2(i)} \hat{a}_j \hat{\Theta}_j \boldsymbol{F}, \tag{41}$$

$$\forall i \in [1..h'], \ \forall (j,k) \in \mathcal{S}'_1(i) \times \mathcal{S}'_1(i) \cup \mathcal{S}'_2(i) \times \mathcal{S}'_2(i), \exists \alpha > 0 : \hat{\Theta}_j = \alpha \hat{\Theta}_k, \tag{42}$$

$$\forall (j,k) \in \mathcal{S}'_1(i) \times \mathcal{S}'_2(i), \exists \alpha < 0 : \hat{\Theta}_j = \alpha \hat{\Theta}_k, \tag{43}$$

$$0 = \sum_{j \in \mathcal{S}_1(i)} \hat{a}_j \hat{\Theta}_j - \sum_{j \in \mathcal{S}_2(i)} \hat{a}_j \hat{\Theta}_j, \tag{44}$$

$$\forall j \in \mathcal{S}_0, \ 0 = \hat{a}_j \hat{\Theta}_j \tag{45}$$

Before proving the theorem, we provide a Lemma which will be useful.

**Lemma 6.** *Assume there exists one point $\boldsymbol{z}_0$ such that $W(\Theta, \boldsymbol{z}_0)$ has no two rows that are colinear. Then, for any linear subspace $U$ containing $\boldsymbol{z}_0$, $W(\Theta, \boldsymbol{z})$ has no two rows that are colinear for $\boldsymbol{z} \in U$ almost everywhere[2]. In particular, the set of points $\boldsymbol{z}$ in $U$ such that $W(\Theta, \boldsymbol{z})$ has at least two colinear rows form a finite union of strict linear subspace of $U$.*

*Proof.* Recall that the $i$th row of $W(\Theta, z_0)$ is given by $\Theta_i z_0$. Consider a pair $i, j$ with $i \neq j$. As no two rows in $W(\Theta, z_0)$ are colinear, there exists only a finite number of $\alpha \in \mathbb{R}$ such that the equation $(\Theta_i - \alpha \Theta_j) \boldsymbol{z} = 0$ yields solutions of $\boldsymbol{z}$ that are not in $\text{Ker } \Theta_i \cap \text{Ker } \Theta_j$. Given each of such $\alpha$, the set of $\boldsymbol{z}$ that verifies the equation form a linear subspace. Taking everything together, the set of $\boldsymbol{z}$ such that $W(\Theta, \boldsymbol{z})$ has at least two colinear elements is a finite union of linear subspaces. Due to the existence of $\boldsymbol{z}_0$ we know that these linear subspaces must not include any linear subspace containing $\boldsymbol{z}_0$ and the claim follows. $\qquad\square$

We now prove the Theorem.

*Proof.* By assumption, there exists at least one point $\boldsymbol{z}_0$ such that $W(\Theta, \boldsymbol{z}_0)$ has no two rows that are colinear. Following Lemma 6, for almost every $\boldsymbol{z}$ in $\mathbb{R}^M$, the first layer weights have no two rows that are colinear. For all $\boldsymbol{z}$, we fix $\hat{\boldsymbol{z}} \in \arg\min_{\hat{\boldsymbol{z}}} L(\hat{\boldsymbol{z}}, (\hat{\Theta}, \hat{\boldsymbol{a}}); \mathcal{D}_{\boldsymbol{z}})$. By assumption, $L(\hat{\boldsymbol{z}}, \hat{\Theta}; \mathcal{D}_{\boldsymbol{z}}) = 0$ for almost every $\boldsymbol{z}$. Considering the intersection of such $\boldsymbol{z}$ and those that generate first layer weights with no two colinear rows, we can apply Theorem 4 for almost every $\boldsymbol{z}$. For each such $\boldsymbol{z}$ the theorem guarantees us a partition $\mathcal{S} = (\mathcal{S}_1(i), \mathcal{S}_2(i))_{i \in [1..h]} \oplus (\mathcal{S}'_1(i), \mathcal{S}'_2(i))_{i \in [1..h']} \oplus (\mathcal{S}_0)$ such that equation 15-equation 22 hold for the generated weights.

A priori these partitions need not be the same. We prove this next. Denote for a fixed $\mathcal{S}$, $U^{\mathcal{S}}$ the set of all points in $\mathbb{R}^M$ for which Theorem 4 can be applied, and holds for this $\mathcal{S}$. As $\bigcup_{\mathcal{S}} U_k^{\mathcal{S}}$ is dense in $\mathbb{R}^M$ and the set of possible partitions $\mathcal{S}$ is finite, there thus has to exist at least one $\mathcal{S}$ so that $\mathbb{R}^M \subseteq \text{span}(U^{\mathcal{S}})$. This means in particular that we can write any point in $\mathbb{R}^M$ as a linear combination of points in $U^{\mathcal{S}}$. Thus for any point in $\mathbb{R}^M$, there exists a $\hat{\boldsymbol{z}}$ such that equation 15-equation 22 holds for $\mathcal{S}$.

It remains to show the existence of an invertible matrix $\boldsymbol{F}$, as well as the constancy of the various parameters $\alpha$ w.r.t. $\boldsymbol{z}$. Using equation 17 in particular, we get that

$$\forall \boldsymbol{z} \in \mathbb{R}^M, \exists \hat{\boldsymbol{z}} : \forall i \in [1..h], a_i \Theta_i z = \left[ \sum_{j \in \mathcal{S}_1(i)} \hat{a}_j \hat{\Theta}_j - \sum_{j \in \mathcal{S}_2(i)} \hat{a}_j \hat{\Theta}_j \right] \hat{\boldsymbol{z}} \tag{46}$$

Because there exists a full rank matrix in the linear span of the $(\Theta_i)_{1 \leq i \leq h}$, then clearly we can find some scalar coefficients $(\lambda_i)_{i \in [1..h]}$ and linearly combine the equations 46 over $i$ such that we

---

[2]By "$\boldsymbol{z} \in U$ almost everywhere" we imply using the Lebesgue measure

obtain an equation of the form $\boldsymbol{A}\boldsymbol{z} = \boldsymbol{B}\hat{\boldsymbol{z}}$ for all $\boldsymbol{z} \in \mathbb{R}^M$ and corresponding $\hat{\boldsymbol{z}}$, where $\boldsymbol{A}$ is full rank. Necessarily then, $\boldsymbol{B}$ is full rank, and multiplying by its pseudoinverse yields $\boldsymbol{F}\boldsymbol{z} = \hat{\boldsymbol{z}}$ where $\boldsymbol{F}$ is an invertible matrix. $\hat{\boldsymbol{z}}$ for which the equations hold is thus a linear function of $\boldsymbol{z}$, which also implies the constancy of the various parameters $\alpha$ w.r.t. $\boldsymbol{z}$.

The proof is concluded by applying the fact that $\boldsymbol{P}\boldsymbol{z} = \boldsymbol{Q}\boldsymbol{z}$ for all $\boldsymbol{z} \in \mathbb{R}^M$ implies $\boldsymbol{P} = \boldsymbol{Q}$ on all equation 15-equation 22.

$\square$

In particular, if we restrict the student width to be equal to that of the teacher, we get the following corollary.

**Corollary 6.1.** *Assume $\mathcal{P}_x$ has full support in the input space, $M \leq n$, $\hat{M} = M$, $\hat{h} = h$, and that there exists a full rank matrix in the linear span of $(\Theta_i)_{1 \leq i \leq h}$, and $(a_i)_{1 \leq i \leq h}$ are all non-zero. Assume furthermore that there exists at least one $\boldsymbol{z}$ such that no two rows of $\bar{W}(\Theta, \boldsymbol{z})$ are colinear. Finally, assume $\psi$ is the ReLU nonlinearity. Then, $\min_{\hat{\boldsymbol{z}}} L(\hat{\boldsymbol{z}}, (\hat{\Theta}, \hat{\boldsymbol{a}}); \mathcal{D}_{\boldsymbol{z}}) = 0$ for almost any $\boldsymbol{z}$ is equivalent to the existence of an invertible matrix $\boldsymbol{F}$, a partition $\mathcal{S} = (\mathcal{S}_1, \mathcal{S}_2)$ and a permutation $\sigma$ of $[1..\hat{h}]$ such that we have*

$$\sum_i a_i \Theta_i = \sum_i \hat{a}_i \hat{\Theta}_i \boldsymbol{F} \tag{47}$$

$$\forall i, \hat{a}_{\sigma(i)} a_i > 0 \tag{48}$$

$$\forall i \in \mathcal{S}_1, \hat{a}_{\sigma(i)} \hat{\Theta}_{\sigma(i)} \boldsymbol{F} = a_i \Theta_i, \tag{49}$$

$$\forall i \in \mathcal{S}_2, \hat{a}_{\sigma(i)} \hat{\Theta}_{\sigma(i)} \boldsymbol{F} = -a_i \Theta_i \tag{50}$$

### A.2.2 MLP WITH INFINITELY DIFFERENTIABLE ACTIVATION FUNCTION

We first provide a reformulated version of the main Theorem 4.2. from Simsek et al. (2021), which is the counterpart to our Theorem 4.

**Theorem 7.** *Assume a teacher and student two-layer network parameterized by resp. $(\boldsymbol{W}, \boldsymbol{a})$ and $(\hat{\boldsymbol{W}}, \hat{\boldsymbol{a}})$, of hidden dimension resp. $h, \hat{h}$ with $h \leq \hat{h}$ and single output unit such that*

$$\mathbb{E}_{\boldsymbol{x} \sim \mathcal{P}_x}[L(\boldsymbol{x})] = 0 \tag{51}$$

*where*

$$L(\boldsymbol{x}) = \frac{1}{2}\|\boldsymbol{a}^\top \psi(\boldsymbol{W}\boldsymbol{x}) - \hat{\boldsymbol{a}}^\top \psi(\hat{\boldsymbol{W}}\boldsymbol{x})\|^2 \tag{52}$$

*Assuming*

- *$\mathcal{P}_x$ has full support in the input space*

- *$(\boldsymbol{W}, \boldsymbol{a})$ is irreducible, i.e.*

    - *no output weight $a_k$ is $0$*
    - *no input weight $\boldsymbol{w}_k$ is $0$*
    - *no two input weights $\boldsymbol{w}_k, \boldsymbol{w}_l$ are identical*

- *$\psi \in \mathcal{C}^\infty$, and $\psi^{(n)}(0) \neq 0$ for infinitely many even and odd values of $n$*

*Then there exists a partition $(\mathcal{S}_i)_{1 \leq i \leq h} \oplus (\mathcal{S}'_i)_{1 \leq i \leq h'}$ of $[1..h]$ for some $h'$ such that we have $\forall i \in [1..h]$,*

$$\forall j \in \mathcal{S}_i, \hat{\boldsymbol{w}}_j = \boldsymbol{w}_i \tag{53}$$

$$\sum_{j \in \mathcal{S}_i} \hat{a}_j = a_i \tag{54}$$

*and* $\forall i \in [1..h']$,

$$\forall (k,l) \in \mathcal{S}_i' \times \mathcal{S}_i', \hat{\boldsymbol{w}}_k = \hat{\boldsymbol{w}}_l \tag{55}$$

$$\sum_{j \in \mathcal{S}_i'} \hat{a}_j = 0 \tag{56}$$

In essence, if the teacher parameters are irreducible, we learn the same parameters up to permutation when the hidden layer is of the same width. If we allow the student to have a wider width, the overparameterization can either:

- Allow a hidden teacher neuron to subdivide into multiple student copies whose contribution on the last layer sums up to that of a teacher neuron.

- Allow for null neurons to appear, i.e. neurons who have no contribution on the final layer.

We now provide the identification statement in multi-task teacher-student setting.

**Theorem 8.** *Assume $\mathcal{P}_x$ has full support in the input space, $M \leq n$, $\hat{M} = M$, $\hat{h} \geq h$, and that there exists a full rank matrix in the linear span of $(\Theta_i)_{1 \leq i \leq h}$, and $(a_i)_{1 \leq i \leq h}$ are all non-zero. Assume furthermore that there exists at least one $\boldsymbol{z}$ such that no two rows of $\overline{W}(\Theta, \boldsymbol{z})$ are identical. Finally, assume the activation function $\psi$ satisfies $\psi \in \mathcal{C}^\infty$, and $\psi^{(n)}(0) \neq 0$ for infinitely many even and odd values of $n$. Then, $\min_{\hat{\boldsymbol{z}}} L(\hat{\boldsymbol{z}}, (\hat{\Theta}, \hat{\boldsymbol{a}}); \mathcal{D}_{\boldsymbol{z}}) = 0$ for all $\boldsymbol{z}$ is equivalent to the existence of an invertible matrix $\boldsymbol{F}$ and a partition $(\mathcal{S}_i)_{1 \leq i \leq h} \oplus (\mathcal{S}_i')_{1 \leq i \leq h'}$ for some $h'$ of $[1,...\hat{h}]$ such that*

$$\forall i \in [1,..h], \forall j \in \mathcal{S}_i, \; \hat{\Theta}_j \boldsymbol{F} = \Theta_i, \tag{57}$$

$$\sum_{j \in \mathcal{S}_i} \hat{a}_j = a_i \tag{58}$$

$$\forall i \in [1,..h'], \forall (j,k) \in \mathcal{S}_i' \times \mathcal{S}_i', \; \hat{\Theta}^{(k)} = \hat{\Theta}^{(j)}, \tag{59}$$

$$\sum_{j \in \mathcal{S}_i'} \hat{a}_j = 0 \tag{60}$$

The proof follows the same steps as Theorem 5 except for the use of Theorem 7 instead of 4 to prove the desired equations.

## A.3   EXAMPLE FAILURE CASES TO BUILD INTUITION

We wish to uncover in the following the conditions under which a student might achieve compositional generalization when trained on a training set of tasks. This subsection provides some failure cases where perfectly learning the training tasks does not transfer to unseen tasks. It does not provide proofs, but rather an intuition on the later stated theorems and the assumptions they require.

In what follows, $(e_m)$ denotes the canonical basis of $\mathbb{R}^M$.

**Wider student**   Let us consider a teacher with one hidden neuron and three modules: $\Theta_1 = (A|B|C)$, for three vectors $A, B, C$ that are independent. The output weight of the unique hidden neuron is $a_1 = \lambda$. We allow the student to have more than one hidden neuron. We consider a training task distribution defined by the binary masks $\{e_1 + e_2, e_2 + e_3\}$. The task distribution is then trivially compositional and connected. Yet, we will show in the following that when $\hat{h} = 3 = h + 2$, a student can be constructed which perfectly fits the teacher on all training tasks, while being unable to generalize compositionally. Let a student network with the following parameter specifications:

|              | $m = 1$ | $m = 2$ | $m = 3$ |
|--------------|---------|---------|---------|
| $\hat{\Theta}_1$ | $A$     | $B$     | $0$     |
| $\hat{\Theta}_2$ | $0$     | $B$     | $C$     |
| $\hat{\Theta}_3$ | $0$     | $B$     | $0$     |

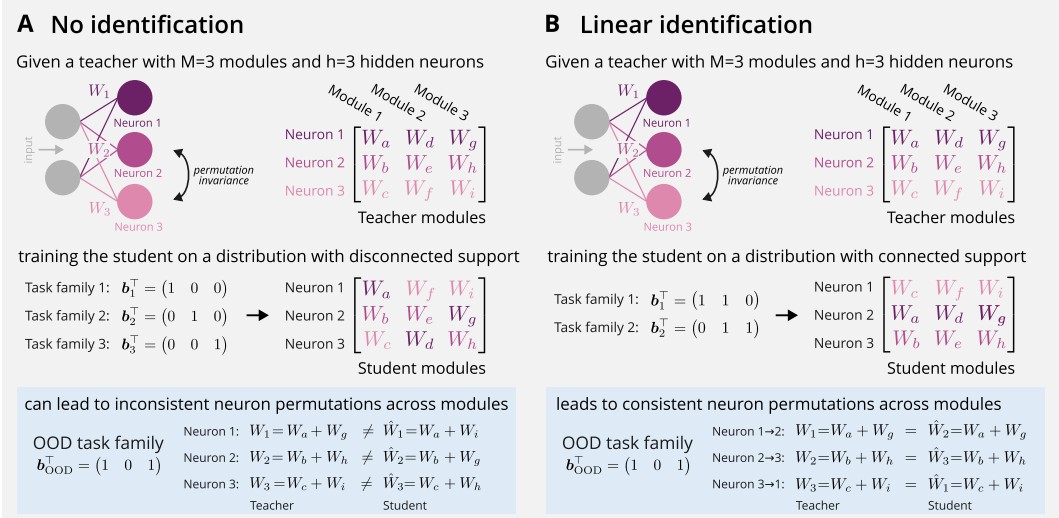

Figure A1: **A No identification.** Toy example of how correct identification of individual modules can lead to inconsistent neuron permutations preventing compositional generalization to unseen module compositions: Consider the simplified setting of a teacher and student network with two input neurons and three hidden neurons. Both the teacher and the student have $M = 3$ modules. The upper right defines the teacher weights for each module. For instance the weights denoted by $W_b$ correspond to the weights connecting neuron 2 to the input for module 1. We now assume the student during training encounters three tasks. For each task exactly one of the teacher modules is used. Since in MLPs the ordering of neurons is permutation invariant, the student can perfectly match the teacher weights, even when it uses a different ordering of the neurons. As a result, the student modules can perfectly fit all three tasks, despite permuting the neuron indices. For instance, neuron 2 in module 3 of the student contains the weights $W_g$ whereas the corresponding neuron in the teacher contains the weights $W_h$. When we now present a new task during out-of-distribution evaluation that mixes two of the modules in the teacher, the student is required to mix the weights of each neuron across these two modules as well. Since the neuron permutations in the student are inconsistent with those of the teacher, the student is unable to create the correct neuron weights and ends up unable to generalize to the OOD task. **B Linear identification.** Toy example of how having connected support helps ensure that neuron permutations across modules are consistent allowing for compositional generalization to unseen module compositions: The teacher is setup identically to **A**. Different to before, the training distribution now has *connected support*, i.e. the binary masks defining the task families share a non-zero entry. After learning the neurons of the student are still permuted compared to the neurons of the teacher but this time the permutation is consistent across modules, i.e. compared to the teacher only rows are permuted. As a result, when presenting a novel task from a task family that mixes modules, the student is able to match each of the teacher neurons and therefore compositionally generalizes. While this example only shows a permutation of the learned student neurons consistent across modules, in general the student modules will be a *linear transformation* of the teacher modules, hence the naming *linear identification*. Importantly this linear transformation is consistent across modules given the conditions of Theorem 2 are satisfied.

|  | output weight |
|---|---|
| $\hat{\boldsymbol{w}}^{(1)}$ | $\lambda$ |
| $\hat{\boldsymbol{w}}^{(2)}$ | $\lambda$ |
| $\hat{\boldsymbol{w}}^{(3)}$ | $-\lambda$ |

One can verify that this behaves as expected on the training tasks. However, if we now take task $e_1 + e_2 + e_3$, one can also verify that it does not behave as the teacher. The same argument can be made to break the compositional generalization of any teacher network width given this training task distribution.

**Disconnected tasks**  We can consider a student with two neurons and two modules. Let $\Theta_1 = (A|B)$ and $\Theta_2 = (C|D)$ for four linearly independent vectors $A, B, C, D$, and output weights all equal $a_1 = a_2 = \lambda$. If we only consider tasks defined by masks $\{e_1, e_2\}$, a valid student is given by $\hat{\Theta}_1 = (A|D)$ and $\hat{\Theta}_2 = (C|B)$ and the same output weights. This student does however not generalize to the task described by mask $e_1 + e_2$. Also see Figure A1 for a more detailed example of this failure case.

**Non-compositional support**  This is a trivial failure case: if there is a teacher module which does not appear in the training task distribution, then it is hopeless for the student to generalize to new tasks involving this unseen module.

### A.4  COMPOSITIONAL GENERALIZATION

We will now provide the formal theorem for the ReLU nonlinearity as the activation function. Theorem 1 in the main text is an informal presentation of this theorem. To show compositional generalization, similarly to Theorem 5, we will show that the $\boldsymbol{z}$-dependent partition $\mathcal{S}_1, \mathcal{S}_2$ and permutation $\sigma$ which we obtain by applying Theorem 4 is in fact common to all task latent variables in the support. Coupled with the assumption that $\mathcal{P}_z$ has compositional support it then suffices to show that the sufficient conditions of Corollary 6.1 hold.

Assuming the support of the task distribution contains task families $Z_k$ as described in the main text, we will proceed in two steps:

1. Given a $k$, we will show that if the student achieves zero loss on all tasks in $Z_k$, then the partition $\mathcal{S}_1, \mathcal{S}_2$ and permutation $\sigma$ for all $\boldsymbol{z} \in Z_k$ is identical.

2. Using the property of connected support, we will "glue" these partitions and permutations together to show that they are in fact the same for all $k$.

Showing the first step requires $\mathcal{P}_z$ to verify a number of properties. First of all, we clearly need that $\mathcal{P}_z$ is such that the probability of choosing a value from $Z_k$ is non-zero for all $k$. However, we additionally need some properties how the probability mass is distributed on each $Z_k$. The key idea of our proof is to show that generalization within $Z_k$ actually occurs as soon as we have zero loss on an appropriate basis of $Z_k$. For this to work an obvious necessary condition is that a (strict) linear subspace of $\mathrm{span}(Z_k)$ should not contain probability mass larger than zero. Note that this implies that the probability that any $d_k$ i.i.d. samples of $Z_k$, where $d_k = \dim(\mathrm{span}(Z_k))$ denotes the dimension of $Z_k$, will *not* form a basis of the space spanned by $Z_k$ is zero. In fact, we need a slight generalization of this idea, as we also need that certain other non i.i.d. sampling of $d_k$ points have zero probability. For this to hold we require that the induced probability distribution obtained by projecting the points in $Z_k$ on the unit sphere admits a density function.

Formally, we thus want the training task distribution $\mathcal{P}_z$ to have the property that we can extract a family $Z_k$ from the support $\mathcal{Z}$ of $\mathcal{P}_z$, with an associated family of binary vectors $\{\boldsymbol{b}_1, \dots, \boldsymbol{b}_K\}$ and spans $\mathrm{span}(Z_k) = \{\boldsymbol{v} \odot \boldsymbol{b}_k \mid \boldsymbol{v} \in \mathbb{R}^M\}$, such that the following three conditions are satisfied:

- $\mathbb{P}_{\boldsymbol{z} \sim \mathcal{P}_z}[\boldsymbol{z} = 0] = 0$
- For any $k$, $\mathbb{P}_{\boldsymbol{z} \sim \mathcal{P}_z}[\boldsymbol{z} \in Z_k] > 0$
- $\mathcal{P}_{\boldsymbol{z} \in Z_k}$ projected on the unit sphere admits a probability density function on the unit sphere

where we denote by $\mathcal{P}_{z \in Z_k}$ the probability distribution $\mathcal{P}_z$ conditioned on drawing a value from $Z_k$.

The above conditions would hold, for example, for any distributions $\mathcal{P}_z$ which is a discrete mixture of distributions $\mathcal{P}_{z \in Z_k}$, such that each $\mathcal{P}_{z \in Z_k}$ is a uniform distribution defined on the set of convex combinations of the canonical vectors corresponding to the non-zero entries of $b_k$, as we do in our experiments.

We are now ready to state and prove the theorem. The proof for the general class of smooth activation function used in Appendix A.2 follows the same steps, we omit the details.

**Theorem 9.** *Assume $\mathcal{P}_x$ has full support in the input space, $M \leq n$, $\hat{M} = M$, $\hat{h} = h$, that $\sum_i a_i \Theta_i$ is full rank, and $(a_i)_{1 \leq i \leq h}$ are all non-zero. Assume furthermore that for all modules $\Theta^{(m)}$, no two rows are colinear, that we can extract a family $Z_k$ from the support $\mathcal{Z}$ of $\mathcal{P}_z$ with the above property, and that $\mathcal{P}_z$ has connected and compositional support. Then,*

$$\mathbb{E}_{z \sim \mathcal{P}_z} \left[ \min_{\hat{z}} \ L(\hat{z}, (\hat{\Theta}, \hat{a}); \mathcal{D}_z) \right] = 0 \implies \forall z, \min_{\hat{z}} \ L(\hat{z}, (\hat{\Theta}, \hat{a}); \mathcal{D}_z) = 0 \qquad (61)$$

*Furthermore, for any $k$, we denote $d_k = \dim(\mathrm{span}(Z_k))$. Then for any set of task latent variables $\mathcal{F} = \bigcup_k \mathcal{F}_k$ such that $\mathcal{F}_k = \{z_{k,1}, .. z_{k,d_k+1}\}$ are sampled i.i.d. from $\mathcal{P}_{z \in Z_k}$, we have*

$$\forall z \in \mathcal{F}, \min_{\hat{z}} \ L(\hat{z}, (\hat{\Theta}, \hat{a}); \mathcal{D}_z) = 0 \implies \forall z, \min_{\hat{z}} \ L(\hat{z}, (\hat{\Theta}, \hat{a}); \mathcal{D}_z) = 0 \qquad (62)$$

In the following, we will prove the result in equation 61. The result in equation 62 requires a more involved intermediate result which we provide at the end of the section.

*Proof of Theorem 9, first part.* In a first step we prove the existence of an invertible matrix $\boldsymbol{F}$, and for each $k$, the existence of permutations $\sigma_k$ of $[1..h]$, and partitions $\mathcal{S}_1^k, \mathcal{S}_2^k$ of $[1..h]$ such that $\forall z \in \mathrm{span}(Z_k)$,

$$\begin{aligned} \forall i \in \mathcal{S}_1^k, \quad \hat{a}_i \hat{\Theta}_i \boldsymbol{F} z &= a_{\sigma_k(i)} \Theta_{\sigma_k(i)} z \\ \forall i \in \mathcal{S}_2^k, \quad \hat{a}_i \hat{\Theta}_i \boldsymbol{F} z &= -a_{\sigma_k(i)} \Theta_{\sigma_k(i)} z. \end{aligned} \qquad (63)$$

We start by arguing that we may assume without loss of generality that all $z \in Z_k$ satisfy some nice properties. First we claim that we may assume that all points $z \in Z_k$ satisfy $\min_{\hat{z}} \ L(\hat{z}, (\hat{\Theta}, \hat{a}); \mathcal{D}_z) = 0$. This holds as $\mathbb{E}_{z \sim \mathcal{P}_z} \left[ \min_{\hat{z}} \ L(\hat{z}, (\hat{\Theta}, \hat{a}); \mathcal{D}_z) \right] = 0$ and our assumption that $\mathbb{P}_{z \sim \mathcal{P}_z}[z \in Z_k] > 0$ implies that the set of points $z$ of $Z_k$ such that $\min_{\hat{z}} \ L(\hat{z}, (\hat{\Theta}, \hat{a}); \mathcal{D}_z) \neq 0$ has zero probability.

Next we argue that we may assume that for all $z \in Z_k$ we have that $W(\Theta, z)$ contains no two rows that are colinear. To see why this is true recall that we assumed that all modules $\Theta^{(m)}$ contain no two rows that are colinear. In other words, all canonical basis vectors corresponding to the non-zero entries of $b_k$ have this property and the claim thus follows from Lemma 6 together with our assumption that $\mathcal{P}_{z \in Z_k}$ projected on the unit sphere admits a probability density function on the unit sphere.

Taken together, this allows us to apply Theorem 4 on all elements of $Z_k$. That is, for all $z$ in $Z_k$, there exists $\sigma_z, \mathcal{S}_1(z), \mathcal{S}_2(z)$ such that we have

$$\forall i \in \mathcal{S}_1(z), \quad \hat{a}_i \hat{\Theta}_i \hat{z} = a_{\sigma_z(i)} \Theta_{\sigma_z(i)} z \qquad (64)$$

$$\forall i \in \mathcal{S}_2(z), \quad \hat{a}_i \hat{\Theta}_i \hat{z} = -a_{\sigma_z(i)} \Theta_{\sigma_z(i)} z \qquad (65)$$

$$\sum_i \hat{a}_i \hat{\Theta}_i \hat{z} = \sum_i a_i \Theta_i z \qquad (66)$$

where here, and henceforth, $\hat{z} = \arg\min_{\phi_z} \ L(\phi_z, \hat{\Theta}; \mathcal{D}_z)$.

A priori these $\sigma_z, S(z) = (\mathcal{S}_1(z), \mathcal{S}_2(z))$ need not be the same. We prove this next.

We group the set of all points in $Z_k$ into sets $U^{(\sigma,S)}$ so that the points in $U^{(\sigma,S)}$ all obtain $(\sigma, S)$ when applying Theorem 4. Clearly, $\bigcup_{(\sigma,S)} U^{(\sigma,S)} = Z_k$. Furthermore, because $\mathcal{P}_{\boldsymbol{z} \in Z_k}$ when projected onto the unit sphere admits a density function, it follows that $Z_k$ contains an infinite number of points of which any subset of $d_k$ elements form a basis of $\mathrm{span}(Z_k)$. Since the set of possible permutations and partitions $(\sigma, S)$ is finite, there has to exist at least one pair $(\sigma, S)$ so that $U^{(\sigma,S)}$ contains a basis of $\mathrm{span}(Z_k)$. By linearity, this implies in turn that Theorem 4 holds for this $(\sigma, S)$ for every point in $\mathrm{span}(Z_k)$.

We can thus fix $\sigma_k, \mathcal{S}_1^k, \mathcal{S}_2^k$ such that for all $\boldsymbol{z} \in \mathrm{span}(Z_k)$:

$$\forall i \in \mathcal{S}_1^k, \quad \hat{a}_i \hat{\Theta}_i \hat{\boldsymbol{z}} = a_{\sigma_k(i)} \Theta_{\sigma_k(i)} \boldsymbol{z} \tag{67}$$

$$\forall i \in \mathcal{S}_2^k, \quad \hat{a}_i \hat{\Theta}_i \hat{\boldsymbol{z}} = -a_{\sigma_k(i)} \Theta_{\sigma_k(i)} \boldsymbol{z} \tag{68}$$

$$\sum_i \hat{a}_i \hat{\Theta}_i \hat{\boldsymbol{z}} = \sum_i a_i \Theta_i \boldsymbol{z}. \tag{69}$$

It remains to show the existence of the desired matrix $\boldsymbol{F}$. Towards this end, recall that we assumed that $\sum_i a_i \Theta_i$ is full rank. As we also assumed that $\mathrm{span}(\bigcup_k Z_k) = \mathbb{R}^M$, it follows from equation 69 that $\sum_i \hat{a}_i \hat{\Theta}_i$ also has full rank. Thus, letting $\boldsymbol{F} = (\sum_i \hat{a}_i \hat{\Theta}_i)^\dagger \sum_i a_i \Theta_i$, where $A^\dagger$ denotes the pseudoinverse of $A$, we obtain a matrix that is full rank and satisfies $\hat{\boldsymbol{z}} = \boldsymbol{F} \boldsymbol{z}$ for all $\boldsymbol{z}$ in $\mathrm{span}(Z_k)$, for all $k$. Taking everything together, the equality in equation 63 is obtained.

We now show that the $\mathcal{S}_1^k, \mathcal{S}_2^k, \sigma_k$ are the same for all $k$. By our assumption on the connectedness of the sets $Z_k$, it suffices to show the following implication: if $k \neq l$ are such that there exists a $\boldsymbol{z}$ in $\mathrm{span}(Z_k) \cap \mathrm{span}(Z_l)$ so that $\boldsymbol{W}(\Theta, \boldsymbol{z})$ has no two rows that are colinear, then $\sigma_k = \sigma_l$, and $\mathcal{S}_1^k = \mathcal{S}_1^l, \mathcal{S}_2^k = \mathcal{S}_2^l$.

To see this, fix some $\boldsymbol{z}$ in $\mathrm{span}(Z_k) \cap \mathrm{span}(Z_l)$ so that $\boldsymbol{W}(\Theta, \boldsymbol{z})$ has no two rows that are colinear. From equation 63, we obtain the equality

$$\forall i \in [1..h], a_{\sigma_k(i)} \Theta_{\sigma_k(i)} \boldsymbol{z} = (-1)^{\mathbb{1}_{i \in \mathcal{S}_2^k}} \hat{a}_i \hat{\Theta}_i \boldsymbol{F} \boldsymbol{z} \tag{70}$$

$$= (-1)^{\mathbb{1}_{i \in \mathcal{S}_2^k}} (-1)^{\mathbb{1}_{i \in \mathcal{S}_2^l}} a_{\sigma_l(i)} \Theta_{\sigma_l(i)} \boldsymbol{z} \tag{71}$$

i.e. we get that for any $i$, $\Theta_{\sigma_k(i)} \boldsymbol{z}, \Theta_{\sigma_l(i)} \boldsymbol{z}$ are colinear, which is a contraction unless $\sigma_k(i) = \sigma_l(i)$. Therefore, $\sigma_k = \sigma_l$, as desired.

We now show $\mathcal{S}_2^k = \mathcal{S}_2^l$. Without loss of generality, assume there exists $i$ such that $i \in \mathcal{S}_2^k$ and $i \notin \mathcal{S}_2^l$. We fix such an $i$. Then, using equation 70 as well as $\sigma_k(i) = \sigma_l(i)$,

$$a_{\sigma_k(i)} \Theta_{\sigma_k(i)} \boldsymbol{z} = -a_{\sigma_l(i)} \Theta_{\sigma_l(i)} \boldsymbol{z} \tag{72}$$

$$= -a_{\sigma_k(i)} \Theta_{\sigma_k(i)} \boldsymbol{z} \tag{73}$$

and thus $a_{\sigma_k(i)} \Theta_{\sigma_k(i)} \boldsymbol{z} = 0$, which again is a contradiction of the non-colinearity of the rows of $\boldsymbol{W}(\Theta, \boldsymbol{z})$. The claim thus follows.

$\square$

The upcoming lemma is crucial for demonstrating that students almost surely generalize compositionally after achieving zero loss on a finite task sample from $\mathcal{P}_z$. Loosely speaking this Lemma is a generalization of Lemma 6 in that it asserts that under the condition that no two elements from $(\Theta_i)_i$ are identical up to scaling, a randomly sampled set of $M + 1$ task latent vectors has the following property: there exists a rescaling of the points such that the sum of rescaled points is zero (this trivially holds as we have more points than the dimension of the space) and such that by applying arbitrary sign inversions and multiplication by arbitrary elements of $(\Theta_i)_i$ to each scaled vector, the sum would not be zero unless the matrices and the sign inversion applied to all vectors are the same. In the lemma we choose the $M + 1$ points from the unit sphere, and one can think of them being chosen i.i.d. uniformly. Within the proof of Theorem 9 we will then argue that our assumptions on $\mathcal{P}_z$ allow us to replace this uniform sampling from the sphere by a sampling according to $\mathcal{P}_z$.

Henceforth, measures on the sphere are spherical measures (resp. product measure of spherical measures for Cartesian products of spheres), and Lebesgue measures in $\mathbb{R}^d$ for all $d$.

**Lemma 10.** *Let a family of $\mathbb{R}^{n \times d}$ matrices $(\Theta_i)_{1 \leq i \leq h}$. Assume furthermore that there exists at least one $\boldsymbol{z}_0 \in \mathbb{R}^d$ such that no two elements of $(\Theta_i \boldsymbol{z}_0)_i$ are colinear. Let $S$ be the unit sphere of $\mathbb{R}^d$.*

*Then, for almost every set of $d+1$ points $\hat{z}_1, \ldots, \hat{z}_{d+1}$ in $S^{d+1}$, there exists $\lambda_1, \ldots, \lambda_{d+1}$ such that $(z_e)_e := (\lambda_e \hat{z}_e)_e$ verifies the following conditions:*

$$\sum_{e \in [1..d+1]} z_e = 0$$

$$\forall (i_e)_e \in [1..h]^{d+1}, \forall (s_e)_e \in \{-1, 1\}^{d+1} : \tag{74}$$

$$\sum_{e \in [1..d+1]} s_e \Theta_{i_e} z_e = 0 \implies \forall k, l \in [1..d+1], (i_k = i_l) \wedge (s_k = s_l).$$

*Proof.* We define the matrix $\mathcal{I} = (I_1, I_2..I_{d+1})$ where $I_i$ for all $i$ is the $M \times M$ identity matrix, the isomorphism $\Phi : (z_e)_{e \in [1..d+1]} \to \bigoplus_{e \in [1..d+1]} z_e$ which concatenates $d+1$ vectors of $\mathbb{R}^d$ into a vector of $\mathbb{R}^{d(d+1)}$, and the projection to the unit sphere $P : x \to \frac{x}{\|x\|}$, defined everywhere except at the origin. Finally, let $C = \{(z_e)_{e \in [1..d+1]} \mid \exists Q \subsetneq [1..d+1] : (z_e)_{e \in Q} \text{ linearly dependent}\}$, i.e. the set of family of $d+1$ vectors such that not any subset of cardinality $d$ form a basis of $\mathbb{R}^d$.

The proof will proceed in the following steps:

1. First, we show that for almost any $x \in P(\operatorname{Ker} \mathcal{I}) \setminus \Phi(C)$, property 74 holds for $\Phi^{-1}(x)$.

2. Then, we show that there exists a surjective function $\Psi$ from $S^{d+1} \setminus C$ to $P(\operatorname{Ker} \mathcal{I}) \setminus \Phi(C)$, such that the pre-image of any zero-measure set is also zero-measure, and such that for all $(z_e)_{e \in [1..d+1]} \in S^{d+1} \setminus C$, $\Psi((z_e)_e)$ is a transformation which scales each member by a scalar and concatenates them, i.e. $\Psi((z_e)_e) \in \{\Psi(\lambda_e z_e)_e) \mid (\lambda_e)_{e \in [1..d+1] \in \mathbb{R}^{d+1}}\}$.

We remind the reader that if a surjective function $f : A \to B$ between two metric spaces $A, B$ equipped by metrics $\mu, \mu'$ verifies that all pre-image of nullsets are nullsets, i.e. $\forall X \subset B : \mu'(X) = 0 \implies \mu(f^{-1}(X)) = 0$, then we have that if some property is true for almost every $x \in B$, it must be true for $f(x)$ for almost every $x \in f^{-1}(B) = A$. If the above 2 points are true, then by application of this reasoning, we get that for almost any $(z_e)_e \in S^{d+1} \setminus C$, there exists $(\lambda_e)_e$ such that property 74 holds for $(\lambda_e z_e)_{e \in [1..d+1]}$. Since it can be shown that the measure of $S^{d+1} \setminus C$ is 0, we have that the assertion holds almost everywhere on $S^{d+1}$, which would conclude the proof.

In the remainder we will prove the two points above.

**Step 1:** Let some $i = (i_e)_e \in [1..h]^{d+1}$ and $s = (s_e)_e \in \{-1, 1\}^{d+1}$, such that there exists a pair $(n, m)$ such that $i_n \neq i_m$ or $s_n \neq s_m$. We fix such a pair. Let $M_{i,s} = (s_1 \Theta_{i_1}, ..s_{d+1}\Theta_{i_{d+1}})$. We will now show that $\operatorname{Ker} \mathcal{I} \not\subset \operatorname{Ker} M_{i,s}$, i.e. that there exists a point in $\operatorname{Ker} \mathcal{I}$ which is not in $\operatorname{Ker} M_{i,s}$. Let the family $(z_e)_e = (\mathbb{1}_{e=n \vee e=m}(-1)^{\mathbb{1}_{e=n}} z_0)_e$. Clearly, $\sum_e z_e = z_0 - z_0 = 0$, thus $\Phi((z_e)_e)$ belongs to the Kernel of $\mathcal{I}$. However, $\sum_e s_e \Theta_{i_e} z_e = 0 \implies s_n \Theta_{i_n} z_0 = -s_m \Theta_{i_m} z_0$. Because by assumption no two elements of $(\Theta_i z_0)_i$ are colinear, necessarily $\sum_e s_e \Theta_{i_e} z_e = 0 \implies (s_n = s_m) \wedge (i_n = i_m)$, which contradicts the definition of $i, s$. Therefore, necessarily $\sum_e s_e \Theta_{i_e} z_e \neq 0$, i.e. $\Phi((z_e)_e) \notin \operatorname{Ker} M_{i,s}$.

$\operatorname{Ker} M_{i,s}$ is a linear subspace, and $P(\operatorname{Ker} \mathcal{I})$ is a unit sphere living in a linear subspace. Since $\operatorname{Ker} \mathcal{I} \not\subset \operatorname{Ker} M_{i,s}$, then $P(\operatorname{Ker} \mathcal{I}) \not\subset \operatorname{Ker} M_{i,s}$ and thus $P(\operatorname{Ker} \mathcal{I}) \cap \operatorname{Ker} M_{i,s}$ has zero measure in $P(\operatorname{Ker} \mathcal{I})$. Because a finite union of nullsets is a nullset, it is then clear that $P(\operatorname{Ker} \mathcal{I}) \cap \bigcup_{i,s} \operatorname{Ker} M_{i,s}$ has zero measure in $P(\operatorname{Ker} \mathcal{I})$, where the union is done over $i = (i_e)_e \in [1..h]^{d+1}$ and $s = (s_e)_e \in \{-1, 1\}^{d+1}$, such that there exists a pair $(n, m)$ such that $i_n \neq i_m$ or $s_n \neq s_m$. On the other hand, it is clear that for any family $i, s$ such that all elements of $i$ resp. $s$ are the same, we have $P(\operatorname{Ker} \mathcal{I}) \subset \operatorname{Ker} M_{i,s}$.

This now proves that for almost any $x \in P(\operatorname{Ker} \mathcal{I})$, $\Phi^{-1}(x)$ must satisfy the property 74. We can see that $\Phi(C)$ has zero measure in $P(\operatorname{Ker} \mathcal{I})$, which finalizes the claim.

**Step 2:** Let $S^+ = \{(z_e)_{e \in [1..d+1]} \in S^{d+1} \mid \exists (\lambda_e)_{e \in [1..d+1]} : (\sum_e \lambda_e z_e = 0) \wedge (\forall e, \lambda_e > 0)\}$. We will define two surjective functions, $\Xi : \setminus C \to S^+ \setminus C$ and $\Upsilon : S^+ \setminus C \to P(\operatorname{Ker} \mathcal{I}) \setminus \Phi(C)$ such that their respective pre-image of nullsets are nullsets, and each of which multiplies each member of

the input $(z_e)_e$ by a scalar, before eventually concatenating them. If so, their composition $\Psi = \Upsilon \circ \Xi$ will inherit these properties, which finalizes the proof.

We first define $\Upsilon$. Notice that on $S^+ \setminus C$, the scalar family $(\lambda_e)_{e \in [1..d+1]}$ is unique up to a scalar multiplication. The family becomes unique if we furthermore restrict it to be of norm 1. We can thus define the function $\Upsilon$ which maps to each $(z_e)_{e \in [1..d+1]}$ the unique $\Phi((\lambda_e z_e)_{e \in [1..d+1]})$ such that $(\sum_e \lambda_e z_e = 0) \wedge (\forall e, \lambda_e > 0) \wedge (\sum_e \lambda_e^2 = 1)$. Clearly, $\Upsilon$ defines a bijection between $S^+ \setminus C$ and $P(\mathrm{Ker}\,\mathcal{I}) \setminus \Phi(C)$, with the inverse function $\Upsilon^{-1} : x \to (\frac{z_e}{\|z_e\|})_e$ with $(z_e)_e = \Phi^{-1}(z)$, defined on $P(\mathrm{Ker}\,\mathcal{I}) \setminus \Phi(C)$. Furthermore, $\Upsilon^{-1}$ is differentiable everywhere on $P(\mathrm{Ker}\,\mathcal{I}) \setminus \Phi(C)$, and thus preserves nullsets.

We now define $\Xi$. For all points $(z_e)_e$ in $S^{d+1} \setminus C$, there exists a family of non-zero scalars $(\lambda_e)_e$, unique up to a scalar multiplication, such that $\sum_e \lambda_e z_e = 0$. Given an arbitrary such $(\lambda_e)_e$ for which $\lambda_1 > 0$, we can define the function $\Xi$ from $S^{d+1} \setminus C$ to $S^+ \setminus C$ as $\Xi : (z_e)_e \to (\mathrm{sign}(\lambda_e) z_e)_e$. $\Xi$ is surjective, as it leaves all elements of $S^+ \setminus C \subset S^{d+1} \setminus C$ unchanged. Furthermore, the preimage of any set $X$ by $\Xi$ is included in the finite union $\bigcup_s X_s$, where $s \in \{-1, 1\}^{d+1}$, and $X_s = \{(z_e)_e \mid (s_e z_e)_e \in X\}$. Since $X_s$ is trivially a nullset if $X$ is a nullset, the preimage of nullsets by $\Xi$ are nullsets.

$\square$

We can now prove the part of the theorem stating that a final sample from the training task distribution will almost surely allow the student to generalize compositionally.

*Proof of Theorem 9, second part.* Observe that it suffices to show that for any $k$, equation 63 holds almost surely when sampling $d_k + 1$ points i.i.d from the probability distribution $\mathcal{P}_{z \in Z_k}$. The remainder of the proof is then identical to the continuous case.

Fix some set $Z_k$. As in the continuous case we may assume without loss of generality that all points $z$ in $Z_k$ satisfy that no two rows of $W(\Theta, z)$ are colinear and that they achieve zero loss, i.e. $\min_{\hat{z}} L(\hat{z}, (\hat{\Theta}, \hat{a}); \mathcal{D}_z) = 0$.

Now, consider $\mathcal{F}_k = (z_{k,e})_{1 \le e \le d_k + 1}$, a family of $d_k + 1$ task vectors sampled i.i.d. from $Z_k$ following $\mathcal{P}_{z \in Z_k}$. Due to the assumption that $\mathcal{P}_{z \in Z_k}$ projected onto the unit sphere admits a probability density function, any subset of size $d_k$ of $\mathcal{F}_k$ form a basis of $\mathrm{span}(Z_k)$ almost surely, and in particular $\mathrm{span}((z_{k,e})_{1 \le e \le d_k + 1}) = \mathrm{span}(Z_k)$ with probability one. Furthermore, we can apply Theorem 3 to each vector individually. If we can show that the resulting permutations $\sigma$ and partitions $\mathcal{S}_1, \mathcal{S}_2$ are identical across these vectors, then that would suffice to conclude the proof.

In the previous scenario where the loss was zero at an infinite number of points in $Z_k$, it was possible to extract a basis from $\mathrm{span}(Z_k)$, with each basis element having a consistent permutation and partition. This strategy, however, is not applicable in the current context. To establish the uniformity of permutations and partitions for all vectors, we must first refer to Lemma 10, and show that with probability 1, $\mathcal{F}_k$ has the property that there exists a family of scalar $(\lambda_{k,e})$ such that $(\lambda_{k,e} z_{k,e})_{1 \le e \le d_k + 1}$ verifies equation 74. We will prove this next.

Because we have that any vector $z$ in $Z_k$ satisfy that no two rows of $W(\Theta, z)$ are colinear, equivalently, no two elements of $(a_i \Theta_i z)_i$ have two colinear elements. We can thus apply the lemma to $(a_i \Theta_i)_i$, which would guarantee that almost everywhere on $S^{M+1}$ where $S$ is the unit sphere of $\mathbb{R}^M$, the property holds. In fact, if we consider the sphere $S_k$ of the linear subspace $\mathrm{span}(Z_k)$, it is possible to show that the property holds everywhere on $S_k^{d_k+1}$ except for a set of measure zero. Because $\mathcal{P}_{z \in Z_k}$ projected onto the unit sphere admits a probability density function, it is easy to see that when sampling i.i.d. $d_k + 1$ samples from $\mathcal{P}_{z \in Z_k}$ and projecting each of them on the unit sphere, the family will be on this set of measure zero with probability zero. Because whether the property holds or not is left unchanged by normalization of each vectors in the family, clearly the property must hold with probability 1 for $\mathcal{F}_k$. This concludes the claim.

We now show that all the permutations and partitions associated with the vectors in $\mathcal{F}_k$ are the same.

For all $z$, we fix $\hat{z} \in \arg\min_{\hat{z}} L(\hat{z}, (\hat{\Theta}, \hat{a}); \mathcal{D}_z)$. As argued above already, it follows from Theorem 3 that for any $z$ in $\mathcal{F}_k$, there exists a permutation and partition of $[1..h]$, $\sigma_z$ and $(\mathcal{S}_1(z), \mathcal{S}_2(z))$

such that we have

$$\forall i \in \mathcal{S}_1(\boldsymbol{z}), \hat{a}_i\hat{\Theta}_i\hat{\boldsymbol{z}} = a_{\sigma_{\boldsymbol{z}}(i)}\Theta_{\sigma_{\boldsymbol{z}}(i)}\boldsymbol{z} \tag{75}$$

$$\forall i \in \mathcal{S}_2(\boldsymbol{z}), \hat{a}_i\hat{\Theta}_i\hat{\boldsymbol{z}} = -a_{\sigma_{\boldsymbol{z}}(i)}\Theta_{\sigma_{\boldsymbol{z}}(i)}\boldsymbol{z} \tag{76}$$

$$\sum_i \hat{a}_i\hat{\Theta}_i\hat{\boldsymbol{z}} = \sum_i a_i\Theta_i\boldsymbol{z} \tag{77}$$

Similarly as before, since $\mathrm{span}(\bigcup_k \mathcal{F}_k) = \mathrm{span}(\bigcup_k Z_k) = \mathbb{R}^M$, the last equality can be used to obtain an invertible matrix $\boldsymbol{F}$ such that for any $k$, and any $\boldsymbol{z}$ in $\mathcal{F}_k$, $\hat{\boldsymbol{z}} = \boldsymbol{F}\boldsymbol{z}$, which establishes a linear mapping from $\boldsymbol{z}$ to $\hat{\boldsymbol{z}}$.

In the remainder, we drop the index in $k$ for notational simplicity. We fix the family of scalars $(\lambda_e)$ such that equation 74 holds for $(\lambda_e\boldsymbol{z}_e)_e$.

In particular, we have $\sum_e \lambda_e\boldsymbol{z}_e = 0$. By multiplying by $\boldsymbol{F}$, we have $\sum_e \lambda_e\hat{\boldsymbol{z}}_e = 0$.

For any $i$, multiplying by $\hat{a}_i\hat{\Theta}_i$,

$$\sum_e \lambda_e\hat{a}_i\hat{\Theta}_i\hat{\boldsymbol{z}}_e = 0 \tag{78}$$

which in turn implies

$$\sum_e (-1)^{\mathbb{1}_{i\in\mathcal{S}_2(\boldsymbol{z}_e)}} a_{\sigma_e(i)}\Theta_{\sigma_{\boldsymbol{z}_e}(i)}\lambda_e\boldsymbol{z}_e = 0 \tag{79}$$

Since equation 74 holds for $(\lambda_e\boldsymbol{z}_e)_e$, clearly, $\forall i \in [1..h], \forall e, f, \sigma_{\boldsymbol{z}_e}(i) = \sigma_{\boldsymbol{z}_f}(i), \mathcal{S}_1(\boldsymbol{z}_e) = \mathcal{S}_1(\boldsymbol{z}_f)$ and $\mathcal{S}_2(\boldsymbol{z}_e) = \mathcal{S}_2(\boldsymbol{z}_f)$, as desired.

$\square$

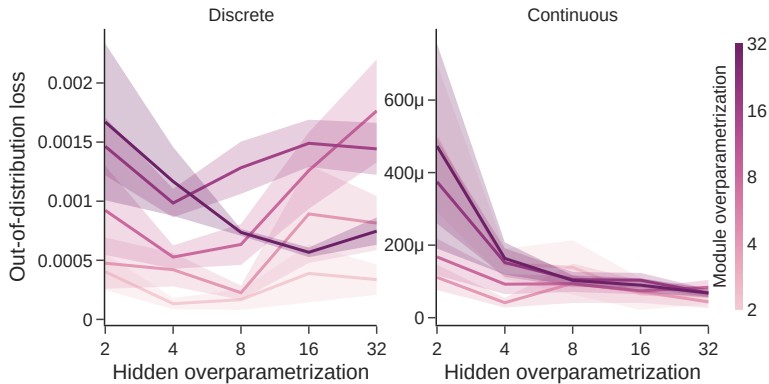

Figure A2: Out-of-distribution loss is sensitive to overparameterization. Numbers denote the factor by which the student dimension is larger than the teacher. Error bars denote the standard error of the mean over 3 seeds.

Table A1: Effect of weight decay on module alignment in the overparameterized regime. Numbers are module alignment averaged over 5 seeds.

| Weight decay strength | $\hat{M} = 16 \cdot M$ | | $\hat{M} = 32 \cdot M$ | |
| | $\hat{h} = 16 \cdot h$ | $\hat{h} = 32 * h$ | $\hat{h} = 16 \cdot h$ | $\hat{h} = 32 \cdot h$ |
| --- | --- | --- | --- | --- |
| 0.001 | 0.7704 | 0.7165 | 0.8369 | 0.8144 |
| 0.0001 | 0.7699 | 0.6752 | 0.8483 | 0.8043 |
| 0.00001 | 0.7633 | 0.6727 | 0.8346 | 0.8292 |
| 0 | 0.7542 | 0.7109 | 0.8421 | 0.8076 |

# B  ADDITIONAL EXPERIMENTS

## B.1  MULTI-TASK TEACHER-STUDENT

**Overparameterization hurts compositional generalization.**  We show in Figure A2 the out-of-disribution (OOD) loss for the experiment presented in Figure 2C. We see that the degradation in OOD loss mirrors that of identification.

**Weight decay does not improve module alignment of overparameterized models.**  We repeat the overparameterization experiment with varying weight decay strengths, for the discrete task distribution. We report the results in Table A1. We see that regularization using weight decay does not seem to help to improve module alignment for overparameterized models.

## B.2  HYPERTEACHER

**Sample complexity scales with the number of tasks.**  We investigated empirically, how many training tasks we need to present in order to obtain a specified OOD accuracy. Figure A3 shows this scaling behavior.

**Sensitivity to finite data samples for task inference.**  In contrast to the infinite data regime considered in the theory, in the hyperteacher we rely on gradient-based meta-learning from finite samples. We investigated the dependence of our results on the number of train shots in the support and query set for each task in Figure A4, finding that reducing the number from the $N = 256$ we consider throughout our experiments leads to a decrease in performance.

**Sensitivity to fraction of held-out module combinations during training.**  In our main results we hold-out 25% of possible module combinations during training to evaluate OOD accuracy and quantify compositional generalization. In Figure A5 we vary this fraction for $M \in \{4, 8, 16\}$ and $K = 3$ revealing that for larger numbers of modules, OOD accuracy stays increasingly robust across

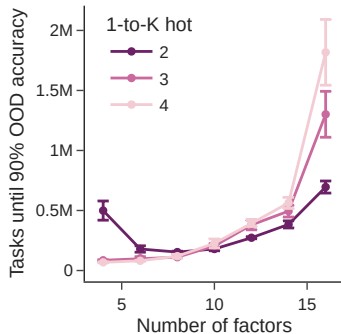

Figure A3: Number of tasks encountered during training before achieving 90% OOD accuracy across modules $M$ and maximum number of modules per task $K$ in the hyperteacher.

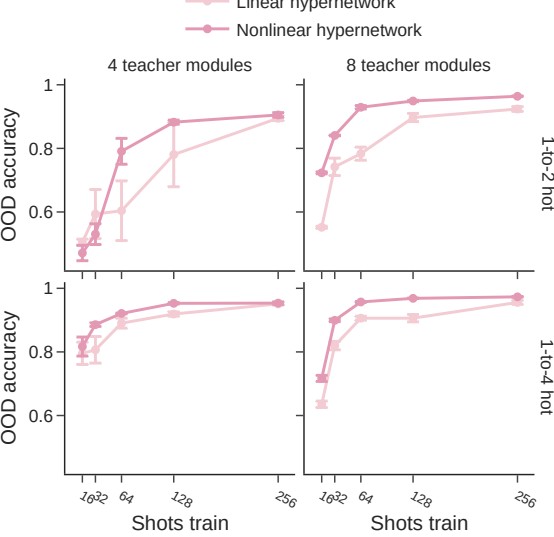

Figure A4: Influence of the number of train shots on OOD accuracy in hyperteacher for $M \in \{4, 8\}$ and $K \in \{2, 4\}$.

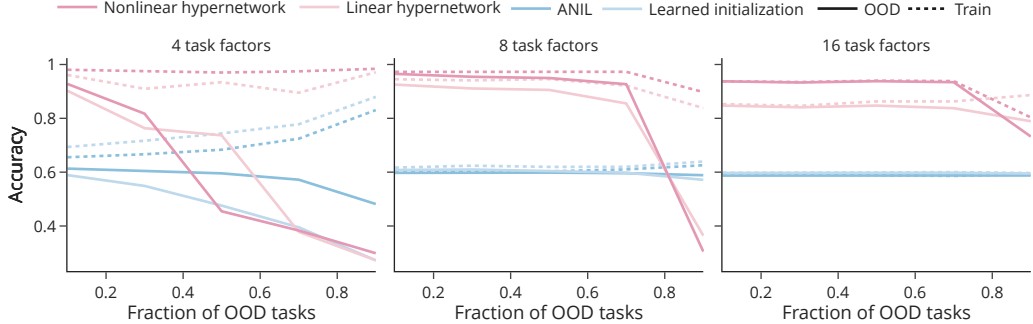

Figure A5: Sensitivity of OOD accuracy to the fraction of held-out module combinations during training for the OOD set in the hyperteacher.

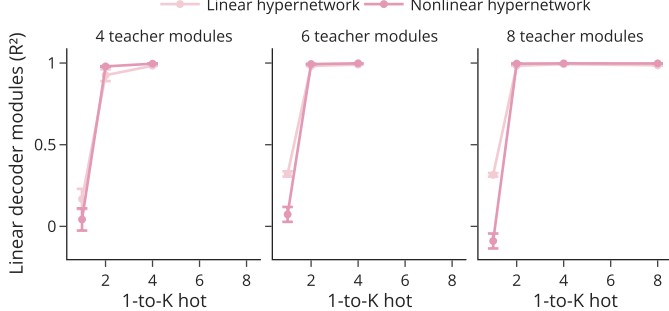

Figure A6: Linear decodability for varying number of modules $M \in \{4, 6, 8\}$ and maximum number of modules per combination $K \in \{1, 2, 4, 8\}$ in the hyperteacher.

larger fractions of tasks held-out during training. Note, that at the same time as expected training accuracy remains comparably stable.

**Linear decodability for varying number of modules and combination size.** To complement Figure 3E in the main text, we show the corresponding linear decodability across modules $M \in \{4, 6, 8\}$ and maximum number of modules per combination $K \in \{1, 2, 4, 8\}$ in Figure A6.

**Detailed metrics.** For completeness we provide detailed training metrics for all methods across a range of settings for $M \in \{4, 8\}$ and $K \in \{1, 2, 4, 8\}$ in Tables A2, A3, A4, A5, A6, A7, A8 and present training curves in Figure A7.

## B.3 COMPOSITIONAL PREFERENCES

**Overparameterization in the compositional preference environment.** Motivated by the sensitivity to overparameterization in the multi-task teacher-student setting, we investigated the effect of varying the hidden dimension and number of modules in the hypernetwork models on the OOD loss

Table A2: Hyperteacher comparison of accuracies across models for M=4 K=1.

|  | ANIL | MAML | Linear hypernetwork | Nonlinear hypernetwork |
|---|---|---|---|---|
| Train accuracy | $79.68 \pm 0.25$ | $85.14 \pm 0.57$ | $80.98 \pm 8.92$ | $97.31 \pm 0.76$ |
| Test accuracy | $79.72 \pm 0.12$ | $85.13 \pm 0.54$ | $80.89 \pm 9.1$ | $97.15 \pm 0.93$ |
| OOD accuracy | $46.52 \pm 0.8$ | $20.53 \pm 0.72$ | $20.42 \pm 1.64$ | $20.93 \pm 2.22$ |
| OOD accuracy (K=1) | $46.48 \pm 0.83$ | $20.58 \pm 0.74$ | $20.59 \pm 1.52$ | $20.9 \pm 2.19$ |
| OOD accuracy (K=2) | $50.47 \pm 0.26$ | $26.71 \pm 0.26$ | $27.8 \pm 2.39$ | $26.78 \pm 3.33$ |
| OOD accuracy (K=4) | $48.18 \pm 1.2$ | $22.74 \pm 0.59$ | $24.57 \pm 2.57$ | $23.04 \pm 2.86$ |

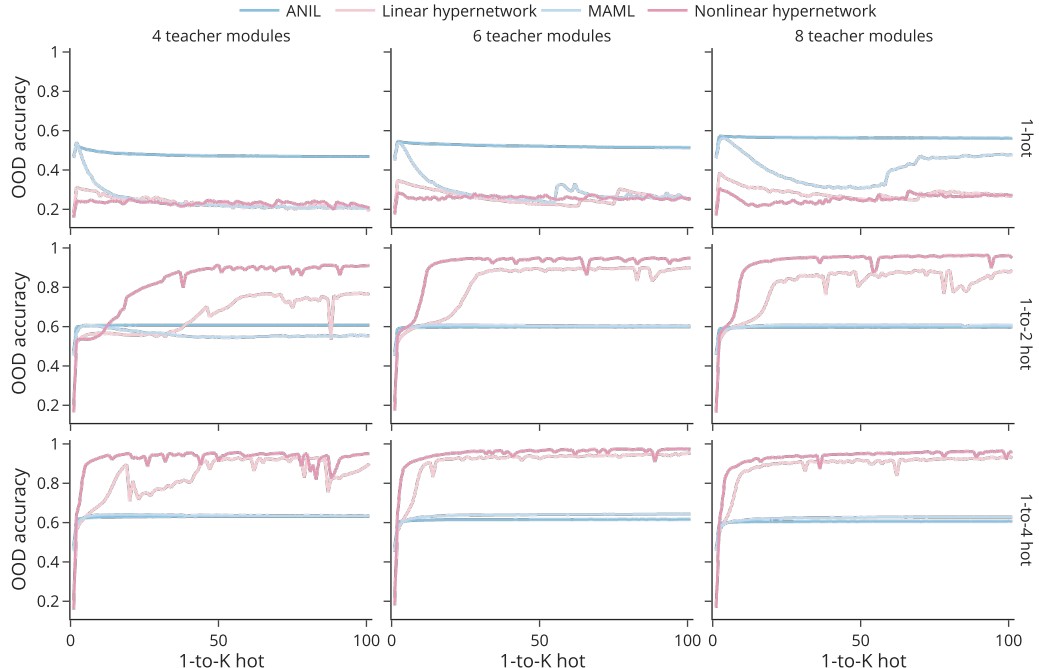

Figure A7: OOD accuracy over the course of training for various models, number of modules $M \in \{4, 6, 8\}$ and maximum number of modules per combination $K \in \{1, 2, 4\}$.

Table A3: Hyperteacher comparison of accuracies across models for M=4 K=2.

|  | ANIL | MAML | Linear hypernetwork | Nonlinear hypernetwork |
|---|---|---|---|---|
| Train accuracy | $66.45 \pm 0.23$ | $68.36 \pm 1.3$ | $88.32 \pm 4.83$ | $98.05 \pm 0.09$ |
| Test accuracy | $66.17 \pm 0.06$ | $68.11 \pm 1.02$ | $88.29 \pm 5.09$ | $98.0 \pm 0.05$ |
| OOD accuracy | $60.78 \pm 0.24$ | $55.41 \pm 0.8$ | $76.35 \pm 10.11$ | $90.9 \pm 0.26$ |
| OOD accuracy (K=1) | $59.12 \pm 0.98$ | $53.33 \pm 1.17$ | $71.32 \pm 9.97$ | $89.24 \pm 0.88$ |
| OOD accuracy (K=2) | $61.38 \pm 0.21$ | $55.98 \pm 0.53$ | $77.41 \pm 10.18$ | $90.79 \pm 0.58$ |
| OOD accuracy (K=4) | $60.61 \pm 1.46$ | $50.33 \pm 1.36$ | $71.96 \pm 14.18$ | $68.31 \pm 6.31$ |

Table A4: Hyperteacher comparison of accuracies across models for M=4 K=4.

|  | ANIL | MAML | Linear hypernetwork | Nonlinear hypernetwork |
|---|---|---|---|---|
| Train accuracy | $65.63 \pm 0.32$ | $68.91 \pm 0.28$ | $92.22 \pm 4.68$ | $97.43 \pm 0.36$ |
| Test accuracy | $65.58 \pm 0.25$ | $68.72 \pm 0.3$ | $92.29 \pm 4.66$ | $97.49 \pm 0.27$ |
| OOD accuracy | $63.14 \pm 0.7$ | $63.29 \pm 0.6$ | $88.61 \pm 6.15$ | $94.9 \pm 0.23$ |
| OOD accuracy (K=1) | $59.77 \pm 1.23$ | $56.29 \pm 1.54$ | $82.55 \pm 8.17$ | $91.79 \pm 0.52$ |
| OOD accuracy (K=2) | $62.82 \pm 0.9$ | $63.16 \pm 0.83$ | $88.67 \pm 6.5$ | $95.13 \pm 0.06$ |
| OOD accuracy (K=4) | $65.55 \pm 3.67$ | $66.69 \pm 3.38$ | $89.0 \pm 7.52$ | $95.63 \pm 1.46$ |

Table A5: Hyperteacher comparison of accuracies across models for M=8 K=1.

|  | ANIL | MAML | Linear hypernetwork | Nonlinear hypernetwork |
|---|---|---|---|---|
| Train accuracy | $66.22 \pm 0.64$ | $65.41 \pm 1.16$ | $88.13 \pm 2.69$ | $95.85 \pm 0.12$ |
| Test accuracy | $66.01 \pm 0.49$ | $65.16 \pm 1.35$ | $87.91 \pm 2.87$ | $95.72 \pm 0.08$ |
| OOD accuracy | $55.8 \pm 0.26$ | $47.42 \pm 2.16$ | $26.67 \pm 3.12$ | $26.91 \pm 0.4$ |
| OOD accuracy (K=1) | $56.06 \pm 0.24$ | $47.68 \pm 2.25$ | $26.73 \pm 3.12$ | $26.9 \pm 0.49$ |
| OOD accuracy (K=2) | $56.9 \pm 0.21$ | $49.21 \pm 2.35$ | $30.6 \pm 3.38$ | $30.45 \pm 0.52$ |
| OOD accuracy (K=4) | $55.77 \pm 0.28$ | $47.63 \pm 2.14$ | $26.51 \pm 3.28$ | $26.5 \pm 0.75$ |
| OOD accuracy (K=8) | $54.07 \pm 0.31$ | $45.77 \pm 2.24$ | $21.85 \pm 2.93$ | $22.81 \pm 1.3$ |

Table A6: Hyperteacher comparison of accuracies across models for M=8 K=2.

|  | ANIL | MAML | Linear hypernetwork | Nonlinear hypernetwork |
|---|---|---|---|---|
| Train accuracy | $60.0 \pm 0.49$ | $61.33 \pm 0.68$ | $89.9 \pm 2.77$ | $96.91 \pm 0.25$ |
| Test accuracy | $60.07 \pm 0.31$ | $61.33 \pm 0.45$ | $90.06 \pm 2.56$ | $96.95 \pm 0.22$ |
| OOD accuracy | $59.76 \pm 0.27$ | $60.6 \pm 0.37$ | $87.45 \pm 2.55$ | $96.05 \pm 0.34$ |
| OOD accuracy (K=1) | $59.53 \pm 0.56$ | $60.76 \pm 0.71$ | $86.73 \pm 2.56$ | $96.01 \pm 0.53$ |
| OOD accuracy (K=2) | $59.76 \pm 0.33$ | $60.48 \pm 0.31$ | $87.49 \pm 2.35$ | $95.9 \pm 0.39$ |
| OOD accuracy (K=4) | $59.84 \pm 0.24$ | $60.01 \pm 0.31$ | $82.56 \pm 1.65$ | $77.79 \pm 1.32$ |
| OOD accuracy (K=8) | $58.6 \pm 0.22$ | $57.99 \pm 0.32$ | $60.6 \pm 6.61$ | $33.08 \pm 1.94$ |

Table A7: Hyperteacher comparison of accuracies across models for M=8 K=4.

|  | ANIL | MAML | Linear hypernetwork | Nonlinear hypernetwork |
|---|---|---|---|---|
| Train accuracy | $60.9 \pm 0.44$ | $63.02 \pm 0.62$ | $93.64 \pm 0.18$ | $95.78 \pm 0.61$ |
| Test accuracy | $60.64 \pm 0.3$ | $62.93 \pm 0.37$ | $93.57 \pm 0.25$ | $95.71 \pm 0.55$ |
| OOD accuracy | $60.63 \pm 0.21$ | $62.78 \pm 0.24$ | $93.5 \pm 0.24$ | $95.68 \pm 0.55$ |
| OOD accuracy (K=1) | $58.77 \pm 0.31$ | $58.81 \pm 0.38$ | $86.81 \pm 0.31$ | $92.6 \pm 1.47$ |
| OOD accuracy (K=2) | $59.83 \pm 0.4$ | $61.39 \pm 0.45$ | $92.3 \pm 0.22$ | $95.25 \pm 0.7$ |
| OOD accuracy (K=4) | $61.2 \pm 0.19$ | $64.0 \pm 0.2$ | $93.92 \pm 0.17$ | $95.88 \pm 0.51$ |
| OOD accuracy (K=8) | $62.32 \pm 0.26$ | $62.85 \pm 0.23$ | $88.18 \pm 1.03$ | $84.43 \pm 4.49$ |

Table A8: Hyperteacher comparison of accuracies across models for M=8 K=8.

|  | ANIL | MAML | Linear hypernetwork | Nonlinear hypernetwork |
|---|---|---|---|---|
| Train accuracy | $61.41 \pm 0.37$ | $63.87 \pm 0.38$ | $91.54 \pm 0.89$ | $94.83 \pm 0.1$ |
| Test accuracy | $61.41 \pm 0.28$ | $64.05 \pm 0.19$ | $91.56 \pm 0.71$ | $94.71 \pm 0.24$ |
| OOD accuracy | $61.53 \pm 0.26$ | $64.11 \pm 0.24$ | $91.65 \pm 0.72$ | $94.75 \pm 0.24$ |
| OOD accuracy (K=1) | $57.71 \pm 0.1$ | $57.75 \pm 0.22$ | $83.5 \pm 0.9$ | $89.26 \pm 0.95$ |
| OOD accuracy (K=2) | $58.99 \pm 0.42$ | $59.77 \pm 0.56$ | $88.32 \pm 1.06$ | $92.97 \pm 0.23$ |
| OOD accuracy (K=4) | $61.37 \pm 0.31$ | $64.12 \pm 0.2$ | $91.75 \pm 0.69$ | $94.82 \pm 0.15$ |
| OOD accuracy (K=8) | $63.63 \pm 1.52$ | $66.73 \pm 1.86$ | $92.04 \pm 0.54$ | $95.49 \pm 0.36$ |

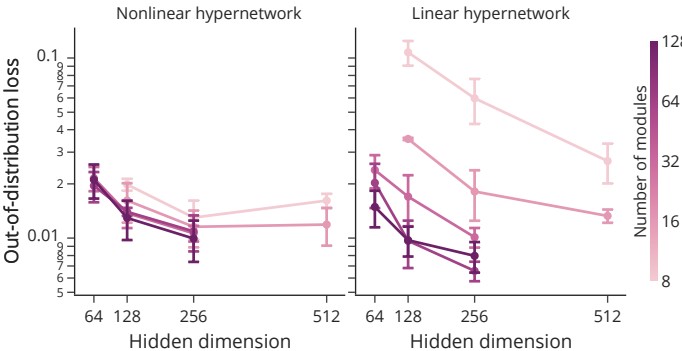

Figure A8: Effect of overparameterization on the OOD loss in the compositional preference environment.

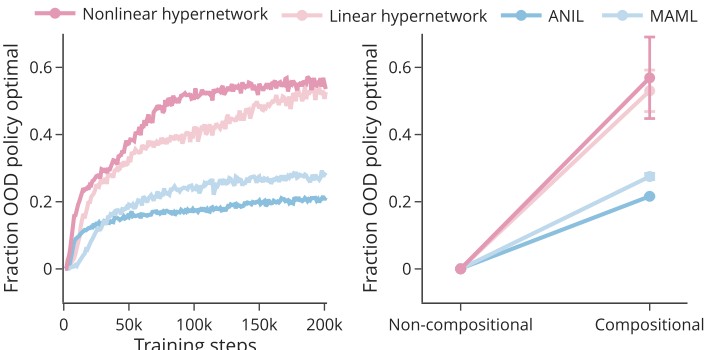

Figure A9: Complementary figure to Figure 4E and 4F reporting an additional metric that measures the fraction of times the learned policy exactly follows the optimal policy along the whole path and therefore obtains the only environment reward by reaching the goal object. Error bars denote the standard error of the mean over 3 seeds.

achieved in the compositional preference environment. Figure A8 shows that within the range of values we were able to accommodate given the maximum GPU memory available to us, overparameterization seems to have no negative influence on the OOD loss. This is in line with our empirical observations for the continuous multi-task teacher-student setting, as preferences for each task are continuous combinations of the selected modules.

### B.4 COMPOSITIONAL GOALS

**Modular architectures outperform monolithic architectures by leveraging compositional structure.** Figure 4E and 4F report the out-of-distribution accuracy of the learned policy with respect to the ground-truth optimal policy. In Figure A9, we complement it with another metric that measures the fraction of times the learned policy exactly follows the optimal policy along the whole path and therefore obtains the only environment reward by reaching the goal object. It is possible that the learned policies only deviate from the optimal trajectory in a way that would also yield the reward which would not count towards this fraction.

**Overparameterization in the compositional goal environment.** Similarly, we investigated the effect of varying the hidden dimension and number of modules in the hypernetwork models on the OOD accuracy achieved in the compositional goals environment. Different from the compositional preference environment, we have a discrete set of goals in this case. While in the discrete multi-task teacher-student setting OOD performance was sensitive to overparameterization, Figure A10 shows relatively stable OOD accuracy across varying hidden and module dimension.

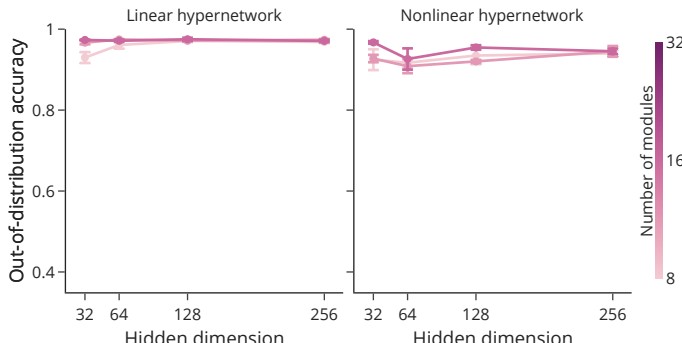

Figure A10: Effect of overparameterization on the OOD accuracy in the compositional goals environment.

## C EXPERIMENTAL DETAILS

### C.1 MULTI-TASK TEACHER-STUDENT SETUP

#### C.1.1 DATA GENERATION

We generate data by first initializing the teacher parameters once. Then, for each task, we sample a task latent variable $z$ which induces a noiseless mapping from inputs $x$ to targets $y$ following equation 3, modulo a rescaling factor by the square root of the input dimension $n$:

$$y = a\psi(\frac{1}{\sqrt{n}}W(\Theta, z)x) \qquad (80)$$

For all experiments in the multi-task teacher student, we fix the activation function $\psi$ to the ReLU nonlinearity, the task latent variable dimension to $M = 6$, input dimension to $n = 16$, hidden dimension of the teacher to $h = 16$, and output dimension to $o = 4$. The teacher parameters $\Theta$ and $a$ are generated by sampling the entries i.i.d. from the centered truncated normal distribution, with standard deviation resp. $\frac{1}{\sqrt{M}}$ and $\frac{1}{\sqrt{h}}$. We define the distribution over inputs $x$ to be the uniform distribution with 0 mean and standard deviation 1 on $\mathbb{R}^n$. Finally, we specify the distribution over task latent variable $z$. We distinguish 2 settings:

**Continuous task distribution.** Here, we consider tasks where modules are sparsely, linearly combined. A task distribution is specified by a set of masks, that are binary vectors of $\mathbb{R}^M$. Given a mask, we sample a task $z$ as follows. We first sample an $M$-dimensional random variable following the exponential distribution. Then, we zero-out entries corresponding to the mask being 0. We then normalize the vector such that the sum equals 1. This procedure simulates the uniform sampling from the simplex spanning the directions in which the mask is non-zero. Finally, we add the mask to the vector and rescale the outcome by $0.5$. This ensures two tasks generated by distinct masks do not have intersecting support (but intersecting span). See Algorithm 1 for the pseudocode.

---

**Algorithm 1** Algorithm to sample the tast latent variable from given a mask.

---

**Require:** mask $m$ of size $M$ **return** task latent variable $z$
   sample $M$-dimensional vector $z$ from the exponential distribution
   $z \leftarrow z \odot m$
   $z \leftarrow \frac{z}{1+\|z\|_1}$
   $z \leftarrow 0.5 \cdot (z + m)$
   return $z$

---

The task distribution is then generated as follows: first, a mask is sampled randomly and uniformly from the pre-specified set. Then, the vector $z$ is sampled following the above procedure.

Table A9: Training task distribution used for our experiments. Each column defines a distribution. For discrete settings, the set of vectors specify those in the training set. We omit the normalizing factor for simplicity. For continuous settings, the vectors specify the set of masks used in the training set to generate the distribution.

| Discrete Connected | Disconnected | Continuous Connected | | Disconnected | | |
|---|---|---|---|---|---|---|
| (1,0,0,0,0,0) | (1,0,0,0,0,0) | (1,1,0,0,0,0) | (1,1,1,0,0,0) | (1,1,0,0,0,0) | (1,0,0,0,0,0) | (1,1,1,0,0,0) |
| (0,1,0,0,0,0) | (0,1,0,0,0,0) | (0,1,1,0,0,0) | (0,0,1,1,1,0) | (0,1,1,0,0,0) | (0,1,0,0,0,0) | (0,0,0,1,1,1) |
| (0,0,1,0,0,0) | (0,0,1,0,0,0) | (0,0,1,1,0,0) | (1,0,0,0,1,1) | (1,0,1,0,0,0) | (0,0,1,0,0,0) | (1,1,0,0,0,0) |
| (0,0,0,1,0,0) | (0,0,0,1,0,0) | (0,0,0,1,1,0) | | (0,0,0,1,1,0) | (0,0,0,1,0,0) | (0,1,1,0,0,0) |
| (0,0,0,0,1,0) | (0,0,0,0,1,0) | (0,0,0,0,1,1) | | (0,0,0,0,1,1) | (0,0,0,0,1,0) | (1,0,1,0,0,0) |
| (0,0,0,0,0,1) | (0,0,0,0,0,1) | (1,0,0,0,0,1) | | (0,0,0,1,0,1) | (0,0,0,0,0,1) | (0,0,0,1,1,0) |
| (1,1,0,0,0,0) | (1,1,0,0,0,0) | | | | | (0,0,0,0,1,1) |
| (0,1,1,0,0,0) | (0,1,1,0,0,0) | | | | | (0,0,0,1,0,1) |
| (0,0,1,1,0,0) | (1,0,1,0,0,0) | | | | | |
| (0,0,0,1,1,0) | (0,0,0,1,1,0) | | | | | |
| (0,0,0,0,1,1) | (0,0,0,0,1,1) | | | | | |
| (1,0,0,0,0,1) | (0,0,0,1,0,1) | | | | | |

**Discrete task distribution.** Here, we focus on task latent variables that are simple normalized many-hot vectors. For example, for $M = 4$, we would consider vectors such as $(0, 0, 0, 1)$, $\frac{1}{\sqrt{2}}(0, 1, 0, 1)$ or $\frac{1}{\sqrt{3}}(0, 1, 1, 1)$. By only considering such variables, we wish to model tasks where modules are combined sparsely without varying the magnitudes with which modules enter the composition. There are a finite number of such vectors, and thus the task distribution is a uniform mixture of diracs. We define the training task distribution by manually specifying which of these vectors are in the distribution.

### C.1.2 STUDENT MODEL

We use the same parameterization and parameter initialization scheme for the student as for the teacher, but we vary the hidden layer width $\hat{h}$ and inferred task latent variable dimension $\hat{M}$.

### C.1.3 TRAINING AND EVALUATION

---
**Algorithm 2** Bilevel training procedure
---
**Require:** outer-batch size $B_{\text{outer}}$, inner batch-size $B_{\text{inner}}$, number of outer steps $N_{\text{outer}}$, number of inner steps $N_{\text{inner}}$, outer learning rate $\eta_{\text{outer}}$, inner learning rate $\eta_{\text{inner}}$
$\quad$ **for** $N_{\text{outer}}$ iterations **do**
$\quad\quad$ Sample $B_{\text{outer}}$ task latent $(\boldsymbol{z}_k)_k$
$\quad\quad$ $\Delta\theta \leftarrow 0$
$\quad\quad$ **for** $k$ **do**
$\quad\quad\quad$ $\phi \leftarrow \phi_0$
$\quad\quad\quad$ **for** $N_{\text{inner}}$ iterations **do**
$\quad\quad\quad\quad$ Sample $B_{\text{inner}}$ data $((x_i, y_i))_i$ from $\mathcal{D}^{\text{support}}(\boldsymbol{z}_k)$
$\quad\quad\quad\quad$ $\phi \leftarrow \phi - \eta_{\text{inner}}\nabla_\phi L(\theta, \phi, ((x_i, y_i))_i)$
$\quad\quad\quad$ **end for**
$\quad\quad\quad$ Sample $B_{\text{inner}}$ data $((x_i, y_i))_i$ from $\mathcal{D}^{\text{query}}(\boldsymbol{z}_k)$
$\quad\quad\quad$ $\Delta\theta \leftarrow \Delta\theta + \nabla_\theta L(\theta, \phi, ((x_i, y_i))_i)$
$\quad\quad$ **end for**
$\quad\quad$ $\theta \leftarrow \theta - \eta_{\text{outer}}\Delta\theta$
$\quad$ **end for**
---

Since there can be an infinite amount of distinct tasks for continuous task distributions, we adopt a training procedure that allows for training in a multi-task setting with infinite tasks. For this purpose, we take the standard algorithm to optimize the cross-validation bilevel problem, detailed in Algorithm 2, and adapt it to the infinite-data multi-task case, using as the fast parameter $\phi$ the

inferred task latent variable $\hat{z}$ and the meta parameters $\theta$ the remainder of the student parameters, $\hat{\Theta}, \hat{a}$.

Concretely, to stay close to the theory, we use the whole task distribution for both support and query set, in particular i.e. $\mathcal{D}^{\text{support}}(z_k) = \mathcal{D}^{\text{query}}(z_k)$. In this case, at equilibrium (over the distribution), we know that the total derivative in $\theta$ does not involve any second-order gradients and can be computed efficiently with partial derivatives. Therefore we ensure the inner loop converges by allowing for a large number of inner steps $N_{\text{inner}}$, and apply the stop_grad operation on $\phi$ before computing the gradient w.r.t $\theta$.

While typically the fast parameter initialization $\phi_0$ can also be learned, here for simplicity we randomly sample the initial value from a normal distribution and keep it fixed throughout training.

In order to measure out-of-distribution performance at the end of training, we wish to sample new task latent vectors $z$ involving unseen combinations of modules, and evaluate the ability of the student to fit the corresponding data. We follow the same inner loop procedure as during training, and report the average outer loss over the task distribution, as well as the module alignment metric described in Section 3.3.

### C.1.4 EXPERIMENTS

We conduct 2 experiments, corresponding to Figure 2B and C.

In the first experiment (Figure 2B), we wish to investigate the effect of the connected support on module identification. We manually select task distributions that have, or do not have the connected support property, for both continuous and discrete task distributions. See Table A9 for the chosen tasks.

In the second experiment (Figure 2C), we investigate the effect of overparameterization on identifiability when the task distribution has connected and compositional support. For both continuous and discrete distributions, we use the task corresponding to the first column of the appropriate section in Table A9.

The training loss on all experiments reached a value lower than $10^{-7}$.

### C.1.5 HYPERPARAMETERS

For all experiments in the multi-task teacher student, we set $B_{\text{outer}} = 64$, $B_{\text{inner}} = 256$, $N_{\text{outer}} = 60000$, $N_{\text{inner}} = 300$. We optimize the inner loop using the Adam optimizer with $\eta_{\text{inner}} = 0.003$, and the outer loop using the AdamW optimizer with various weight decay strengths and an initial learning rate $\eta_{\text{outer}} = 0.001$ annealed using cosine annealing down to $10^{-6}$ at the end of training. We observed that this set of hyperparameters gave enough time for the student to practically reach 0 loss, which is the setting studied in the theory.

## C.2 HYPERTEACHER

### C.2.1 DATA GENERATION

We generate data by first initializing the teacher parameters once. The teacher network is a linear hypernetwork for which the parameterization and initialization scheme are described in Section D. For all our experiments, unless specified otherwise, we fix the input dimension to $n = 16$, hidden dimension of the teacher to $h = 32$, and output dimension to $o = 8$. We define the distribution over inputs $x$ to be the multivariate uniform distribution over the hypercube $[-1, 1]^n$.

Targets are generated by passing the input $x$ through the generated network. To make sure the distribution of the outputs is not too dissimilar across tasks, we normalize each output neuron to have unit variance and zero mean by numerically estimating the first and second moment of the distribution over random inputs. The tasks are sampled following the same procedure as in the discrete task distribution description in Section C.1.1. That is, a finite set of binary vectors of size $M$ fully specifies the task distribution.

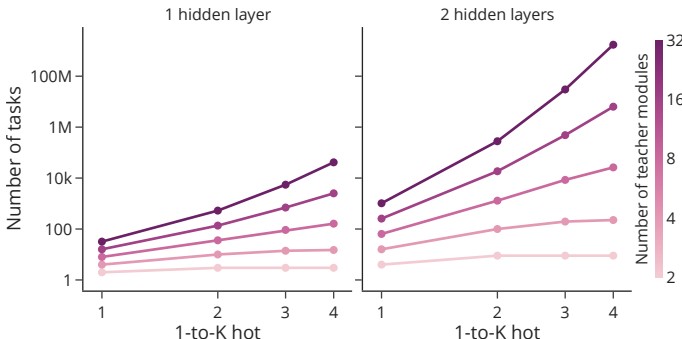

Figure A11: Number of possible tasks in the hyperteacher as a function of the number of teacher modules $M$, the number of sparse combinations $K$ and number of hidden layers $L$ is given by $\sum_k^K \binom{M}{k}^{L-1}$ growing exponentially in $M$.

### C.2.2 TRAINING AND EVALUATION

We train the models following the bilevel cross-validation objective introduced in Section 2. Specifically, we follow the algorithm outlined in 2. The loss function for both inner and outer loop is the KL divergence between the log-softmax of the target $y$ and the log-softmax of the prediction of the network.

For each task, we sample i.i.d 256 input-output pairs for each of the support and query set. The evaluation follows the same procedure and inner optimization, except for the fact that it involves tasks that are combination of modules that have not been seen during training. We report the outer KL divergence averaged over $B_{\text{outer}}$ tasks, as well as the accuracy, which measures the average over tasks of fraction of input for which the index of the largest value of the prediction matches that of the largest value of the target.

### C.2.3 EXPERIMENTS

Here we provide additional details for the experiments presented in Figure 3.

**Compositional vs non-compositional support.** We perform this experiment for different values of $K$. Given $K$, we compute the set of binary masks with up to $K$ non-0 entries, and randomly select three-quarters of them and use the obtained list for generating training tasks. We make sure that the obtained list has compositional support, that is, all modules appear at least once. The remaining quarter is used as the OOD task for evaluation. For non-compositional support, we again enumerate all binary masks with up to $K$ non-0 entries, but split it into 2 sets such that for training, only masks where the last module is inactive are used, while for the evaluation, masks which always have the last module activated, are used to generate the tasks.

**Connected vs disconnected support.** For these experiments, we consider $M = 8$.

For connected task support, we used the task distribution defined by the set of masks $\{e_i\}_{1 \leq i \leq M} \cup \{e_i + e_{i+1 \mod M}\}_{1 \leq i \leq M} \cup \{e_i + e_{i+2 \mod M}\}_{1 \leq i \leq M}$ where $(e_i)_{1 \leq i \leq M}$ represent the canonical basis of $\mathbb{R}^M$. For disconnected task support, we used the task distribution defined by the set of masks $\{e_i\}_{1 \leq i \leq M} \cup \{e_i + e_j\}_{i,j \in [1,...M/2], i \neq j} \cup \{e_i + e_j\}_{i,j \in [M/2+1,...M], i \neq j}$. These tasks were chosen such that the number of tasks for the connected and disconnected experiment are comparable. For the OOD task distribution used for evaluation, we used all binary masks with $K = 2$ that have been neither used by the connected nor disconnected task distribution.

**Other experiments.** For the other experiments, we generate tasks following the same split as for the compositional experiment. For the overparameterization experiment, we further set the teacher hidden layer width to $h = 16$.

Table A10: Hyperparameter grid-search space for the hyperteacher. Best found parameters are marked in bold.

| Hyperparameter | Nonlinear hypernet. | Linear hypernet. | ANIL | MAML |
|---|---|---|---|---|
| lr_outer | $\{0.001, \mathbf{0.003}\}$ | $\{\mathbf{0.001}, 0.003\}$ | $\{\mathbf{0.001}, 0.003\}$ | $\{\mathbf{0.001}, 0.003\}$ |
| lr_inner | $\{0.1, \mathbf{0.3}\}$ | $\{0.1, \mathbf{0.3}\}$ | $\{0.01, \mathbf{0.03}\}$ | $\{\mathbf{0.01}, 0.03\}$ |
| grad_clip | $\{\mathbf{1}, 2\}$ | $\{\mathbf{1}, 2\}$ | $\{\mathbf{1}, 2\}$ | $\{1, \mathbf{2}\}$ |
| weight_decay | $\{\mathbf{0.01}, 0.0001\}$ | $\{\mathbf{0.01}, 0.0001\}$ | $\{0.01, \mathbf{0.0001}\}$ | $\{0.01, \mathbf{0.0001}\}$ |

### C.2.4 HYPERPARAMETERS

For all hyperteacher experiments, we set $B_{\text{outer}} = 128$, $N_{\text{outer}} = 200000$ and use full-batch inner optimization. For all models, we optimize both the inner and outer loop with the AdamW optimizer. Unless specified otherwise, we set the following architectural hyperparameters, ensuring the total number of parameters across models is comparable.

- Linear Hypernetwork: base network hidden layer $L = 3$, hidden units $\hat{h} = 128$. Number of modules $\hat{M} = 4M$

- Nonlinear Hypernetwork: base network hidden layer $L = 3$, hidden units $\hat{h} = 128$. Number of modules $\hat{M} = 4M$

- MAML: network hidden layer $L = 3$, hidden units $\hat{h} = 368$

- ANIL: network hidden layer $L = 3$, hidden units $\hat{h} = 512$

We fix the inner loop steps $N_{\text{inner}}$ to 10 for all models, except for ANIL where $N_{\text{inner}} = 100$. For all models, we tune the hyperparameters independently for the inner learning rate, the outer learning rate, outer weight decay as well as the outer gradient clipping on the range specified in Table A10. The hyperparameters are tuned to optimize for the compositional generalization loss on the compositional support experiment detailed above. The same set of hyperparameters were then used for all other experiments.

### C.3 COMPOSITIONAL PREFERENCES

### C.3.1 DATA GENERATION

**Environment.** We consider a grid world of size $5 \times 5$ as seen in Figure 4A. There are 5 actions: move up, right, down, left, and terminate. Black cells represent walls. The agent deterministically moves from one cell to another following the direction if the grid is wall(or border)-free, and otherwise remains in place. The episode terminates when the terminate action is taken. Finally, there are 4 objects placed on wall-free cells of the grid. Each object has exactly one of the 8 colors. Each object remains static throughout the episode, but disappears when the agent steps on it, potentially producing a reward in a task-specific manner, which will be explained below. The state space is then the grid, which shows the positions of the walls, the objects, their respective feature, and the position of the agent, as in Figure 4A.

**Task.** A task is defined by a task latent variable $\boldsymbol{z}$, of dimension $M = 8$. Such a variable defines the reward function of the environment in the following way: first, a preference vector is computed by linearly combining a set of $M$ pre-computed preference template vectors of dimension 8, using the task latent variable as the coefficient. The resulting preference vector specifies a mapping from the 8 colors to a preference value. During the task, when the agent steps on an object, it gets a reward equal to the preference value of that color.

For each task, several instances of the environment are then created. While the wall configuration remains fixed, for every new instance of the environment, we sample the position of all objects uniformly from all wall-free cells (making sure there is no overlap), as well as the initial position of the agent. For each instance, we compute the optimal action-value function with discount factor 0.9 and time horizon 8. We then let an agent behave greedily following the obtained action value function (forcing the agent to take the terminate action when no positively rewarding objects are left

in the grid), and build the task dataset $\mathcal{D}(\boldsymbol{z}) = ((x_i, y_i))_i$ by collecting the observations $\boldsymbol{x}$ along the trajectory, and the corresponding action value vector $\boldsymbol{y}$, over all instances. Finally, tasks are sampled following the same procedure as in the continuous task distribution description in Section C.1.1. A finite set of binary masks, each of size $M$, thus fully specifies the task distribution.

### C.3.2 TRAINING AND EVALUATION.

We train the models following the bilevel cross-validation objective introduced in Section 2. Specifically, we follow the algorithm 2. The loss function for both inner and outer loop is the Mean Squared Error (MSE) loss between the action values $\boldsymbol{y}$ and the prediction of the network.

For each task, we instantiate 32 different instances, 16 of which being used for the support set and the remaining 16 for the query set, and create the dataset following the procedure outlined in the previous section. The evaluation follows the same procedure and inner optimization, except for the fact that it involves tasks that are combinations of modules that have not been seen during training. We report the outer MSE loss averaged over $B_{\text{outer}}$ tasks.

### C.3.3 EXPERIMENTS

We detail the experiments corresponding to Figure 4B and C.

**Connected vs disconnected support.** For connected task support, we used the task distribution defined by the set of mask $\{e_i\}_{1 \leq i \leq M} \cup \{e_i + e_{i+1 \mod M}\}_{1 \leq i \leq M} \cup \{e_i + e_{i+2 \mod M}\}_{1 \leq i \leq M}$ where $(e_i)_{1 \leq i \leq M}$ represent the canonical basis of $\mathbb{R}^M$. For disconnected task support, we used the task distribution defined by the set of mask $\{e_i\}_{1 \leq i \leq M} \cup \{e_i + e_j\}_{i,j \in [1,...M/2], i \neq j} \cup \{e_i + e_j\}_{i,j \in [M/2+1,...M], i \neq j}$. These tasks were chosen such that the number of tasks for the connected and disconnected experiment are comparable. For the OOD task distribution used for evaluation, we used all binary masks with $K = 2$ that have been neither used by the connected nor disconnected task distribution.

**Compositional vs non-compositional support.** For compositional support, we compute the set of binary masks with up to 3 non-0 entries, and randomly select three-quarter of them and use the obtained list for generating training tasks. We make sure that the obtained list contain a compositional support. The remaining quarter is used as the OOD task for evaluation. For non-compositional support, we enumerate all binary masks with up to 3 non-0 entries, and split it into 2 sets following the value of the last entry of the mask. For training, we use the set of mask where the last module is inactive, while for the evaluation, we use the set of mask which always has the last module activated, to generate the tasks. Again, these settings were chosen such that the number of tasks for the compositional and non-compositional experiment are comparable.

### C.3.4 HYPERPARAMETERS

For all experiments, we set $B_{\text{outer}} = 128$, full-batch inner optimization, $N_{\text{outer}} = 100000$. For all models, we optimize the inner loop with AdamW optimizer with the default weight decay of 0.00001, and the outer loop with AdamW optimizer with tuned weight decay. Unless specified otherwise, we set the following architectural hyperparameters to ensure the total number of parameters across models is comparable.

- Linear Hypernetwork: base network hidden layer $L = 3$, hidden units $\hat{h} = 64$. Number of modules $\hat{M} = 32$
- Nonlinear Hypernetwork: base network hidden layer $L = 2$, hidden units $\hat{h} = 64$. Number of modules $\hat{M} = 32$
- MAML: network hidden layer $L = 3$, hidden units $\hat{h} = 368$
- ANIL: network hidden layer $L = 4$, hidden units $\hat{h} = 512$

We fix the inner loop steps $N_{\text{inner}}$ to 10 for all models, except for ANIL where $N_{\text{inner}} = 100$. For all models, we tune the hyperparameters independently for the inner learning rate, the outer learning

Table A11: Hyperparameter grid-search space for the compositional preference environment. Best found parameters are marked in bold.

| Hyperparameter | Nonlinear hypernet. | Linear hypernet. | ANIL | MAML |
|---|---|---|---|---|
| lr_outer | $\{\mathbf{0.0003}, 0.001\}$ | $\{\mathbf{0.0003}, 0.001\}$ | $\{\mathbf{0.0003}, 0.001\}$ | $\{0.0003, \mathbf{0.001}\}$ |
| lr_inner | $\{\mathbf{0.1}, 0.3\}$ | $\{\mathbf{0.1}, 0.3\}$ | $\{\mathbf{0.01}, 0.03\}$ | $\{\mathbf{0.01}, 0.03\}$ |
| grad_clip | $\{1, \mathbf{2}\}$ | $\{1, 2\}$ | $\{1, \mathbf{2}\}$ | $\{1, \mathbf{2}\}$ |

rate as well as the outer gradient clipping from the range specified in table A11. The hyperparameters are tuned to optimize for the compositional generalization loss on the compositional support experiment detailed above. The same set of hyperparameters were then used for all other experiments.

## C.4 COMPOSITIONAL GOALS

### C.4.1 DATA GENERATION

For the compositional goals environment we consider mazes of size $11 \times 11$ with the same movement dynamics as described above in the compositional preference environment. In addition to moving up, down, left and right, the agent can choose one of the 'object interaction' actions. Each environment has 5 randomly placed objects, one of which is the target object. The agent is placed randomly and has to reach the target object followed by performing the target interaction to obtain a reward. Each task is defined by a goal vector that specifies which of the 5 maze configurations is used, which of the 5 object types is assigned as the target object, which of the 2 possible target interactions is the correct one and finally in which of the 4 quadrants of the maze the target object will be located (c.f. Figure 4D.)

Goals are compositional in the sense that any of these factors can be arbitrarily combined leaving a total of $5 \cdot 5 \cdot 2 \cdot 4 = 200$ possible goals. Given a goal, we first sample a new random position of the agent and all objects, compute the optimal behavior policy and use it to draw 1 sample demonstration of the optimal policy for the support set. We then sample new random configuration of the objects and the agent given the same goal for the query set.

In the setting of compositional support, we randomly hold-out 25% of the possible goals ensuring that every goal factor appears at least in one of the goals of the remaining 75% of goals used for training.

To produce a training distribution with non-compositional support, we consistently hold out one of the goal quadrants from the set of goals and use all goals that contain the held-out goal quadrant for the OOD set.

### C.4.2 TRAINING AND EVALUATION

We train the models following the bilevel cross-validation objective introduced in Section 2. Specifically, we follow the algorithm outlined in Algorithm 2. The loss function is the cross-entropy loss between the optimal action and the prediction of the network.

For each task, we sample a goal, instantiate two environments consistent with the goal and draw samples containing the optimal trajectory from the agent location to the goal for both. We use the demonstration of one instantiation of the environment for the support set and the second one for the query set.

For all experiments, we use full batch optimization in the inner loop, and fix the outer batch size to 128, i.e. $B_{\text{outer}} = 128$.

The evaluation follows the same procedure as the inner optimization, except for the fact that it involves tasks based on goals that have not been seen during training. We report the outer loss averaged over 1024 tasks.

Table A12: Hyperparameter grid-search space for the compositional goal environment. Best found parameters are marked in bold.

| Hyperparameter | Nonlinear hypernet. | Linear hypernet. | ANIL | MAML |
|---|---|---|---|---|
| lr_outer | $\{\textbf{0.001}, 0.003\}$ | $\{0.001, \textbf{0.003}\}$ | $\{\textbf{0.001}, 0.003\}$ | $\{\textbf{0.001}, 0.003\}$ |
| lr_inner | $\{\textbf{0.1}, 0.3\}$ | $\{0.1, \textbf{0.3}\}$ | $\{\textbf{0.01}, 0.03\}$ | $\{\textbf{0.01}, 0.03\}$ |
| grad_clip | $\{1, \textbf{2}\}$ | $\{1, \textbf{2}\}$ | $\{1, \textbf{2}\}$ | $\{\textbf{1}, 2\}$ |

### C.4.3 HYPERPARAMETERS

For all experiments, we set $B_{\text{outer}} = 128$, $B_{\text{inner}} = 256$, $N_{\text{outer}} = 200000$. We optimize both the inner and outer loop with the AdamW optimizer. For the outer loop we use a weight decay of 0.001 for the inner loop we use a weight decay of 0.0001. Unless specified otherwise, we set the following architectural hyperparameters, ensuring the total number of parameters across models is approximately comparable.

- Linear Hypernetwork: base network hidden layer $L = 2$, hidden units $\hat{h} = 32$. Number of modules $\hat{M} = 8$

- Nonlinear Hypernetwork: base network hidden layer $L = 2$, hidden units $\hat{h} = 32$. Number of modules $\hat{M} = 8$

- MAML: network hidden layer $L = 2$, hidden units $\hat{h} = 384$.

- ANIL: network hidden layer $L = 2$, hidden units $\hat{h} = 512$.

We fix the inner loop steps $N_{\text{inner}}$ to 10 for all models, except for ANIL where $N_{\text{inner}} = 100$. For all models, we tune the hyperparameters independently for the inner learning rate, the outer learning rate as well as the outer gradient clipping from the range specified in Table A12. The hyperparameters are tuned to optimize for the compositional generalization loss on the compositional support experiment detailed above, for all models. The same set of hyperparameters were then used for all other experiments.

## D META-LEARNING MODELS

In this section we provide details of the models we train in meta-learning experiments. We use the same architectures across experiments and settings.

We keep the notation of $\theta$ and $\phi$ to respectively denote meta-parameters and fast-parameters, as used in Section 2 and Algorithm 2.

### D.1 MAML

#### D.1.1 PARAMETERIZATION

We consider a standard ReLU MLP with $L$ hidden layers of width $h$, with bias. During meta learning, all parameters of the network are adapted and are thus part of the fast parameter $\phi$. The only task-shared parameter $\theta$ is thus the value at which we initialize the fast parameters at the beginning of each task, i.e. $\theta = \phi_0$.

#### D.1.2 INITIALIZATION

We initialize $\phi_0$ such that parameters corresponding to weights are initialized following the truncated normal distribution with mean 0 and standard deviation $\frac{1}{\sqrt{H}}$ where $H$ is the size of the input to the layer. Parameters corresponding to biases are initialized at 0.

## D.2 ANIL

### D.2.1 PARAMETERIZATION

We consider a standard ReLU MLP with $L$ hidden layers of width $h$, with bias.

Here, during meta learning, only the readout layer of the network is part of the fast parameters $\phi$. The parameters for the rest of the network, as well as the value $\phi_0$ at which we initialize the fast parameters at the beginning of each task are part of the task-shared parameters $\theta$.

### D.2.2 INITIALIZATION

We initialize $\theta$ and $\phi$ such that parameters corresponding to weights are initialized following the truncated normal distribution with mean $0$ and standard deviation $\frac{1}{\sqrt{H}}$ where $H$ is the size of the input to the layer. Parameters corresponding to biases are initialized at $0$.

## D.3 LINEAR HYPERNETWORK

### D.3.1 PARAMETERIZATION

We consider a base network consisting in a ReLU MLP, with $L$ hidden layers of width $h$. Crucially, we use the NTK parameterization of the MLP, where at each layer, the pre-activation units are rescaled by $\frac{1}{\sqrt{H}}$ where $H$ is the dimension of the input to the layer. As such, if at initialization the hypernetwork outputs weights that are centered and unit variance, the forward pass would be approximately variance preserving.

The weights are generated by the linear hypernetwork in the following way:

- First, given an embedding $z$, the linear hypernetwork normalizes the embedding to the unit norm.
- Then, the normalized embedding is linearly multiplied by the parameter of the hypernetwork $\Theta$, to generate the flattened weights
- Finally, the flattened weight are reshaped and plugged into the base network

The biases are not generated by the hypernetwork.

Crucially, we use 3 hypernetworks to generate respectively the first layer weight, all hidden layer weights, and the last layer weight. There are therefore 3 embeddings in total, one for each hypernetwork.

When training the model in the meta-learning setting, we use as fast parameters $\phi$ the set of embedding vectors as well as the biases in the base network. The meta parameter $\theta$ are the hypernetwork parameters, as well as the initialization value for the fast parameters.

### D.3.2 INITIALIZATION

The hypernetwork weights are initialized following the truncated normal distribution with mean $0$ and standard deviation $\frac{1}{\sqrt{M}}$ where $M$ is the embedding vector dimension.

The meta-parameters corresponding to the initialization values of the embeddings are initialized following a uniform and unit variance distribution, and those corresponding to biases are initialized to $0$.

## D.4 NONLINEAR HYPERNETWORK

### D.4.1 PARAMETERIZATION

The parameterization of the nonlinear hypernetwork closely follows that of the linear counterpart (see Section D.3.1. Instead of using a linear layer to produce the weights of the base network given the fast parameters $\phi$, the fast parameters are first nonlinearly transformed in a MLP.

Specifically, the hypernetwork is a 4-layer MLP with the ELU activation function applied at every hidden layer. Layer normalization is applied before each activation function, as well as to the input.

When training the model in the meta-learning setting, we use as fast parameters $\phi$ the set of embedding vectors as well as the biases in the base network. The meta-parameters $\theta$ are the hypernetwork parameters, as well as the initialization values for the fast parameters.

### D.4.2 INITIALIZATION

The hypernetwork parameters are initialized such that all biases are set to 0 and MLP layer weights are initialized following the truncated normal distribution with mean 0 and standard deviation $\frac{1}{\sqrt{H}}$ where $H$ is the size of the input to the layer.

The meta-parameter corresponding to the initialization value of the embeddings are initialized following a uniform and unit variance distribution, and those corresponding to biases are initialized to 0.

## E    EXTENDED RELATED WORK

We complement the related work discussed in the main text with a more comprehensive review in the following.

The debate on the aptitude of neural networks for compositionality dates back to the first wave of connectionist models (Rumelhart & McClelland, 1986; Fodor & Pylyshyn, 1988; Smolensky, 1991; Hadley, 1994; Phillips, 1995) fueled by the proposition that combinatorial structures are a key property of human cognitive abilities. The widespread success of deep learning has lead many efforts to evaluate the capacity of deep networks for compositional generalization.

A number of benchmarks have been designed to specifically investigate the ability for compositional generalization primarily in the context of language (Bastings et al., 2018; Lake & Baroni, 2018; Kim & Linzen, 2020; Ruis et al., 2020; Hupkes et al., 2020; Keysers et al., 2020; Dankers et al., 2022), visual question answering (Johnson et al., 2017; Bahdanau et al., 2020), visual scenes (Xu et al., 2022) and reinforcement learning (Barreto et al., 2020; Zhao et al., 2022). Commonly, these benchmarks demonstrate shortcomings of current deep networks at achieving compositional generalization from recurrent neural networks (Lake & Baroni, 2018; Loula et al., 2018), to convolutional networks (Dessì & Baroni, 2019) and transformers (Keysers et al., 2020).

A large number of architectures have been designed with compositionality in mind. A common theme among them is to attempt to decompose skills or reusable atoms of knowledge, across tasks in order to recombine them in novel ways for new tasks (Alet et al., 2018; Ponti et al., 2022; Kingetsu et al., 2021). Compositional plan vectors (Devin et al., 2020) learn composable embeddings to encode different tasks and condition a policy. Similarly, Frady et al. (2023) use composable vector symbols to represent compositional visual scenes given latent supervision. To impart a strong bias for compositionality, some works aim to directly learn symbolic expressions (Liu et al., 2020; Vankov & Bowers, 2020). Similar to our setup, meta seq2seq (Lake, 2019) conditions a sequence model on demonstrations to facilitate compositional generalization. Building modular systems that exhibit compositional generalization is also a main goal of graph neural networks (Battaglia et al., 2018). In this context compositional structure is typically built into the system while here we attempt to discover this structure from unstructured observations. Complementary to this, Ghazi et al. (2019) have proposed a framework to make the function implemented by individual modules interpretable.

With the advent of pretrained large language models, compositional abilities have seemingly come into reach (Zhou et al., 2023; Orhan, 2022; Furrer et al., 2021; Csordás et al., 2021). Many specialized architectures that show compositional behavior on one of the aforementioned benchmarks (Russin et al., 2019; Li et al., 2019; Gordon et al., 2020; Andreas, 2020; Liu et al., 2020; Lake, 2019; Nye et al., 2020; Kaiser & Sutskever, 2016) are outperformed by pretrained large language models (Furrer et al., 2021). However, it remains an open question whether the ability for compositional generalization extends beyond the training distribution (Srivastava et al., 2023; Press et al., 2023; Dziri et al., 2023).

# F ADDITIONAL DETAILS

## F.1 COMPUTE RESOURCES

We used Linux workstations with Nvidia RTX 2080 and Nvidia RTX 3090 GPUs for development and conducted hyperparameter searches and experiments using 5 TPUv2-8, 5 TPUv3-8 and 1 Linux server with 8 Nvidia RTX 3090 GPUs over the course of 9 months. In total, we spent an estimated amount of 6 GPU months.

## F.2 SOFTWARE AND LIBRARIES

For the results produced in this paper we relied on free and open-source software. We implemented our experiments in Python using JAX (Bradbury et al., 2018, Apache License 2.0) and the Deepmind Jax Ecosystem (Babuschkin et al., 2020, Apache License 2.0). For experiment tracking we used wandb (Biewald, 2020, MIT license) and for the generation of plots we used plotly (Inc, 2015, MIT license).

