# OpenReview forum: "Discovering modular solutions that generalize compositionally"
_ICLR.cc/2024/Conference — ICLR 2024 poster_

### Official Review · Reviewer_KGfv · 2023-10-27

**Soundness:** 3 good
**Presentation:** 2 fair
**Contribution:** 3 good
**Rating:** 8
**Confidence:** 3

**Summary:**

The authors provide a theoretical analysis of the compositional generalization capabilities of linear hypernetworks under a teacher-student setup where the teacher network is the ground-truth data-generating model, and is itself a linear hypernetwork. The key result is that under 3 assumptions (compositional support, connected support, and no over-parameterization), the student can provably _identify_ the modules of the teacher. This identification is further demonstrated to be a necessary and sufficient condition for compositional generalization. The authors provide a series of experiments to validate their theoretical findings and test how sensitive the empirical results are to various violations of the assumptions.

######## Post-rebuttal update ########

Given the authors' response and updated manuscript, I am updating my score to an 8 (this 8 should be considered closer to a 7, but 7 is not an option).

**Strengths:**

######## Strengths ########
- The problem of compositional generalization is of great interest to the ML/AI communities, and there are very few theoretical treatments of the problem
- The combination of theoretical results and empirical evidence that tests beyond the bounds of the theory is well balanced

**Weaknesses:**

######## Weaknesses ########
- The experimental setting is unclear, especially the sequential decision-making portion, which makes it hard to assess its impact
- It is unclear whether the three assumptions (compositional support, connected support, and no over-parameterization) are necessary conditions or just sufficient
- The choice of linear hypernetworks as the base modular model somewhat limits the intepretability of the compositionality

######## Recommendation ########

I perceive this work as borderline acceptable. The balance of strengths and weaknesses, I believe, works out in favor of the strengths slightly. That being said, I encourage the authors to address my comments below to improve their manuscript.

######## Arguments ########

There are very few theoretical treatments of compositionality in the context of neural nets in the literature. In particular, I am only aware of [1] (which the authors actually fail to mention). The two works study very distinct settings, and so I do believe that this submission proposes a novel piece of work. Moreover the authors execute a fairly comprehensive experimental evaluation of their ideas, which is rare for theoretical works. To me, this merits acceptance of the work.

However, there are a few shortcomings that the authors should address.
- Clarity of the experimental setting
    - Sec 4.1
        - How is compositional generalization measured? Does the agent train the task-specific coefficients on the new task without training the basis vectors? This isn't really described anywhere.
        - "certain overparameterization is necessary for meta-learning to succeed" -- what does this mean? Doesn't this contradict the theory? Why is it necessary? Haven't all previous results used the "right" number of modules? Also, all results in Fig. 3F seem to show that more module overparameterization is consistently better. It isn't clear how this connects to the theory or the remaining results and claims.
    - Sec 4.2
        - Am I reading right that the agents are trained on _tens of millions_ of tasks? Though it seems that convergence occurs much earlier on. Could the authors zoom in to the left portion of 4E?
        - The training setting for 4.2 is very unclear. Is there a teacher? If not, what is the training process? Why is "accuracy" used as the metric in a sequential decision making problem? How many total tasks are there (training on 10M out of 1B maybe isn't that bad, but 10M out of 30M isn't super encouraging)? Does accuracy correlate with actual performance on the task (i.e. does the agent actually compositionally generalize to new RL tasks)? How are we measuring "OOD generalization" in this setting?
- Necessary or sufficient conditions
    - The authors' aim, as stated in the introduction, is to understand "what properties of the data generating process allow to discover modular solutions enabling compositional generalization"
    - However, the phrasing of the first lines of page 4 suggests that assumpions i-iii are _sufficient_ conditions, but not _necessary_ conditions for identification
    - This means that what we really get is an understandong of _some_ of the properties that allow to discover modular solutions that compositionally generalize
    - The authors do take steps to understand empirically what the "cost" is of violating those assumptions, and the assumptions themselves are fairly intuitive, so that's a positive
- Interpretability of linear hypernetworks
    - Other modular solutions (such as the neural module networks of [2]) have very intuitive explanations of what each module does _and how that composes with other modules_
    - Linear parameter combinations within a hypernet don't seem to have such a straightforward intuition
    - The authors' attempt at understanding what kinds of composition the model can learn is in the experiments of Sec 4.2, but since those are not clearly explained, it's hard to get any intuition from them. I encourage the authors to include an intuitive description of what the compositionality is in those tasks and how linear hypernets should capture that compositionality.


[1] Ghazi et al., "Recursive Sketches for Modular Deep Learning." ICML, 2019.

[2] Andreas et al., "Neural Module Networks." CVPR, 2016

**Questions:**

######## Additional feedback ########

The following points are provided as feedback to hopefully help better shape the submitted manuscript, but did not impact my recommendation in a major way.

Abstract
- Throughout the abstract and much of the text, it's unclear what types of problems the authors are tackling. There's a mention of "demonstrations" and "action-value functions", which somewhat hints at a solution geared toward RL. Much later, it becomes clear that a "demonstration" is the output of the teacher network, and that action-value prediction is just one application of the linear hypernets.

Sec 3
- Up to this point, it was still unclear to me what the authors meant by identification of modules
    - Is it that the student can determine which modules to use, or that it can find the right set of modules?
    - Is this in a setting where the student knows the latent codes?
    - These are later clarified, but it might be worth doing so earlier on

Sec 5
- It's odd that the authors combined a related work section with the discussion, but I think it works okay.

Typos/style/grammar
- Intro, page 1 first paragraph: series of tasks --> set of tasks [a series is sequential]

---

> ### Author Response · Authors · 2023-11-16
> **Part 1**
>
> Thank you for your comprehensive review and for the useful feedback that we incorporated in the revised version of our manuscript.
> We are grateful that you appreciated the combination of theoretical results and empirical evidence despite your concerns with the clarity of the exposition of the experimental setting.
> Following your suggestions, we have made substantial efforts to improve the presentation of the latter and address your individual questions point by point below.
> We hope this strengthens your assessment for acceptance of our work.
>
>
> > The experimental setting is unclear, especially the sequential decision-making portion, which makes it hard to assess its impact
>
> We have substantially reworked the experimental section to clarify the experimental setting in our updated version of the manuscript.
> In addition, we provide detailed descriptions of the experimental setting in Appendix C.
>
>
> > It is unclear whether the three assumptions (compositional support, connected support, and no over-parameterization) are necessary conditions or just sufficient
>
> Thank you for raising this issue. We have now added a sentence immediately after Theorem 1 clarifying this point. In short, under the same conditions with no additional assumptions, the three conditions of connected support, compositional support as well as no over-parameterization are not only sufficient but necessary conditions for the implication in Theorem 1 to hold. We illustrate this by constructing counterexamples to the implication when some of the three conditions are violated, which we detail in Appendix A2.
>
>
> > The choice of linear hypernetworks as the base modular model somewhat limits the intepretability of the compositionality
>
> This is indeed an interesting point to consider as it is not clear if parameter module composition as done by hypernetworks generally leads to interpretable modules.
> One part of the problem is certainly that interpreting individual modules boils down to interpreting the neural network they individually parameterize which is an open research problem.
> In general, it is not clear if the compositional structure discovered by hypernetworks reflects the abstract compositional structure of the task (e.g. compositional goals or compositional preferences).
> In the case of the modular teacher-student and the extended hyperteacher setting, our linear identification result shows, that it is possible to decode the ground-truth modules given the learned embeddings.
> We show that this is indeed possible in practice in Figures 3C and D.
> Future work is needed to understand under what conditions parameter modules reflect the potentially more abstract compositional structure in more complex settings.
>
>
> > There are very few theoretical treatments of compositionality in the context of neural nets in the literature. In particular, I am only aware of [1] (which the authors actually fail to mention).
>
> Thank you for pointing us to this reference which we indeed missed.
> We have added it to the updated manuscript.
>
> ### Section 4.1
>
> > How is compositional generalization measured? Does the agent train the task-specific coefficients on the new task without training the basis vectors
>
> Yes, this is correct!
> We were indeed missing a paragraph on the evaluation during test time which we have now added to Section 2.
> In short, we use a support set to allow the model to infer the task-specific coefficients while keeping the basis vectors fixed and measure performance on an independent query set.
>
> > "certain overparameterization is necessary for meta-learning to succeed" -- what does this mean? Doesn't this contradict the theory? Why is it necessary? Haven't all previous results used the "right" number of modules? Also, all results in Fig. 3F seem to show that more module overparameterization is consistently better. It isn't clear how this connects to the theory or the remaining results and claims.
>
> Thank you for raising this important point.
> We have now added a sentence clarifying this point in Section 3.3.
> In short, while the compositional generalization *when achieving 0 training loss* can only be guaranteed for non over-parameterized students, optimization, i.e. achieving 0 loss, is itself difficult for non over-parameterized models.
> Fortunately, the empirical results suggest that gradient descent seems to find the generalizing solutions even when the student is slightly over-parameterized, up to some extent.
> Interestingly, the other conditions, and in particular ensuring connected support, seem to be much more important.

---

> > ### Author Response · Authors · 2023-11-16
> > **Part 2**
> >
> > ### Section 4.2
> > > Am I reading right that the agents are trained on tens of millions of tasks? Though it seems that convergence occurs much earlier on. Could the authors zoom in to the left portion of 4E?
> > > How many total tasks are there (training on 10M out of 1B maybe isn't that bad, but 10M out of 30M isn't super encouraging)?
> >
> > Thank you for bringing this to our attention.
> > The Figure label was indeed misleading since most of these "tasks" share the same latent structure.
> > In particular in the compositional grid environment, there are 5 different target objects, 5 different maze layouts, 4 different goal quadrants and 2 different target interactions which results in 200 total tasks out of which only 150 are used for training and the remaining 50 are used for OOD evaluation.
> > During training the same compositional goal (out of the 150) is shown multiple times and the X label is more accurately described as "Training iterations".
> > We have changed this in our updated version of the manuscript.
> >
> > While the modular architectures outperform the monolithic architectures already after a fraction of the total training steps shown, we wanted to ensure that this is not purely due to ease of optimization and holds when allowing for longer training.
> >
> > > The training setting for 4.2 is very unclear. Is there a teacher? If not, what is the training process?
> >
> > Thank you for raising this issue.
> > We have added the following description of the task to the updated version of the manuscript:
> >
> > We now consider a setting where an agent has compositional preferences shared across tasks but the functional mapping to its action-value function for each specific task is unknown.
> > In particular, we study the compositional preference grid world inspired by [1] where an agent can obtain rewards by collecting objects with different colors.
> > For each task, up to $K=3$ out of $M=8$ preference modules are sampled and summed to obtain a hidden preference vector which specifies the reward the agent obtains when moving to an object of a specific color (see Figure 4A).
> > We test the ability for compositional generalization by presenting the agent with tasks of unseen combinations of preference modules.
> > The agent then has to infer the current preferences from observing optimal trajectories and is evaluated to predict the action-value function for a new environment configuration where agent and object locations are resampled.
> > For more details on the task construction see Appendix C3.
> >
> > > Why is "accuracy" used as the metric in a sequential decision making problem?
> >
> > We measure the accuracy with respect to the ground truth optimal policy.
> > This is due to our particular meta-learning setup where the model has to learn a composable behavior policy from demonstrations of optimal trajectories.
> > We clarified this point in the updated version of the manuscript.
> >
> > > Does accuracy correlate with actual performance on the task (i.e. does the agent actually compositionally generalize to new RL tasks)?
> >
> > Since the accuracy is measured with respect to the ground truth optimal policy, it is a conservative measure of actual performance on the task.
> > That is there could be suboptimal policies that obtain similar reward to the optimal policy but would still result in a lower accuracy.
> >
> > > How are we measuring "OOD generalization" in this setting?
> >
> > Thank you for asking for this clarification, which we have added to the updated version of the manuscript.
> > We measure compositional generalization in the two grid world environments by withholding 25\% of the possible compositions.
> > In the compositional preferences environment this amounts to withholding certain combinations of the preference modules, in the compositional goals environment this amounts to withholding certain goal compositions.
> > For example the agent might during training never encounter tasks with the goal of "performing interaction 2 on the purple object in goal quadrant 3 in the maze layout 1" since they are used for OOD evaluation.
> >
> > > Necessary or sufficient conditions: The authors' aim, as stated in the introduction, is to understand "what properties of the data generating process allow to discover modular solutions enabling compositional generalization". However, the phrasing of the first lines of page 4 suggests that assumpions i-iii are sufficient conditions, but not necessary conditions for identification. This means that what we really get is an understandong of some of the properties that allow to discover modular solutions that compositionally generalize.
> >
> > Thank you for pointing out that this part was not well communicated in the initial version of the manuscript.
> > Please see our response to your related point above.

---

> ### Author Response · Authors · 2023-11-16
> **Part 3**
>
> > Interpretability of linear hypernetworks:  Other modular solutions (such as the neural module networks of [2]) have very intuitive explanations of what each module does and how that composes with other modules. Linear parameter combinations within a hypernet don't seem to have such a straightforward intuition. The authors' attempt at understanding what kinds of composition the model can learn is in the experiments of Sec 4.2, but since those are not clearly explained, it's hard to get any intuition from them. I encourage the authors to include an intuitive description of what the compositionality is in those tasks and how linear hypernets should capture that compositionality.
>
> Thank you for this suggestion.
> We have tried to better motivate and explain the two environments in this light in our updated version of the manuscript.
>
> > Throughout the abstract and much of the text, it's unclear what types of problems the authors are tackling. There's a mention of "demonstrations" and "action-value functions", which somewhat hints at a solution geared toward RL. Much later, it becomes clear that a "demonstration" is the output of the teacher network, and that action-value prediction is just one application of the linear hypernets.
> > Up to this point [Section 3], it was still unclear to me what the authors meant by identification of modules.  Is it that the student can determine which modules to use, or that it can find the right set of modules? Is this in a setting where the student knows the latent codes? These are later clarified, but it might be worth doing so earlier on
>
> Thank you for raising these issues.
> We have made substantial efforts to improve the presentation of the paper and attempted to clarify these points earlier in the text.
>
>
> > It's odd that the authors combined a related work section with the discussion, but I think it works okay.
>
> Indeed as you point out, we have treated related work as part of our discussion section. We clarify this now by changing the section title to "Related work and Discussion". In addition, we added an "Extended related work" section to Appendix~E.
>
>
> [1] Barreto, André, et al. "Fast reinforcement learning with generalized policy updates." Proceedings of the National Academy of Sciences 117.48 (2020): 30079-30087.

---

> > ### Comment · Reviewer_KGfv · 2023-11-18
> > **Thank you for the thorough response**
> >
> > I thank the authors for their thorough response. In particular, their answers here (and their updates to the draft) have clarified the main concerns I had with the presentation.
> >
> > I still have one question about the RL evaluation. The authors claim that "Since the accuracy is measured with respect to the ground truth optimal policy, it is a conservative measure of actual performance on the task." But, as I understand, the authors are measuring accuracy of action-value prediction, and don't obtain 100% accuracy (per Fig. 4 E/F, the accuracy converges to ~90%). But we don't know what level of _accumulated reward_ that corresponds to by acting given the learned action-value function, compared to the optimal action-value function. Without this, it is unclear how useful 90% accuracy is.

---

> > > ### Author Response · Authors · 2023-11-21
> > >
> > > Thank you for your response, we are glad that our answers and updates to the draft have clarified your main concerns.
> > >
> > > As for your additional question, we would like to clarify that in Figure 4E and 4F we report the out-of-distribution accuracy of the learned policy with respect to the ground-truth policy, i.e. we compare the actual actions and not the action-values.
> > > We choose this metric since it measures how well the different architectures perform on the imitation learning objective they were optimized on, allowing us to understand to what extent modular and monolithic architectures learn the compositional structure of the ground-truth policy.
> > > However, we agree with you that this metric does not necessarily inform about the cumulative reward the learned policies obtain.
> > > In order to better contextualize how the learned policies perform, we now complement Figure 4E and 4F with Figure A9 in the updated appendix that shows the fraction of times the learned policy exactly follows the optimal policy along the whole path and therefore obtains the only environment reward by reaching the goal object.
> > > It is possible that the learned policy only deviates from the optimal trajectory in a way that would also yield the reward which would not count towards this fraction.
> > > The conclusions we drew from Figure 4E and 4F remain identical: modular but not monolithic architectures can compositionally generalize when we have compositional support but not when the support is non-compositional.

---

> > > > ### Comment · Reviewer_KGfv · 2023-11-21
> > > >
> > > > Great! Something like Figure A9 is exactly what I was looking for, and indeed it validates the authors' claims. Am I correct in interpreting the results of A9 as being _stronger_ than just achieving the final reward? If I'm not mistaken, the agent could follow a different trajectory and still obtain a reward, yes?
> > > >
> > > > Overall, 2 of my 3 main concerns were adequately addressed by the authors, and hence I will increase my score to 7. In particular:
> > > > - The authors add multiple clarifications in their response and updated manuscript that make the experimental setting a lot more precise.
> > > > - The authors further provide examples that illustrate that the conditions for compositional generalization are not only sufficient but also necessary. It's unclear to me that these examples provide definitive proof, but they are useful.
> > > > - The authors accept the limited interpretability of linear hypernetworks and provide some discussion around that. I would encourage them to include this discussion in their final manuscript.

---

> > > > > ### Author Response · Authors · 2023-11-21
> > > > >
> > > > > We are glad to hear that we were able to address most of your concerns and that you will hence increase your score to a 7. We briefly respond to your open points hoping that we can convince you that also your remaining concern is addressed.
> > > > >
> > > > > > Am I correct in interpreting the results of A9 as being stronger than just achieving the final reward? If I'm not mistaken, the agent could follow a different trajectory and still obtain a reward, yes?
> > > > >
> > > > > Yes, that is exactly correct!
> > > > >
> > > > > > The authors further provide examples that illustrate that the conditions for compositional generalization are not only sufficient but also necessary. It's unclear to me that these examples provide definitive proof, but they are useful.
> > > > >
> > > > > To complete the argument, we have added the (straightforward) example showing why compositional support is necessary for compositional generalization.
> > > > > We will update the final version of the main text to improve the clarity of the whole point.
> > > > >
> > > > > > The authors accept the limited interpretability of linear hypernetworks and provide some discussion around that. I would encourage them to include this discussion in their final manuscript.
> > > > >
> > > > > We have updated the manuscript again to incorporate this point in the discussion with the space available to us:
> > > > > *We show that linearly combining parameters can capture nonlinear latent structure raising an important question for future research on the interpretability of the resulting modules.*

---

### Official Review · Reviewer_9UrZ · 2023-10-31

**Soundness:** 3 good
**Presentation:** 3 good
**Contribution:** 2 fair
**Rating:** 6
**Confidence:** 3

**Summary:**

This paper proposes a multi-task teacher-student approach with modular architecture for compositional generalization.
It uses hypernetworks to convert the generalization to the identification of modules and theoretically show the result up to linear transformation.
Experiments also support the ability.

**Strengths:**

- It focuses on an important question of how to learn modular structure for compositional generalization.
- The hypernetwork and modularity approach cast the generalization problem into an identification
problem.
- Both theory and experiments support the result.
- It proposes connected support to address permutation invariance.

**Weaknesses:**

My largest concern is that the framework is very constrained, and some constraints, e.g., linearity, may be essential to deriving the results.
It indicates that there may be difficulty when generalizing the result to more complex situations.

(1) Linear assumption in hypernetwork.

(2) It uses two two-layer neural networks.

(3) It assumes knowing the correct (teacher) architecture.

(4) The theory assumes knowing the number of modules (M) and hidden units (h).

**Questions:**

Please respond to the weakness section.

---

> ### Author Response · Authors · 2023-11-16
>
> Thank you for your encouraging review bringing to attention the importance of the question on how to learn modular structure for compositional generalization.
> Below we aim to contextualize your concerns on the constraints of our framework replying to your questions point-by-point. We hope these clarifications will convince you to vote for acceptance of the paper, and we remain available for any further questions.
>
> > My largest concern is that the framework is very constrained, and some constraints, e.g., linearity, may be essential to deriving the results. It indicates that there may be difficulty when generalizing the result to more complex situations.
> >
> > (1) Linear assumption in hypernetwork.
>
> We would like to emphasize that we study a general class of compositional problems that can be captured as parameterized functions whose parameters are combinations of parameter modules.
> While we assume that parameter modules are linearly combined to make our theoretical derivation in the modular teacher-student setup tractable, we have carefully designed our experiments to understand how our results extend to complex situations where these assumptions are violated.
> We have tried to clarify this point in the introductory paragraphs to the experiments.
>
> Specifically, we train both a linear and a nonlinear hypernetwork and we consider two tasks - the compositional goals and compositional preferences grid worlds - where the data generating distribution has no known closed-form expression as a linear hypernetwork.
> Despite this setting not being linear, our theoretical insights extend to this setting as we can show that both compositional and connected support are required to achieve compositional generalization in the linear and nonlinear hypernetwork.
>
> We would also like to highlight that the MLP parameterized by the linear hypernetwork is nonlinear and we prove our theoretical results both for the ReLU nonlinearty and a general class of smooth activation functions.
>
>
> > (2) It uses two two-layer neural networks.
>
> Our theoretical analysis of the modular teacher-student setup focuses on two-layer networks, as we are not aware of identification results in the standard teacher-student setup for arbitrary depth which our modular teacher-student setting generalizes.
>
> Moreover, our experiments use neural networks of varying depth beyond two layers - both for the learned models and for the teacher.
>
> > (3) It assumes knowing the correct (teacher) architecture.
> >
> > (4) The theory assumes knowing the number of modules (M) and hidden units (h).
>
> We would like to clarify that in the teacher-student analysis, our theory explicitly considers the case where the number of modules and hidden units is unmatched, showing that it is theoretically possible to construct counterexamples in these cases.
>
> Moreover, we run a number of experiments on the effect of over-parameterization in the student when verifying the theory in Figure 2C (formerly 2D), in the hyperteacher setting in Figure 3F, in the compositional preferences environment in Figure A8 and in the compositional goals environment in Figure A9.
> They show that the sensitivity to over-parameterization is less crucial than suggested by the theory, especially in the more complex environments and compositional generalization is possible despite a mismatch in the number of modules (and hidden units in case a teacher is present).
> We further clarify this point when presenting our experimental results.

---

> > ### Comment · Reviewer_9UrZ · 2023-11-19
> >
> > Thank you for the detailed answers. I have further comments on the following points.
> >
> > > (1) Linear assumption in hypernetwork.
> >
> > As far as I understand, theoretical derivation is still important for non-linear combinations.
> > Though the derivation may not be easily tractable, there are many ground-truth non-linear combinations of parts in the real world.
> > Also, the experiments are not in real and complex settings, so a strong theoretical result is important.
> >
> > > (3) It assumes knowing the correct (teacher) architecture.
> >
> > Does the theory assume that the teacher architecture (e.g., fully connected or convolutional) is known?
> > If so, why is it reasonable?
> > How about in human-computer interaction settings, where the teacher is a human?

---

> ### Author Response · Authors · 2023-11-21
>
> Thank you for taking the time to consider our responses. We are replying to your remaining questions point-by-point below. We hope these clarifications will convince you to vote for acceptance of the paper, and we remain available for any further questions.
>
> >    (1) Linear assumption in hypernetwork.
> > As far as I understand, theoretical derivation is still important for non-linear combinations. Though the derivation may not be easily tractable, there are many ground-truth non-linear combinations of parts in the real world.
>
> We agree that there are many ground-truth nonlinear combinations of parts in the real world that are very interesting and important to consider.
> We would like to clarify however that this is distinct from linearly combining the parameters of a (nonlinear) neural network as is done in the linear hypernetwork we consider in the theory.
> Our experimental results in Figure 4C, 4E and 4F suggest that it is indeed possible to model such nonlinear combinations of parts as you describe (e.g. compositional goals) with linear parameter compositions.
> In particular, note that the linear hypernetwork performs very similarly to the nonlinear hypernetwork in these experiments, both clearly outperforming the nonlinear monolithic baselines.
>
> >    (3) It assumes knowing the correct (teacher) architecture.
> > Does the theory assume that the teacher architecture (e.g., fully connected or convolutional) is known? If so, why is it reasonable? How about in human-computer interaction settings, where the teacher is a human?
>
> The analysis does not assume that the teacher architecture is known given that the irreducibility assumption is not violated.
> The teacher can be any architecture that can be implemented by a MLP, including a convolutional architecture.
> Other parts that specify the architecture in terms of width of the hidden dimension $h$ and the number of modules $M$ are part of the analysis.
>
> > Also, the experiments are not in real and complex settings, so a strong theoretical result is important.
>
> We would like to emphasize that our teacher-student analysis extends state-of-the-art teacher-student analyses, like for example [1,2], to the full multi-task case where task latent variables must be inferred.
> We believe it is a strong theoretical result in the context of the literature.
>
> [1] Tian, Yuandong. "Student specialization in deep rectified networks with finite width and input dimension." International Conference on Machine Learning. PMLR, 2020.
>
> [2] Simsek, Berfin, et al. "Geometry of the loss landscape in overparameterized neural networks: Symmetries and invariances." International Conference on Machine Learning. PMLR, 2021.

---

### Official Review · Reviewer_bgW6 · 2023-10-31

**Soundness:** 3 good
**Presentation:** 2 fair
**Contribution:** 3 good
**Rating:** 6
**Confidence:** 4

**Summary:**

This work explores the ability of hyper-networks to detect the ground-truth task generating functions and have modules (rows of the linear hyper-network weights) specialise to these underlying factors of variation. Moreover these modules can then be composed in previously unused combinations to generalise to new tasks. Experiments demonstrating the benefit of modular meta-learning algorithms over monolithic meta-learners are also shown.

**Strengths:**

## Originality
This paper uses an established technique, the teacher-student setup, to understand the necessary conditions for a dataset to promote systematic generalisation. Moreover, a connection between the teach-student setup and meta-learners is drawn, which I have not seen been made frequently before. Also considering ANIL, MAML and hyper-networks as the class of monolithic and modular meta-learners is interesting a new. Thus, the combination of concepts, theoretical technique and models considered in this work is  new.

## Quality
The theoretical setup of this work seems appropriate for addressing the main concerns of this work - under what dataset conditions do meta-learners naturally generalise compositionally. The overall structure and logical setup of the work is good and sections naturally lead from one to the next. Section 3.3 directly tests the theory which is presented and in the case of Discrete task distributions the experiments support the theoretical findings. As far as I can tell the assumptions of the setup are clearly stated and the setup is clearly defined.

## Clarity
Figures are visually clear and well designed.

## Significance
I reiterate here that I think this work uses a very interesting combination of previous ideas. As a result the reported findings are interesting and I could certainly see the results leading to future work and guiding research on meta learning. Thus, I think this work does have the potential for high significance. I would say this is contingent on some of the weaknesses discussed below being mitigated.

**Weaknesses:**

I have two broad concerns for this work. One is on clarity and the other is on quality. I will begin with clarity as I think this may be a factor in the concerns on quality.

## Clarity
Overall I found this work to be relatively unclear. Notation is used but not properly introduced, for example $(U_k)$ where it is not mentioned what the double subscript $k$ refers to, or why the second subscript is necessary. Another example is where it is said $\forall k,l$ where it is necessary to infer from context what $k,l$ is referring to. Also where it says $U_{k_i}$ what does the $i$ or $k$ refer to and why is this necessary. Similarly in Theorem 3.1 where it says "The if $\mathcal{L}(W_2,\Theta)$...", what is this new $\mathcal{L}$ referring to? Is it the loss and this is just a mistake in the font? If so, why does this loss function not accept the same  parameters as previous loss functions. Finally, the notation introduced in the paragraph beginning "We can now present our main result" is particularly confusing because the superscript $(i)$ is overloaded three times, once referring to a row, the other referring to a column and the third for a full matrix sliced out from a tensor.

Secondly, Definitions 3.1,3.2 and 3.3 are not clear and no intuition or interpretation is provided. For example, where defining irreducibility, the fact that all rows of $W_1$ are pairwise different means that each hidden neuron will be activated for a different feature - and extremely important point for a work concerned with whether meta-learners can extract ground-truth features from a "metateacher". This is not stated. Similarly, how this interacts with nonlinearities on the hidden neurons is ignored and what it means for no columns of $W_2$ to be $0$ is also not mentioned. Why would a full column of $W_2$ be $0$? Similarly, "Compositional Support" seems to just be saying that $U$ is a basis of $\mathbf{R}^M$ and so I am not certain of how this new terminology is necessary. Is compositional support a weaker case of having a basis as $U$ does not need to be the **minimal** spanning set? Would Definition 3.1 and 3.2 together then imply that $U$ is a basis? Definition 3.3 is just extremely difficult to parse in general and so is Theorem 3.1. This is due to a lot of terminology being mentioned without definition and needing to be inferred based on context. For example, what is $\hat{M}$ and why is Theorem 3.1 making a distinction between even and odd values of $n$? The difficulty in following these definitions alone makes the rest of the work difficult to follow.

My final point on clarity is that the figures, while visually neat and well done, are vague and their captions unhelpful. This is particularly bad when the figures are relied on heavily to explain concepts. For example, where it is said "See Figure 2B for an illustration of a connected support" and then the caption does not explain what a connected support is or how the connected vs disconnected task families connects to the actual Definition 3.3. Essentially, every figure caption should be elaborated on and potentially more information be placed in the figures to depict what is actually going on. For example, in Figure 1B, the tiling of the $x,y$ space is not connected at all to the rest of the work beyond the notation of $p(\tau)$.

## Quality
My concerns on quality are likely due to misunderstanding from the above. I would like to reiterate that I do believe this work has potential significance. I am, however, struggling to connect this work to the general literature on compositional generalisation. For example, assuming that $P_x$ has full support over the input space. I see how the learned weights of the linear hyper-network are compositional, in the sense that they operate similar to a set of basis vectors, which in the case of a linear mapping is also features for the network. However, this is then more similar to feature learning rather than exact modules. Or is the idea here that the features being learned which align with ground truth is the same as identifying a composable module? How would this then tie in with disentanglement, which implies compositionality. While I would be open to such claims - the most obvious on to me here being that feature learning and module learning in the limited setting you study are the same thing - I think that argument needs to be made explicit. This would be a different take on modularity in general though and systematic generalisation which tends to focus more on how separate pieces are learned to be separate in spite of covariances and this makes the problem easier [1]. In your case it seems more like a claim that if the input space is sufficiently explored then the network will learn the ground truth features but just because this is the only way to learn the task (to learn a full-rank basis set) and is not in fact learning to identify or solve a smaller problem which is then composable.

The experiments of Section 4 seem to be more in line with the standard notions of compositionality [2], however due to the above issues grounding the theory to larger scale models of compositionality I also struggle to see how Section 4 fits in with the rest of the work, beyond just demonstrating that hyper-networks are better in compositional domains than ANIL and MAML. Is there a greater connection beyond this (I do think this result is important in its own right though)?

I would be open to increasing my score quite substantially if it is shown that I have indeed missed something crucial. Alternatively if the clarity issues are addressed and the connection to prior work made more explicit I would also increase my score. Likely to a 7. I would certainly prioritise improving the clarity, with the figures being particularly low hanging fruit which would make quite a big difference if improved upon.

## Minor Points
1. "with each row representing one parameter module" - I believe I understood what you were saying, but this was not an easily understandable sentence.

[1] Hadley, Robert F. "Systematicity in connectionist language learning." Mind & Language 9.3 (1994): 247-272.
[2] Ruis, Laura, et al. "A benchmark for systematic generalization in grounded language understanding." Advances in neural information processing systems 33 (2020): 19861-19872.

**Questions:**

I have raised a number of questions in my discussion above. I think the only outstanding question or point of clarification I have at this point is the following:
Could the authors please explain what Figure 2A is aiming to show with the permutation invariance? It is only referenced in condition (ii) but then not explained in the figure. I think I would benefit from understanding this point better.

---

> ### Author Response · Authors · 2023-11-16
> **Part 1**
>
> Thank you for expressing appreciation for the significance of our work and your very constructive feedback. Based on your input, we have significantly revised the presentation of the theory in the main text as well as the appendix, and are overall confident the quality of the paper improved as a result. We address your questions point by point below, and do our best to clarify important miscommunications. Please do not hesitate to let us know if anything remains unclear.
>
>
> > Notation is used but not properly introduced [...]
>
> Thank you for raising this issue.
> We have revised our notation by simplifying it where possible and ensuring that symbols are properly introduced.
> We hope this makes it easier to parse our theory and addresses your questions concerning the notations.
>
> >  Definitions 3.1,3.2 and 3.3 are not clear
>
> Following the revised notation, we have tried to improve the clarity of the definitions in Section 3.2.
> We refrain from restating them here to avoid cluttering our response and would like to kindly ask you to revisit our updated manuscript.
>
> > For example, where defining irreducibility, the fact that all rows of are pairwise different means that each hidden neuron will be activated for a different feature - and extremely important point for a work concerned with whether meta-learners can extract ground-truth features from a "metateacher". This is not stated. Similarly, how this interacts with nonlinearities on the hidden neurons is ignored and what it means for no columns of to be is also not mentioned. Why would a full column of be ?
>
> Thank you for raising this point.
> To understand the purpose of the irreducibility condition, consider the standard (i.e. single task) teacher-student setting with two-layer neural networks.
> Note that there are many different two-layer neural networks that are functionally equivalent.
> When aiming at characterizing the exact set of neural networks that are functionally equivalent to a given teacher network, we are free to pick the teacher network to be any member of that equivalence class. A convenient choice is to choose it to be minimal, i.e. with the smallest possible number of hidden units. The definition of irreducibility is a necessary (and sufficient in the case of the smooth nonlinearity originally used in the main text) condition for such minimality: whenever two input weights are equal, one can merge the associated hidden neurons without changing the function implemented by the network. Likewise, if an output weight is 0, one can remove the associated hidden neuron.
> We have added a paragraph that explains this point at the beginning of Appendix A.1, and revised the definition of irreducibility to more clearly communicate the above point.
>
> > Compositional Support seems to just be saying that $U$ is a basis $\mathbb{R}^m$ of and so I am not certain of how this new terminology is necessary.  Is compositional support a weaker case of having a basis as $U$ does not need to be the minimal spanning set? Would Definition 3.1 and 3.2 together then imply that $U$ is a basis?
>
> Compositional support indeed simply ensures that all modules have been encountered at least once during training.
> We adopt the notion of "compositional support" from related work [1].
> It is a useful definition for our theory as violating it will pose an obvious but important problem for compositional generalization and we can use this property as an experimental intervention to inform to what extent a learned model has discovered underlying compositional structure (c.f. Figures 3C, 4C and 4F).
>
> Indeed as you correctly state, compositional support is a weaker case than having a basis of $\mathbb{R}^m$ as $Z$ (formerly $U$) does not need to be the minimal spanning set.
>
> > Definition 3.3 is just extremely difficult to parse in general and so is Theorem 3.1.
>
> Thank you for raising this point.
> We have revised the formalism used to state Definition 3.3 and Theorem 3.1 in an attempt to make them easier to parse.

---

> ### Author Response · Authors · 2023-11-16
> **Part 2**
>
> > The figures, while visually neat and well done, are vague and their captions unhelpful.
> > For example, where it is said "See Figure 2B for an illustration of a connected support" and then the caption does not explain what a connected support is or how the connected vs disconnected task families connects to the actual Definition 3.3.
> > For example, in Figure 1B, the tiling of the $x,y$ space is not connected at all to the rest of the work beyond the notation of $p(\tau)$.
>
> Thank you for pointing out the issues with our figures and corresponding captions.
> According to your suggestions, we have tried to improve their clarity.
> In particular, we give a detailed explanation of the example visualized in Figure 2A (formerly 2B) in the beginning of section 3.2, and have reworked Figure 1B to better aid its purpose of illustrating the modular teacher-student setup and how we can use it to investigate compositional generalization.
> It now shows a concrete toy example that illustrates how during training only a subset of module compositions of the teacher hypernetwork is used to construct the tasks and how this allows us to test for compositional generalization by presenting a novel composition.
>
> > For example, assuming that $\mathcal{P}_x$ has full support over the input space. I see how the learned weights of the linear hyper-network are compositional, in the sense that they operate similar to a set of basis vectors, which in the case of a linear mapping is also features for the network. However, this is then more similar to feature learning rather than exact modules. Or is the idea here that the features being learned which align with ground truth is the same as identifying a composable module?
>
> In our setting, the linear hypernetwork takes a low dimensional task embedding $z$ and generates the weights of a base network which is a two-layer MLP. Each of these generated networks will have their own feature extractors over the input $x$, and the linear hypernetwork thus parametrizes a complex family of feature extractors by linearly combining the parameter modules (meta-features) through $z$.
>
> In our teacher-student result, we show a characterization of what these modules and the distribution over $z$ must be, in order for the identification of these meta-features. Crucially, while we assume the input distribution $\mathcal{P}_x$ provided for each of the generated networks to have full support, we only require the support of the (hidden) task latent variables $\mathcal{P}_z$ to be compositional and connected for Theorem 1 (compositional generalization) to hold. In practice, this means we can show tasks from a subset of module combinations during training (in fact, a finite number of tasks linear in the number of modules), holding out a large number of combinations to test for out-of-distribution generalization as a measure of compositional generalization during evaluation.
>
> > How would this then tie in with disentanglement, which implies compositionality. While I would be open to such claims - the most obvious on to me here being that feature learning and module learning in the limited setting you study are the same thing - I think that argument needs to be made explicit. This would be a different take on modularity in general though and systematic generalisation which tends to focus more on how separate pieces are learned to be separate in spite of covariances and this makes the problem easier [1]. In your case it seems more like a claim that if the input space is sufficiently explored then the network will learn the ground truth features but just because this is the only way to learn the task (to learn a full-rank basis set) and is not in fact learning to identify or solve a smaller problem which is then composable.
>
> Each parameter module defines a number of features in the teacher network that are composable in the parameter space of the teacher.
> Our result on linear identification in the student intuitively says that the learned student modules are a linear combination of the teacher modules.
> This means for each module in the teacher, there is a linear combination of student modules that will recover the teacher features in the student (up to permutation) on each task.
> These results are therefore somewhat complementary to the typical study of disentangelment in feature learning in the sense that each teacher module defines a whole set of features and we tackle the question whether a student is able to discover the teacher modules when only exposed to tasks that contain a subset of the possible module combinations.
> We hope that this explanation helped clarify your point.
> Please let us know if this did not fully answer your question.

---

> ### Author Response · Authors · 2023-11-16
> **Part 3**
>
> > The experiments of Section 4 seem to be more in line with the standard notions of compositionality [2], however due to the above issues grounding the theory to larger scale models of compositionality I also struggle to see how Section 4 fits in with the rest of the work, beyond just demonstrating that hyper-networks are better in compositional domains than ANIL and MAML. Is there a greater connection beyond this (I do think this result is important in its own right though)?
>
> We have tried to improve the introductory paragraphs in the experimental section to make the connection to the rest of the paper more comprehensible in an attempt to clarify that the experiments of Section 4 consider a similar setup as the theory in Section 3 with the difference that
>
> 1. We learn from finite data, i.e. we no longer assume the input distribution $\mathcal{P}_\vx$ has full support
> 2. As a result we cannot consider the simplified objective of equation (2) but instead employ the bilevel optimization objective of equation (1), better suited for the many-shot setting, and computationally friendlier as it does not require a full optimization in the inner loop.
> 3. We move beyond the modular teacher-student setting, introducing a mismatch in the data generating process and the model architecture. In particular, we consider tasks for which it is a priori not clear how well neural networks can capture the underlying compositional structure.
> 4. We contrast the performance of our modular architectures with monolithic architectures to highlight the benefits of the former despite their less frequent use in practice.
>
> Within this practical and more complex setting, we test predictions made by our theory and show that the central assumptions of compositional and connected support play a crucial role.
>
> > Could the authors please explain what Figure 2A is aiming to show with the permutation invariance? It is only referenced in condition (ii) but then not explained in the figure. I think I would benefit from understanding this point better.
>
> Thank you for pointing out the missing explanation of Figure 2A in our initial submission.
> We have extended our explanation of this and other failure cases in Appendix A2 and moved the figure to this section. The caption for what is now Figure A1 now reads as follows:
>
> Toy example of how correct identification of individual modules can lead to inconsistent neuron permutations preventing compositional generalization to unseen module compositions: Consider the simplified setting of a teacher and student network with two input neurons and three hidden neurons. Both the teacher and the student have $M=3$ modules. The upper right defines the teacher weights for each module and each neuron. For instance the weights denoted by B correspond to the weights connecting neuron 2 to the input for module 1. We now assume the student during training encounters three tasks. For each task exactly one of the teacher modules is used. Since in MLPs the ordering of neurons is permutation invariant, the student can perfectly match the teacher outputs, even when it uses a different ordering of the neurons. As a result, the student modules can perfectly fit all three tasks, despite permuting the neuron indices. For instance, neuron 2 in module 3 of the student contains the weights H whereas the corresponding neuron in the teacher contains the weights G. When we now present a new task during out-of-distribution evaluation that mixes all three modules in the teacher, the student is required to mix the weights of each neuron across modules. Since the neuron permutations in the student are inconsistent with those of the teacher, the student is unable to create the correct neuron weights and ends up unable to generalize to the OOD task.
>
> [1] Wiedemer, Thaddäus, et al. "Compositional generalization from first principles." arXiv preprint arXiv:2307.05596 (2023).

---

> > ### Comment · Reviewer_bgW6 · 2023-11-18
> > **Response to New Draft (Part 1 of 2)**
> >
> > Firstly, let me apologize for the formatting issue in my original review. It seems the authors managed to get the meaning I was intending, but still I should have caught this error.
> >
> > I will respond to the new draft of the paper and make comments as I read it. Hopefully this will give the authors a better sense of a readers experience while going through it.
> >
> > I am at the end of the Methods section now. This is much clearer than the original version and I commend the authors for their quick effort. However, there is still room for improvement.  For example in the Compositionality paragraph "task-shared function g" is brought up but not described or used until the following page. Thus, at this point while reading this is just another ungrounded symbol taxing my working memory. Similarly the latent code z is brought up but left competely abstract. It is the brought up again half a page later where it is now mentioned that it has compositional structure, but even by the end of the methods section very little is told to use about z. Why not reference Figure 1B here since it is very simply depicted there. But also, why not say z has binary elements with at least two non-zero elements for any task? I think restructuring section 2 so that it is first **Modular and Monolithic Architectures**, then **Compositionality** and finally **Meta-learning** would provide a more grounded and seamless introduction of notation. At least then the actual architecture would be introduced first too. Also the two different uses of $\phi$ is jarring. I understand the authors are trying to use the same symbols for task-specific parts of the architecture, but for the hyper-network it is a vector used inside of the $g$ function while for ANIL it is a matrix which acts on the output of the $g$ function. Finally, where it says "we allow the model to infer the task latent code" and then says "...by adapting task-specific parameters $\phi_z(\theta)$" this is confusing. As far as I can tell the function $\phi$ is producing task-specific parameters from task-general parameters. So it is not inferring $z$ (the task latent code) and nor is it adapting task-specific parameters (surely it adapts the task-general feature to be more task-specific)? This entire discussion on the data split is also quite complex and formal to express the fact that the training data needs to be split to do learning and meta-learning on separate portions of the dataset. Especially considering this split is then immediately ignored. At this point while reading it seems like a complicated detour. Not enough discussion is then given on how equation 2 is then optimized. Is this just the same as using equation 1 without splitting the data?
> >
> > I'm at the end of Section 3.1. The first half of Section 3.1 is what Section 2 should be. You essentially re-introduce the model architecture and latent variables here for you model. It is clearer here, but similarly, why not reference Figure 1? Also you should mention what values **z** can take still, such as $z^{(i)} \in R$. There's also a couple point where things are said which seem like contradictions but might be due to a lack of detail. The worst case of this is where it says "hidden dimension potentially different from the teacher" but assumption (iii) says "no over-parametrization of the student wrt the teacher".
> >
> > In Section 3.2. Why are "task families" now being introduced? We are at least half way through the paper and still having to learn new notation and definitions. The example is helpful though. I'm still not convinced by the notion of neuron permutations being inconsistent. I see more explanation is in the appendix, but this is quite crucial to the main work and needs to be clearer here. Does this just mean that the same hidden neurons might be used by both tasks? Is the main idea of connected support just that it forces the network to partition it's hidden layer for each task? Theorem 1 then says that the student matches the teacher number of hidden neurons which contradicts what is said above. Saying "under some additional technical conditions" is frustrating. I appreciate presenting theory in a short format is difficult, but this is not a theorem statement. I know it says "informal" as well but this is just unhelpful. I would usually argue for a proof sketch but at this point a proper theorem would be helpful. Does Theorem 2 imply Theorem 1? If this is the case why not just give a full account of Theorem 2 and then note that it implies compositional generalization? I am running out of characters and so will split this here. I will post the second part soon with the rest. I hope this is at least helpful to the authors. I do not intent to be petty or antagonistic but I do still see room for improvement (beyond the already significant improvement).

---

> > > ### Comment · Reviewer_bgW6 · 2023-11-18
> > > **Response to New Draft (Part 2 of 2)**
> > >
> > > I am at Section 4. Why is Section 3,3 made separate from Section 4? Is Section 4 not empirical verification? Section 4 seems more like a results section. My original issues with the paper centred less on the experiments and from what I can tell the other reviewers also seem to believe that they support the findings of this work. The captions of Figure 3 and 4 are much more detailed which is great. I'm still not certain of the main concept of the theory which is limiting my evaluation of the experiments and so I will move on to this now.
> > >
> > > I think on a conceptual level I would really benefit from some indication of what it means for the hyper-network to learn the correct modules. Does it imply that a module learns the same features as if it was trained on a task vector of $z = [1,0,0,0,0]$ for example. In other words it identifies a single base or primitive task? This then allows for composition because it has learned different features than if it specialized to $z = [1,0,0,0,1]$ lets say? Is this what you mean by "linear combination of teachers"? It seems like it is very really stated in concrete terms what is desired. Even the example at the top of page 5 is not completed somewhere. I would really like to "close the loop" so to speak on what is being discussed.
> > >
> > > As a slightly off-topic question. How would this work align with findings on double descent where generalization is hurt most when a network is only sufficiently parametrized and improves when it is over-parametrized? It seems your results would indicate the opposite. I'm just asking here, the authors can ignore this if they like.
> > >
> > > I would like to acknowledge that I have read the authors rebuttals prior to re-reading the new draft and so the rebuttal has been incorporated into my understanding of this work and subsequent questions. However, if I am still missing something key I encourage the authors to point this out. To acknowledge the significant improvement that has already been made I am raising my score to a 5, but still feel more is needed before I advocate for acceptance. I'm happy to engage further.

---

> > > > ### Author Response · Authors · 2023-11-20
> > > > **Part 1**
> > > >
> > > > We appreciate your updated score and thank you very much for actively engaging with additional helpful feedback and suggestions.
> > > > We reply to your individual questions and suggestions point by point below.
> > > >
> > > >
> > > > > I think on a conceptual level I would really benefit from some indication of what it means for the hyper-network to learn the correct modules. Does it imply that a module learns the same features as if it was trained on a task vector of $z = [1,0,0,0,0]$ for example. In other words it identifies a single base or primitive task? This then allows for composition because it has learned different features than if it specialized to $z = [1,0,0,0,1]$ lets say? Is this what you mean by "linear combination of teachers"? It seems like it is very really stated in concrete terms what is desired. Even the example at the top of page 5 is not completed somewhere. I would really like to "close the loop" so to speak on what is being discussed.
> > > >
> > > > This is indeed a very important point we now try to better convey in the newly updated manuscript.
> > > > We would kindly ask you to specifically  consider the new panel B we have created in Figure A1 that now complements the failure case of no identification with the successful case of *linear identification*, illustrating what it means for the hypernetwork to learn the correct modules.
> > > >
> > > > To answer your particular question whether "a module learns the same features as if it was trained on a task vector of $z = [1,0,0,0,0]$ for example": *Linear identification*, which we show is a necessary and sufficient condition for compositional generalization, is slightly different from what you describe.
> > > >
> > > > First, what you describe has a possible failure case. Even if for every individual one-hot task vector $z = [1,0,0,0,0]$, $z = [0,1,0,0,0], \dots, z = [0,0,0,0,1]$ each module in the student learns the same features as the corresponding teacher module, we cannot ensure that the features will consistently correspond to the same neuron index across modules due to the permutation invariance of neural networks. But this becomes crucial when considering a new task composition that mixes neurons across modules. For instance, $z = [1,1,1,1,1]$ requires all neuron indices to be consistent across modules as otherwise we mix a different set of neuron weights in the student than in the teacher to construct a particular feature.
> > > >
> > > > Second, if you intent to say that each module in the teacher has exactly one corresponding module in the student, this is stronger than linear identification. The latter only requires that for each teacher module, there is a linear combination of student modules that recovers the teacher. Crucially this linear combination needs to be consistent across neurons.
> > > >
> > > >
> > > > > Similarly the latent code z is brought up but left competely abstract. It is the brought up again half a page later where it is now mentioned that it has compositional structure, but even by the end of the methods section very little is told to use about z. Why not reference Figure 1B here since it is very simply depicted there.
> > > >
> > > > Thank you for this suggestion, we indeed were missing a proper reference to Figure 1B after its revision in the updated version. We have added it to the newly updated version.
> > > >
> > > > > But also, why not say z has binary elements with at least two non-zero elements for any task?
> > > > > Also you should mention what values z can take still, such as $z^{(i)} \in \mathbb{R}$.
> > > >
> > > > Thank you for pointing out that we failed to specify the domain of $z$, we have fixed this in the newly updated manuscript.
> > > > We are not exclusively considering binary task latent codes but as you correctly state $z^{(i)} \in \mathbb{R}$.
> > > > To make this point clearer earlier on in the paper, we update Figure 1B accordingly.
> > > > We would like to emphasize that our theory considers this more general case and allows the sparse values of the task latent code to take arbitrary values in $\mathbb{R}$. In some of the experiments however, we do exclusively use such binary training tasks (i.e. "discrete" task distribution), as an interesting  special case of the theoretical result which considers "continuous" distributions.
> > > >
> > > >
> > > > > I think restructuring section 2 so that it is first Modular and Monolithic Architectures, then Compositionality and finally Meta-learning would provide a more grounded and seamless introduction of notation. At least then the actual architecture would be introduced first too.
> > > >
> > > > Thank you for this suggestion. We have indeed considered doing so in an earlier version of our manuscript but found it more difficult to motivate this choice of architecture before having introduced the notion of compositionality we consider in this paper.

---

> > > > > ### Author Response · Authors · 2023-11-20
> > > > > **Part 2**
> > > > >
> > > > > > Also the two different uses of $\phi$ is jarring. I understand the authors are trying to use the same symbols for task-specific parts of the architecture, but for the hyper-network it is a vector used inside of the $g$ function while for ANIL it is a matrix which acts on the output of the $g$ function.
> > > > >
> > > > > Thank you for bringing this potentially confusing aspect to our attention.
> > > > > We are following with our notation what we perceive to be the standard in the meta-learning literature.
> > > > > The motivation behind this particular choice of using the same symbol to denote the task-specific and task-shared parameters across architectures is that it allows to state one unified meta-learning objective.
> > > > > We have nevertheless removed the symbols from the Figure visualizing the architectures to partially alleviate the confusion it might create.
> > > > >
> > > > > > Finally, where it says "we allow the model to infer the task latent code" and then says "...by adapting task-specific parameters $\phi_z(\theta)$ " this is confusing. As far as I can tell the function $\phi$ is producing task-specific parameters from task-general parameters. So it is not inferring (the task latent code) and nor is it adapting task-specific parameters (surely it adapts the task-general feature to be more task-specific)?
> > > > >
> > > > > Thank you for raising this point. We have now reformulated the sentence to improve its clarity. We now state "We allow the model to infer the task latent code from the demonstrations in the support set $\mathcal{D}_z^{\text{support}}$ by optimizing the task-specific parameters $\phi$, obtaining $\phi_z(\theta)$,...".
> > > > >
> > > > > We would like to emphasize that for each task, we do optimize the task-specific parameters $\phi$ given task-shared parameters $\theta$ using gradient-based optimization.
> > > > > The resulting task-specific parameters will therefore depend on the current task-shared parameters $\theta$, hence why we indicate the dependence $\phi_z(\theta)$. We then update the task-shared parameters across tasks given the optimized task-specific parameters.
> > > > >
> > > > >
> > > > > > This entire discussion on the data split is also quite complex and formal to express the fact that the training data needs to be split to do learning and meta-learning on separate portions of the dataset. Especially considering this split is then immediately ignored. At this point while reading it seems like a complicated detour. Not enough discussion is then given on how equation 2 is then optimized. Is this just the same as using equation 1 without splitting the data?
> > > > >
> > > > > Thank you for bringing this to our attention.
> > > > > We have tried to improve the clarity of this sentence motivating this complex exposition as an important aspect for the methods of the experiments in Section 4 that we can only simplify in the theory since we assume the infinite data limit: "To derive our theory, we assume the infinite data limit which allows us to consider a simpler multi-task learning problem with $\mathcal{D}_z^{\text{query}} = \mathcal{D}_z^{\text{support}}$ resulting in a single objective..."
> > > > >
> > > > > We would like to emphasize that in our experiments in Section 4 we consider the practical setting where we have finite data.
> > > > > In this case the cross-validation objective (splitting the data for each task into a support set and query set) is important as it is known to find solutions that overfit less to the little data provided.
> > > > >
> > > > > > The first half of Section 3.1 is what Section 2 should be. You essentially re-introduce the model architecture and latent variables here for you model. It is clearer here, but similarly, why not reference Figure 1?
> > > > >
> > > > > This is again a good idea, we added the reference to Figure 1 now.
> > > > > The specific model architecture we specify in Section 3.1 is precisely the one studied in the theory. For the experiments in Section 4 however, we consider more variants of the general class of architectures described more generally in the corresponding paragraphs of Section 2. Hence we unfortunately require some level of redundancy.
> > > > >
> > > > > > There's also a couple point where things are said which seem like contradictions but might be due to a lack of detail. The worst case of this is where it says "hidden dimension potentially different from the teacher" but assumption (iii) says "no over-parametrization of the student wrt the teacher".
> > > > >
> > > > > We would like to clarify that we are interested in understanding the conditions that allow for compositional generalization.
> > > > > Varying the hidden dimension of the student in relation to the teacher is a central part of our study.
> > > > > Theoretically, we identify "no over-parameterization" as one of the important conditions to enable compositional generalization.
> > > > > Empirically, we find that it depends on the precise setting how sensitive the ability for compositional generalization is to over-parameterization as we show in Figure 2C, 3F, A8 and A9.

---

> > > > > > ### Author Response · Authors · 2023-11-20
> > > > > > **Part 3**
> > > > > >
> > > > > > > In Section 3.2. Why are "task families" now being introduced? We are at least half way through the paper and still having to learn new notation and definitions. The example is helpful though.
> > > > > >
> > > > > > Indeed it is an intricate balance when and how much notation to introduce and we generally agree with your assessment.
> > > > > > In this particular case, the distinction between a task (a sparse vector) and a task family (a set of task vectors that all share the same sparsity pattern defined with a binary mask) is important to understand our theoretical result.
> > > > > > We therefore introduce it as close as possible to the definitions and theorems where it is most clearly motivated why this additional concept is needed.
> > > > > > We acknowledge however that the naming might be improved, and are open to suggestions.
> > > > > >
> > > > > > > I'm still not convinced by the notion of neuron permutations being inconsistent. I see more explanation is in the appendix, but this is quite crucial to the main work and needs to be clearer here. Does this just mean that the same hidden neurons might be used by both tasks? Is the main idea of connected support just that it forces the network to partition it's hidden layer for each task? Theorem 1 then says that the student matches the teacher number of hidden neurons which contradicts what is said above.
> > > > > >
> > > > > > We hope our prior responses, especially to the first point along with the added panel B in Figure A1, address your questions here. Please do not hesitate to ask for further clarification if this is not the case. In short: Since each module of the hypernetwork can be thought of as fully parameterizing all hidden neurons of the target network and the hypernetwork is able to linearly combine modules in parameter-space, intuitively the idea is that connected support (+ the other conditions of Theorem 1) ensures that such linear combinations combine what would be the same neuron in the teacher.
> > > > > >
> > > > > > > Saying "under some additional technical conditions" is frustrating. I appreciate presenting theory in a short format is difficult, but this is not a theorem statement. I know it says "informal" as well but this is just unhelpful. I would usually argue for a proof sketch but at this point a proper theorem would be helpful.
> > > > > >
> > > > > > Thank you for raising that this is a potential point of frustration for the reader.
> > > > > > We have added the additional technical conditions directly to the theorems in the main text.
> > > > > > In addition, just before the theorems we provide links to the respective full proofs in the appendix.
> > > > > >
> > > > > > > Does Theorem 2 imply Theorem 1? If this is the case why not just give a full account of Theorem 2 and then note that it implies compositional generalization?
> > > > > >
> > > > > > Theorem 2 does not directly imply Theorem 1 although the two are closely related. We believe they are still distinct results that profit from being presented separately.
> > > > > > Specifically, Theorem 2 states that when the student achieves zero loss for all task latent variables (i.e. not only on the training tasks), then the student modules are a linear combination of the teacher modules.
> > > > > > Theorem 1 on the other hand establishes the conditions under which zero loss on the training distribution leads to zero loss on all task latent variables.
> > > > > >
> > > > > > > I am at Section 4. Why is Section 3,3 made separate from Section 4? Is Section 4 not empirical verification? Section 4 seems more like a results section.
> > > > > >
> > > > > > Thank you for bringing up this potential point of confusion. You are correct that Section 4 is also an empirical verification of the theory. It differs since it considers a more general setting (finite data with full bilevel optimization, different forms of compositional data, comparison to monolithic architectures) as we denote in the introduction of Section 4.
> > > > > > We have changed the section heading of section 3.3 to "Empirical verification in the theoretical setting" to make this distinction clearer.

---

> > > > > > > ### Author Response · Authors · 2023-11-20
> > > > > > > **Part 4**
> > > > > > >
> > > > > > > > As a slightly off-topic question. How would this work align with findings on double descent where generalization is hurt most when a network is only sufficiently parametrized and improves when it is over-parametrized? It seems your results would indicate the opposite. I'm just asking here, the authors can ignore this if they like.
> > > > > > >
> > > > > > > This is indeed a great remark and question, and one of the reasons we consider our result to be interesting.
> > > > > > >
> > > > > > > Since our theoretical analyses consider the equilibrium condition without taking into account the inductive bias of the optimization procedure itself, at this point we cannot say anything theoretically about the presence or absence of the double descent phenomenon.
> > > > > > >
> > > > > > > However, we do know from the construction of our counter-examples that achieving zero loss on sparse training task vectors will no longer guarantee generalization when training an over-parameterized model. Furthermore, empirically, it seems that despite the use of gradient descent, the generalization degrades above a certain point of over-parameterization for discrete task distributions.
> > > > > > >
> > > > > > > Therefore, we believe that the double descent phenomenon, typically achieved through the increased smoothness of the model as it is more and more over-parameterized, does not apply when considering the type of generalization we consider, namely compositional generalization. This is intriguing, as it perhaps questions the ever increasing scale of state of the art models when it comes to obtaining models which can generalize compositionally.
> > > > > > >
> > > > > > > We note however that for continuous task distributions, the degradation was not observed despite over-parameterization. We believe investigating the effect of the training task distribution in conjunction with the optimization procedure on the capacity for compositional generalization is a very interesting direction for future research.
> > > > > > >
> > > > > > >
> > > > > > > Please do not hesitate to let us know if anything remains unclear, and we hope our change in the manuscript will convince you to vote for an acceptance of the paper.

---

> > > > > > > > ### Comment · Reviewer_bgW6 · 2023-11-22
> > > > > > > > **Response to Part 4**
> > > > > > > >
> > > > > > > > I will note that I have not read a third draft of the paper if one was uploaded after my first round of rebuttal responses. That said, based on the authors most recent set of comments, I understand the work far better. I also think that it is clear in what ways the authors can improve on clarity based on this discussion. Thus, I believe a final draft will meet the standard for acceptance and I am increasing my score to a 6.
> > > > > > > >
> > > > > > > > Thank you again to the authors for the constructive discussion.

---

> > > > > > > ### Comment · Reviewer_bgW6 · 2023-11-22
> > > > > > > **Response to Part 3**
> > > > > > >
> > > > > > > On the definition of task families, I do acknowledge that it is difficult. This point was more to guide the authors on my confusion. Given the improved Figure 3B and overall presentation this is also less of an issue. Perhaps Figure 3B could once again be reference or used as an example. More importantly, the statement in the above comment: "the distinction between a task (a sparse vector) and a task family (a set of task vectors that all share the same sparsity pattern defined with a binary mask)" is perfect. This was clear and concise and I understand perfectly why this is needed. I urge the authors to add this into the paper and aim for more of these sorts of statements.
> > > > > > >
> > > > > > > The above discussion on linear identification does indeed clear-up my confusion for why permutation invariance matters. I hope this discussion is helpful for guiding how these concepts are introduced in the paper - but I am now much more comfortable with why the paper raises this as an important point. This is also making it clear to me why connected support would be an important point. To be concrete, I recommend the authors make the notion of linear identification as concrete in the paper as they did in Part 1 above. Even going so far as to explain that the one-to-one student-teach module correspondence can still fail. Figure 3B is a significant improvement, but I think some more could be done with the caption to really hammer home this concept.
> > > > > > >
> > > > > > > Thank you for clarifying Theorem 1 and 2. I agree now that they can be left separate. I think something similar to "Specifically, Theorem 2 states that when the student achieves zero loss for all task latent variables (i.e. not only on the training tasks), then the student modules are a linear combination of the teacher modules. Theorem 1 on the other hand establishes the conditions under which zero loss on the training distribution leads to zero loss on all task latent variables." should be added to the paper if it hasn't already been.

---

> > > > > > ### Comment · Reviewer_bgW6 · 2023-11-22
> > > > > > **Response to Part 2**
> > > > > >
> > > > > > Once again, thank you for the elaboration. I accept all the above. I am still a bit uncertain about the manner in which over-parametrization is spoken about. I would recommend the authors perhaps be clear about when they are discussing hidden layer width as over-parametrization and when they are discussion the number of modules. This would definitely help a lot, for example in the last subsection of Section 3.3.

---

> > > > > ### Comment · Reviewer_bgW6 · 2023-11-22
> > > > > **Response to Part 1**
> > > > >
> > > > > I thank the authors for continuing to elaborate more. Did the authors upload a third version of the paper? Based on Figure 1B it seems so.
> > > > >
> > > > > I must say I think a lot of what was just stated needs to be said in the paper. The discussion around learning to match every teacher module with a student module having the failure case is even more concrete in the above comment - and I now think I see better why the permutation invariance argument is necessary.
> > > > >
> > > > > As for the suggestion on restructuring section 2, if the authors say it did not help then I will accept this. I hope my suggestions will still be incorporate to improve the clarity with the current structure of section 2.

---

### Official Review · Reviewer_4knX · 2023-11-01

**Soundness:** 2 fair
**Presentation:** 1 poor
**Contribution:** 2 fair
**Rating:** 6
**Confidence:** 2

**Summary:**

This paper studies the problem of compositional generalization in modular architectures. The authors show that in the teacher-student setting, it is possible to identify the underlying modules up to linear transformations purely from demonstrations. They further show that meta-learning from finite data can discover modular solutions that generalize compositionally in modular but not monolithic architectures. The authors also demonstrate how modularity implemented by hypernetworks allows discovering compositional behavior policies and action-value functions.

**Strengths:**

1. The paper theoretically shows that students can learn the underlying modules from the teachers under certain conditions.
2. The results are supported by empirical experiments.

**Weaknesses:**

1. The paper is not very well written and is hard to follow. There are no simple examples explaining the problems the authors are trying to solve.
2. There is no section that explicitly discusses related work.

**Questions:**

1. Could you rearrange the paper to have a background section explaining things like MAML, hypernetworks, etc, for readers who are not familiar with these concepts?
2. Could you explain why the network modules have to be hypernetworks in your setting?


-----------
update: raised to 6.

---

> ### Author Response · Authors · 2023-11-16
>
> Thank you for your comments and for taking the time to review our work.
> We have spend considerable effort to improve the presentation of the paper in order to address your main point about the difficulty of understanding our initial submission.
> Below we address your individual suggestions and questions point by point.
> We hope these clarifications will convince you to vote for acceptance of the paper, and we remain available for any further questions.
>
> > The paper is not very well written and is hard to follow. There are no simple examples explaining the problems the authors are trying to solve.
>
> We have revised the presentation of the paper in general and the theoretical exposition in particular. We hope this makes the paper easier to follow. In particular, section 3.2 which contains our main theoretical result has been revised. It now starts with a concrete example accompanied by an illustration in Figure 2B and states definitions and theorems in a way we hope is easier to parse.
>
> > There is no section that explicitly discusses related work.
>
> Thank you for making us aware of this potential misunderstanding.
> We have treated related work as part of our discussion section. We clarify this now by changing the section title to "Related work and Discussion". In addition, we added an "Extended related work" section in Appendix~E.
>
> > 1. Could you rearrange the paper to have a background section explaining things like MAML, hypernetworks, etc, for readers who are not familiar with these concepts?
>
> We have revised the methods section to make the exposition to hypernetworks and MAML easier to digest for readers not familiar with these concepts and added a missing reference to the corresponding appendix section D, where we provide a detailed description of all models.
>
> > 2. Could you explain why the network modules have to be hypernetworks in your setting?
>
> Thank you for pointing out that this was not clearly communicated.
> We clarified the setting we consider in Section 2:
>
> To endow our data generating distributions with compositional structure, for each task we recombine sparse subsets of up to $K$ components from a pool of $M$ modules.
> Each task can be fully described by a  latent code $z$ that specifies the combination of module parameters $(\Theta^{(m)})_{1 \leq m \leq M}$ which parameterizes a task-shared function $g$ with parameters
>
> $\omega(z) = \sum^{M}_{m=1} z^{(m)} \Theta^{(m)}$.
>
> Given this vantage point, hypernetworks are one possible choice to capture this class of compositional problems using neural networks.
> We believe they are a natural choice, as they can be seen as implementing functions with composable parameters using neural networks both to implement the function itself as well as to implement the parameterization of the function using module composition.
> The strong performance of the hypernetworks in our experiments on the compositional preferences and compositional goals grid worlds provide evidence, that this module composition in weight space is indeed an expressive architecture to capture varied compositional structure.

---

> > ### Comment · Reviewer_4knX · 2023-11-20
> > **Thank you for the response**
> >
> > Thanks for your explanation and revision to improve the readability.
> > I would slightly raise my score to 6 but still in a borderline position.

---

### Author Response · Authors · 2023-11-16
**Joint reply to all reviewers**

We thank all reviewers for their constructive criticism and useful suggestions that have helped us improve our paper. We provide below a summary of the major changes made in response to the reviews:

1. We have significantly revised the presentation of the theory in Section 3 of the manuscript by providing an easy-to-follow, motivating example in Section 3.2, substantially simplifying the formalism and as a result making the definitions and theorems considerably easier to parse. In addition, we have restructured the corresponding Appendix A to make the detailed proofs easier to follow.
2. We have improved the clarity of the experimental setting by adding more elaborate explanations of how the experiments verify the theoretically identified conditions for compositional generalization in complex settings with different forms of underlying compositionality. We would like to emphasize the strong empirical performance we observe for our modular hypernetworks in achieving compositional generalization with an untypically large gap towards monolithic baselines like ANIL and MAML.
3. We clarified that related work is discussed in Section 5, changing the section heading to "Related work and discussion" and in addition complement it with an "Extended related work" section in Appendix E.

---

### Comment · Area_Chair_Nibd · 2023-11-17
**Discussion between authors and reviewers**

Dear Reviewers,

Thanks for the reviews. The authors have uploaded their responses to your comments, please check if the rebuttal address your concerns and if you have further questions/comments to discuss with the authors. If the authors have addressed your concerns, please adjust your rating accordingly or vice versa.

AC

---

### Meta-Review · Area_Chair_Nibd · 2023-12-04

**Metareview:**

This paper investigates how to extract modular structures for better compositional generalization(CG). It proposes a teacher-student setting to achieve the goal and provides theoretical analysis on when the compositional structure can be learned.
The paper also demonstrates how meta-learning can be used to discover modulars in a hypernetwork and generalize better than monolithic baselines.

Strengths:
+ The paper theoretically proves that the student can learn underlying modules from the teachers under certain conditions.
+ The proposed method has empirical support.
+ It has potential impact on future work and guide the research on CG and meta learning.

Weaknesses:
- The teacher-student setting requires known GT modular models.
- It is constrained to linear combination of modules in hypernetwork.

**Justification For Why Not Higher Score:**

Due to the weaknesses listed above indicating the limitations of the work.

**Justification For Why Not Lower Score:**

This paper addresses an important problem in ML on how to learn modular structure for compositional generalization. It casts the generalization problem into an identification problem. Both theory and experiments support the claims. I think  the revised paper meets the standard for acceptance.

---

### Decision · Program_Chairs · 2024-01-16

Accept (poster)